# Sublinear Time Quantum Sensitivity Sampling

## Abstract

We present a unified framework for quantum sensitivity sampling, extending the advantages of quantum computing to a broad class of classical approximation problems. Our unified framework provides a streamlined approach for constructing coresets and offers significant runtime improvements in applications such as clustering, regression, and low-rank approximation. Our contributions include:

- $k$-**median and** $k$-**means clustering:** For $n$ points in $d$-dimensional Euclidean space, we give an algorithm that constructs an $\epsilon$-coreset in time $\widetilde{O}(n^{0.5}dk^{2.5}\operatorname{poly}(\epsilon^{-1}))$ for $k$-median and $k$-means clustering. Our approach achieves a better dependence on $d$ and constructs smaller coresets that only consist of points in the dataset, compared to recent results of [Xue, Chen, Li and Jiang, ICML'23].

- $\ell_p$ **regression:** For $\ell_p$ regression problems $\min_{x\in\mathbb{R}^d}\|Ax-b\|_p$ where $A\in\mathbb{R}^{n\times d}$ and $b\in\mathbb{R}^n$, we construct an $\epsilon$-coreset of size $\widetilde{O}_p(d^{\max\{1,p/2\}}\epsilon^{-2})$ in time $\widetilde{O}_p(n^{0.5}d^{\max\{0.5,p/4\}+1}(\epsilon^{-3}+d^{0.5}))$, improving upon the prior best quantum sampling approach of [Apers and Gribling, QIP'24] for all $p \in (0,2)\cup(2,22]$, including the widely studied least absolute deviation regression ($\ell_1$ regression).

- **Low-rank approximation with Frobenius norm error:** We introduce the first quantum sublinear-time algorithm for low-rank approximation that approximates the best rank-$k$ solution to a matrix $A \in \mathbb{R}^{d\times n}$ and does not rely on data-dependent parameters, and runs in $\widetilde{O}(n^{0.5}dk^{0.5}\epsilon^{-1})$ time. Additionally, we present quantum sublinear algorithms for kernel low-rank approximation and tensor low-rank approximation, broadening the range of achievable sublinear time algorithms in randomized numerical linear algebra.

## 1 Introduction

Given a set of points $A = \{a_1, \ldots, a_n\} \subset \mathbb{R}^d$, a universe $X$, and a cost function $\operatorname{cost} : \mathbb{R}^d \times X \to \mathbb{R}_{\geq 0}$, we study the problem of constructing a *coreset* of $A$: a weighted subset $B$ of points along with a nonnegative weight vector $w \in \mathbb{R}_{\geq 0}^{|B|}$ such that

$$\sum_{b\in B} w_b \cdot \operatorname{cost}(b, x) = (1 \pm \epsilon) \cdot \operatorname{cost}(A, x)$$

for all $x \in X$, where $\operatorname{cost}(A, x) = \sum_{i=1}^n \operatorname{cost}(a_i, x)$. A coreset is particularly useful because it enables applying any existing approximation (or exact) algorithm on the smaller summary, yielding a good approximation to the original problem. Applications of coresets span clustering (Chen, 2009; Langberg & Schulman, 2010; Feldman & Langberg, 2011; Varadarajan & Xiao, 2012; Braverman et al., 2022; Huang & Vishnoi, 2020; Braverman et al., 2021; Cohen-Addad et al., 2021; 2022b;a; Huang et al., 2024), graph sparsification (Benczúr & Karger, 1996; Spielman & Teng, 2004; Spielman & Srivastava, 2011; Batson et al., 2012), hypergraph sparsification (Bansal et al., 2019; Kapralov et al., 2022; Jambulapati et al., 2023b; Lee, 2023), $\ell_p$ regression (Drineas et al., 2006; Clarkson, 2005; Dasgupta et al., 2009; Cohen & Peng, 2015; Woodruff & Yasuda, 2023), submodular optimization (Rafiey & Yoshida, 2022; Jambulapati et al., 2023a), generalized linear models (Mai et al., 2021; Munteanu et al., 2022; Musco et al., 2022; Jambulapati et al., 2024), and subspace approximation (Cohen et al., 2015a; 2017; Woodruff & Yasuda, 2025).

Coresets can be constructed via *sensitivity sampling*: the sensitivity of the $i$-th point is defined as

$$s_i = \max_{x \in X} \frac{\mathrm{cost}(a_i, x)}{\mathrm{cost}(A, x)}.$$

Sensitivity sampling draws point $i$ with probability proportional to $s_i$, assigning it weight $1/s_i$ to ensure the estimator is unbiased. The seminal work of Langberg & Schulman (2010) shows that this yields a coreset with a simple and elegant proof: for any *fixed $x \in X$*, sampling proportional to sensitivity ensures low variance, and Bernstein's inequality implies that $O(\epsilon^{-2}S)$ samples suffice, where $S = \sum_{i=1}^n s_i$ is the *total sensitivity*. Recall our goal is to approximate the cost for *all $x \in X$*, to do so, we union bound over a discretization of $X$ with size $\exp(\dim(X))$, where $\dim(X)$ is a notion akin to VC dimension (Feldman & Langberg, 2011). Thus, a coreset can be constructed with $O(\epsilon^{-2} \cdot \dim(X) \cdot S \log(1/\delta))$ samples, where $\delta$ is the failure probability of the algorithm.

Algorithmically, a challenge arises: it is necessary to either compute or efficiently approximate $s_i$ or an upper bound on it. Most of the work in sensitivity sampling focuses on this task, and for many problems it can be achieved in nearly-linear time in $nd$ (Huang & Vishnoi, 2020; Spielman & Srivastava, 2011; Cohen & Peng, 2015; Cohen et al., 2017; Woodruff & Yasuda, 2025). Since scanning the entire dataset already takes $\Omega(nd)$ time, achieving nearly-linear time is close to optimal.

Typically, coresets are constructed for downstream *optimization problems*. For instance, coresets for $\ell_p$ regression help solve the original regression problem (Jambulapati et al., 2022; Adil et al., 2024), and coresets for subspace approximation yield column subset selection for low-rank approximation (Cohen et al., 2017). In certain structured settings, some of these optimization problems admit *sublinear time* algorithms. For example, if the input matrix $A$ is positive semidefinite (PSD) (Musco & Woodruff, 2017; Bakshi et al., 2020) or Toeplitz (Musco & Sheth, 2024), one can obtain a rank-$k$ approximation in $n \cdot \mathrm{poly}(k/\epsilon)$ time, despite $A$ having size $n \times n$.

Can sensitivity sampling—and subsequently solving downstream optimization problems—be accomplished in sublinear time, even without structural assumptions? In this work, we explore this question through the lens of *quantum computing*, analyzing the time complexity of sensitivity sampling under quantum algorithms. Notably, tasks like linear regression, low-rank approximation, and clustering have quantum algorithms running in $o(nd)$ time (Kerenidis & Prakash, 2017; Kerenidis et al., 2019; Gilyén et al., 2022; Shah & Jaiswal, 2025), though these often rely on special input representations that support efficient weighted sampling. Moreover, their runtimes often depend on data-specific parameters such as $\|A\|_F$, condition number $\kappa(A) = \sigma_{\max}(A)/\sigma_{\min}(A)$, or dataset radius. In contrast, we seek quantum algorithms that (1) operate in sublinear time, (2) are *independent of input representation*, and (3) have *runtime independent of data-specific parameters*.

In this work, we provide a generic quantum algorithm applicable to *sensitivity sampling* in general. Throughout the paper, we use $s$ to denote the size of the coreset and let $\mathcal{T}_{\mathrm{sensitivity}}(s, X)$ represent the time to approximate one sensitivity over a set of $s$ points and universe $X$. Our algorithm runs in time $\widetilde{O}(\sqrt{ns}) \cdot \mathcal{T}_{\mathrm{sensitivity}}(s, X)$ and $\widetilde{O}(\sqrt{ns})$ queries to the points in $A$, which implies that as long as $s^{0.5} \cdot \mathcal{T}_{\mathrm{sensitivity}}(s, X) = o(n^{0.5}d)$, we achieve sublinear runtime. Moreover, our algorithm avoids dependence on data-specific parameters: the sample size and sensitivity approximation time depend only on $n$, $d$, $1/\epsilon$, $1/\delta$, and other problem-related parameters (e.g., $k$ in clustering and low-rank approximation, or $p$ in $\ell_p$ regression). We summarize the main result in the following theorem.

**Theorem 1.1** (Informal version of Theorem B.3). *Let $A \in \mathbb{R}^{n \times d}$ and $X$ be a universe, there exists a quantum algorithm that constructs an $\epsilon$-coreset $C$ of expected size $s := O(\epsilon^{-2} \cdot \dim(X) \cdot S \log(1/\delta))$ with probability at least $1 - \delta$, where $\dim(X)$ is the VC dimension of $X$, $S$ is the total sensitivity and $\epsilon, \delta \in (0, 1)$. Moreover, if there exists a classical oracle that can output a constant factor two-sided approximation to one sensitivity over a set of $s$ points and universe $X$ in $\mathcal{T}_{\mathrm{sensitivity}}(s, X)$ time, then the quantum algorithm can be implemented in time $\widetilde{O}(\sqrt{ns} \cdot \mathcal{T}_{\mathrm{sensitivity}}(s, X))$ and $\widetilde{O}(\sqrt{ns})$ queries to the points in $A$.*

Our approach is simple and general: it constructs the sample by uniformly subsampling half of the points, recursively computing approximate sensitivities on this subset, and then resampling based on these estimates. This scheme was first used for leverage score sampling (Cohen et al., 2015b) and in recent quantum linear programming algorithms (Apers & Gribling, 2024). We extend this strategy to sensitivity sampling.

As a key application, we adapt our framework to solve the low-rank approximation problem. Given a matrix $A \in \mathbb{R}^{d \times n}$, the goal is to find matrices $U, V$ of rank $k$ such that

$$\|A - UV^\top\|_F^2 \leq (1 + \epsilon) \cdot \|A - A_k\|_F^2,$$

where $A_k$ is the best rank-$k$ approximation to $A$. We provide a quantum algorithm that constructs a column coreset of $A$, resulting in a low-rank approximation algorithm that runs in time $\widetilde{O}(n^{0.5} d k^{0.5} \epsilon^{-1})$. We note a recent result by Chen et al. (2025), which provides a quantum algorithm for approximating the top-$k$ eigenvectors of a Hermitian matrix. Their method computes an orthonormal basis $W \in \mathbb{R}^{n \times k}$ such that $\|WW^\top - \sum_{i=1}^{k} v_i v_i^\top\| \leq \epsilon$, where $v_i$ is the $i$-th eigenvector of $A$, in time $\widetilde{O}(n^{1.5} k / (\epsilon \gamma))$, where $\gamma$ is the spectral gap between $\lambda_k$ and $\lambda_{k-1}$. While powerful, their method targets spectral norm error and depends on $\gamma^{-1}$. In contrast, our algorithm selects and reweights subsets of rows and columns and approximates the Frobenius-norm optimal rank-$k$ solution, with no dependence on $\gamma$. This makes our method suitable for downstream applications for small $\gamma$.

Our algorithm for $(k, p)$-clustering has further implications for the *data selection pipeline* used in training foundation models. As discussed in Axiotis et al. (2024), if the loss function $\ell$ and all $k$-center solutions satisfy the $(p, \Lambda)$-well-behaved property, then a subset of $s = O(\epsilon^{-2})$ points suffices for training or fine-tuning. The pipeline proceeds as follows: (1) compute $k$ centers $x = (x_1, \ldots, x_k)$ using a clustering algorithm, and (2) sample $s$ points using the loss values $\ell(x_i)$ to obtain a coreset.

Using our quantum algorithm for $(k, p)$-clustering, we first construct a coreset of size $\mathrm{poly}(k/\epsilon)$ in time $\widetilde{O}(n^{0.5} d \cdot \mathrm{poly}(k/\epsilon))$, then solve for the centers $x_1, \ldots, x_k$ using only the coreset. The second round of sampling requires at most $k$ queries to the loss function and can also be implemented in $\widetilde{O}(n^{0.5} d \cdot \mathrm{poly}(k/\epsilon))$ time. Classical algorithms for this pipeline would require $\Omega(n)$ time. Hence, our method is the first sublinear-time quantum algorithm for data selection pipelines.

We summarize our results in the following tables. Table 1 compares our coreset construction runtimes with prior work, and Table 2 compares runtimes for solving the corresponding optimization problems.

|  | Reference | Previous | Ours |
|---|---|---|---|
| $k$-Median Clustering | Xue et al. (2023) | $n^{0.5} d^{1.5} k^{0.5}$ | $n^{0.5} d k^{2.5}$ |
| $k$-Means Clustering | Xue et al. (2023) | $n^{0.5} d^{1.5} k^{0.5}$ | $n^{0.5} d k^{2.5}$ |
| $(k, p)$-Clustering | Xue et al. (2023) | $n^{0.5} d^{1.5} k^{0.5}$ | $n^{0.5} d k^{2.5}$ |
| $\ell_{p \neq 2}$ Regression † | Apers & Gribling (2024) | $n^{0.5} d^7$ | $n^{0.5} d^{(0.5 \vee p/4)+1.5}$ |
| $(k, p < 2)$-Subspace Approx. | Woodruff & Yasuda (2025) | $nd$ | $n^{1-p/4} d k^{p/4}$ |
| $(k, p \geq 2)$-Subspace Approx. | Woodruff & Yasuda (2025) | $nd$ | $n^{1-1/p} d k^{0.5}$ |

Table 1: Comparison of running times for constructing an $\epsilon$-coreset for the respective problems. We set $\epsilon = O(1)$, assume $n \gg d, k$ and ignore all dependencies on functions that only depend on $p$ for simplicity of presentation. For clustering and $\ell_p$ regression, we compare against prior fastest quantum algorithms, while for subspace approximation, we compare against prior fastest classical algorithms as we are unaware of quantum algorithms for these problems. †: We use $a \vee b$ to denote $\max\{a, b\}$.

**Our contributions.** We summarize our main contributions below:

- We introduce a general quantum weighted sampling framework. Given weights $w \in \mathbb{R}_{\geq 0}^n$ satisfying mild conditions and access to a classical oracle that approximates the weight of a point over a small set, the framework constructs a coreset using $\widetilde{O}(\sqrt{n \cdot \mathrm{sum}(w)})$ queries to the input, where $\mathrm{sum}(w) \geq \sum_{i=1}^n w_i$ is an upper bound on the total weight. We show that sensitivity, leverage scores, and Lewis weights all meet these conditions, implying that coreset construction with these weights can be accelerated within our framework.

- We design the first sublinear time quantum algorithms for several fundamental low-rank approximation tasks: Frobenius-norm approximation, PSD and kernel low-rank approximation, and tensor low-rank approximation. Our algorithms are purely sampling-based, and avoid dependence on data-dependent parameters.

- We develop improved quantum algorithms for $(k, p)$-clustering in the high-dimensional regime $d \gg k$, and further demonstrate how our framework can accelerate data selection pipelines for training foundation models.

| | Reference | Previous | Ours |
|---|---|---|---|
| Low-Rank | Cohen et al. (2017) | $nd$ | $n^{0.5}dk^{0.5}$ |
| PSD Low-Rank | Bakshi et al. (2020) | $nk^{\omega-1}$ | $n^{0.75}k^{2.25}$ |
| Kernel Low-Rank | Bakshi et al. (2020) | $nk\,\mathcal{T}_{\mathsf{K}}$ | $n^{0.75}k^{1.25}\mathcal{T}_{\mathsf{K}}$ |
| Tensor Low-Rank: Rank-$k$ | Song et al. (2019) | $n^3 + 2^{k^2}$ | $n^2k^{0.5} + 2^{k^2}$ |
| Tensor Low-Rank: Bicriteria | Song et al. (2019) | $n^3$ | $n^2k^{0.5}$ |

Table 2: Comparison of running times for a variety of low-rank approximation problems. For all of these problems, we only compare with the prior best classical algorithms, as they are either not studied in the context of quantum algorithms, or the respective quantum algorithms require that the input is given in the form of a data structure, or have data-dependent parameters in the running time. We assume all matrices/tensors are dense. We ignore lower order terms for ease of comparison. For kernel low-rank approximation, we use $\mathcal{T}_{\mathsf{K}}$ to denote the time of evaluating kernel function on any two data points.

**Roadmap.** In Section 2, we provide a technical overview of the main results, including our generic algorithm for constructing coresets and specific applications to low-rank approximation. In Section 3, we summarize our results and discuss open problems. Section A presents preliminary definitions and notation. In Section B, we describe a generic weighted sampling algorithm for coreset construction and discuss its adaptations to regression and subspace approximation. Section C demonstrates how to use weighted sampling to generate a column subset of a matrix and apply it to low-rank approximation. Section D shows how Grover search can be used to accelerate Nyström approximation of kernel matrices, improving upon the runtime of Bakshi et al. (2020). In Section E, we extend our approach to $(k, p)$-subspace approximation. Section F provides algorithms for low-rank approximation of third-order tensors in the Frobenius norm. Section G presents an improved quantum algorithm for constructing coresets for $(k, p)$-clustering and applies it to data selection. In Section H, we establish a quantum query lower bound for additive-multiplicative spectral approximation, a key subroutine for computing low-rank approximations. In Section I, we provide a detailed discussion on simulating query access to a random string with QRAM. In Section J, we give the query complexity of our algorithms which is independent of QRAM and the type of queries to the input.

**Quantum computation model.** We adopt the standard quantum computation model used in, e.g., Apers & De Wolf (2022); Apers & Gribling (2024). This model supports quantum subroutines operating on $O(\log n)$ qubits, allows quantum queries to the input, and grants access to a quantum-read/classical-write RAM (QRAM) of size $\mathrm{poly}(n)$ bits. Each quantum read or classical write to QRAM incurs unit cost. We measure *time complexity* by the number of QRAM operations, and *query complexity* by the number of input queries made by the algorithm. QRAM is a common model studied in quantum algorithms, however it is known that practically realizing the QRAM architecture is challenging.

## 2 TECHNICAL OVERVIEW

We give an overview of our techniques in this section. In Section 2.1, we introduce our recursive sampling framework for sensitivity sampling, based on Grover search. In Section 2.2, we generalize the quantum sensitivity sampling framework via approximators. In Section 2.3, we design quantum, sublinear time algorithms for low-rank approximation that are based purely on sampling rows and columns. In Section 2.4, we further extend the sampling-based low-rank approximation to the tensor setting. Finally in Section 2.5, we discuss our coreset algorithm for $(k, p)$-clustering and its advantages over prior constructions.

### 2.1 SENSITIVITY SAMPLING VIA GROVER SEARCH

One of the primary advantages of quantum algorithms over their classical counterparts is their ability to search and sample more efficiently. The search procedure developed by Grover (Grover, 1996) addresses the database search problem: given a function $f : [n] \to \{0, 1\}$, we aim to list up to $m$ indices for which $f(i) = 1$. Assuming access to an oracle that, given an index $i$, outputs the value

$f(i)$, Grover's seminal work shows that, instead of requiring $n$ queries to the oracle, the problem can be solved using only $O(\sqrt{mn})$ oracle calls with quantum computation. This provides a notable advantage as long as $m < n$, which is often the case in applications. Grover search has since been utilized to achieve speedups in problems such as edit distance (Boroujeni et al., 2021; Gibney et al., 2024), solving graph Laplacian systems (Apers & De Wolf, 2022), and solving linear programs (Apers & Gribling, 2024). In particular, Apers & Gribling (2024) develops a method to sample from the leverage score distribution of an $n \times d$ matrix $A$, in time $O(n^{0.5}d^{1.5})$. For tall, skinny matrices, this approach leads to a runtime that is sublinear in the input size of $A$. Subsequently, the authors construct spectral approximations of $A$ to speed up various essential procedures within a linear program solver.

The key procedure they utilize is a quantum sampling algorithm based on Grover search: associate each item with a value $p_i \in [0, 1]$, and the goal is sample a subset with the probability of $i$-th item being $p_i$. Given $p_i$'s, we can sample the subset in $\widetilde{O}(\sqrt{n \sum_{i=1}^n p_i})$ time and if $\sum_i^n p_i \leq o(n)$, then we can achieve this goal in sublinear in $n$ time. However, this sampling procedure requires an oracle that returns the value of $p_i$ upon query, akin to the oracle for $f(i)$ in Grover search. We refer to Lemma A.14 for a formal statement.

Since the $i$-th leverage score of $A$ is defined as $a_i^\top (A^\top A)^\dagger a_i$, where $M^\dagger$ is the pseudoinverse of matrix $M$, implementing the oracle by computing the Gram matrix $A^\top A$ and its pseudoinverse is prohibitively slow. To address this issue, Apers & Gribling (2024) observes that the algorithm due to Cohen et al. (2015b) can implement such an oracle efficiently: this algorithm proceeds by recursively halving rows — it first uniformly samples half of the rows of $A$, denoted by $A'$, then recursively computes the leverage score matrix of $A'$. For an $n \times d$ matrix $A$, it suffices to sample $O(d \log d \epsilon^{-2})$ rows according to leverage scores; hence the sampled matrix $SA' \in \mathbb{R}^{d \log d \times d}$ is small where $S \in \mathbb{R}^{d \log d \times (n/2)}$ is the sampling matrix that selects the sampled rows and scales them properly. In fact, $SA'$ serves as a sketch for the leverage score of matrix $A$ with $a_i^\top (A'^\top S^\top S A')^\dagger a_i = (1 \pm \epsilon) \cdot a_i^\top (A^\top A)^\dagger a_i$ for all $i$. Thus, an oracle can be efficiently implemented by computing $(A'^\top S^\top S A')^\dagger$ in $\widetilde{O}(d^\omega)$ time, and by leveraging a trick from Spielman & Srivastava (2011), the quantity $a_i^\top (A'^\top S^\top S A')^\dagger a_i = \left\| (A'^\top S^\top S A')^{\dagger/2} a_i \right\|_2^2$ can be accelerated using a Johnson-Lindenstrauss transform (Johnson & Lindenstrauss, 1984). Consequently, this approach results in an algorithm that constructs a leverage score sampler for $A$ in time $\widetilde{O}(n^{0.5}d^{1.5}\epsilon^{-1} + d^\omega)$, with the sum of probabilities being $O(s)$.

Can we extend the leverage score sampling algorithm of Apers & Gribling (2024) to generic sensitivity sampling? The first hope is that, instead of sampling directly according to sensitivities, it might be sufficient to sample based on an overestimate of sensitivities. Consider the following simplified algorithm: form $A'$ by sampling each point of $A$ with probability $1/2$ uniformly, and define the generalized sensitivity as $s_i(A, A') = \max_{x \in X, \text{cost}(A',x) \neq 0} \frac{\text{cost}(a_i,x)}{\text{cost}(A',x)}$, i.e., we change the denominator to $\text{cost}(A', x)$. Note that this is *not* necessarily an overestimate of $s_i$. To see this, let $x^*$ be the point that realizes the sensitivity for $s_i$. If $\text{cost}(A', x^*) = 0$, then $s_i(A, A')$ will not be realized by $x^*$, and it is possible that $s_i > s_i(A, A')$. On the other hand, we can see that $s_i(A, A' \cup \{a_i\})$ serves as an overestimate. To understand this, consider the case where $s_i(A, A')$ does not hold: if $\text{cost}(A', x^*) = 0$, then either $\text{cost}(a_i, x^*) = 0$ and $s_i = s_i(A, A' \cup \{a_i\}) = 0$, or $\text{cost}(a_i, x^*) \neq 0$ and $s_i(A, A' \cup \{a_i\}) = \frac{\text{cost}(a_i,x^*)}{\text{cost}(a_i,x^*)} = 1$, an upper bound on any $s_i$. If $\text{cost}(A', x^*) \neq 0$, then $\frac{\text{cost}(a_i,x^*)}{\text{cost}(A',x^*)} \geq \frac{\text{cost}(a_i,x^*)}{\text{cost}(A,x^*)}$, as the denominator for $A'$ is smaller. We note that $s_i(A, A' \cup \{a_i\})$ is in fact the overestimate used by Cohen et al. (2015b) to obtain their initial uniform sampling bound.

The recursive framework follows directly: uniformly sample half of the points $A'$, then compute a coreset of $A'$, called $C$. To compute the overestimates of $s_i$, we use $s_i(A, C \cup \{a_i\})$, which is efficient since $C$ is a small-size coreset. Note that $s_i(A, C \cup \{a_i\})$ is a valid approximation of $s_i(A, A' \cup \{a_i\})$—as $C$ is a coreset of $A'$, it approximates the cost of $A'$ with respect to all $x \in X$, and they have the same kernel.[1] Moreover, it is not hard to see that $C \cup \{a_i\}$ is also a coreset of $A' \cup \{a_i\}$, and thus the sensitivity is preserved. To summarize, in each round of recursion, we are given a size-$s$ coreset, and assuming we can approximate each $s_i(A, C \cup \{a_i\})$ in $\mathcal{T}_{\text{sensitivity}}(s, d)$ time, then the overall runtime is $\widetilde{O}(\sqrt{ns}) \cdot \mathcal{T}_{\text{sensitivity}}(s, d)$, with recursion depth at most $\log n$ as we halve the points at each step, giving the desired runtime for sensitivity sampling.

---

[1]Given a set of points $A$ and a cost function, the kernel of $A$ is $\ker(A) = \{x \in X : \text{cost}(A, x) = 0\}$.

## 2.2 GENERIC WEIGHTED SAMPLING VIA APPROXIMATOR

While the preceding algorithm handles *all* sensitivity sampling, in many applications, the exact sensitivities can be difficult to compute, and thus proxies are often sought as efficient alternatives. Take the $\ell_p$ regression problem as an example, where the sensitivity is $s_i = \max_{x \in \mathbb{R}^d, Ax \neq 0} \frac{|a_i^\top x|^p}{\|Ax\|_p^p}$. For $p = 2$, this corresponds to the leverage score, which can be quickly approximated. However, for general $p$, this is more complex, and algorithms for $\ell_p$ sensitivities tend to be less efficient than those for leverage scores (Padmanabhan et al., 2023). Instead, constructing a coreset for $\ell_p$ regression is typically done *not* via sensitivity sampling, but through *Lewis weights sampling* (Bourgain et al., 1989; Ledoux & Talagrand, 1991; Talagrand, 1995; Schechtman & Zvavitch, 2001; Cohen & Peng, 2015; Woodruff & Yasuda, 2023). These weights are defined as the fixed-point solution for the following equation: $w_i^{2/p} = a_i^\top (A^\top W^{1-2/p} A)^{-1} a_i$, where $w_i$ represents the $i$-th leverage score of the matrix $W^{1/2-1/p} A$. Lewis weights have several desirable properties, such as $\sum_{i=1}^n w_i = d$, and they serve as proper upper bounds for $\ell_p$ sensitivities for all $p \in (0, \infty)$. Moreover, Lewis weights can be approximated in nearly-linear time (Cohen & Peng, 2015; Lee, 2016; Jambulapati et al., 2022; Fazel et al., 2022; Apers et al., 2024).

To adapt our sensitivity sampling framework to work with Lewis weights sampling, we encounter a notable challenge: given a coreset $B$ of $A$, it is guaranteed that any vector in the subspace of $A$ has its $\ell_p$ norm preserved by $B$, but the Lewis weights are not defined purely in terms of the $\ell_p$ norm of vectors in the subspace. Instead, they measure the $\ell_2$ norm of the subspace after a density transformation induced by $W^{1/2-1/p}$. Consequently, it might well be the case that $B$ is a coreset of $A$, and the Lewis weights of $A$ are not preserved by $B$. On the other hand, we can instead define a notion of an $\epsilon$-*approximator* of $A$: we say $B$ is an $\epsilon$-approximator of $A$ for $\ell_p$ regression if $B$ is a coreset and $(1 - \epsilon) A^\top W_A^{1-2/p} A \preceq B^\top W_B^{1-2/p} B \preceq (1 + \epsilon) A^\top W_A^{1-2/p} A$, where $W_A, W_B$ are the diagonal Lewis weights matrices for $A$ and $B$. Note that this is a different approximation notion than that of a coreset, as the cost becomes *global* rather than local: for generic sensitivity-based arguments, one relies on the fact that adding a single point to the set will not affect the weights of other points, and hence if $B$ is a coreset of $A$, then $B \cup \{p\}$ is also a coreset of $A \cup \{p\}$, but this is not true for an approximator of $A$, as adding a single row to both $A$ and $B$ would potentially affect the weights to all existing rows. In Cohen & Peng (2015), they provide a classical recursive sampling algorithm by noting that, if we sample according to the generalized Lewis weights with respect to an approximator, then the resulting weighted sample is *also* an approximator. We further abstract their construction, and provide the most general framework for quantum sublinear weighted sampling: let $w(A, B) \in \mathbb{R}_{\geq 0}^{|A|}$ be the generalized weights of $A$ with respect to $B$, we say $B$ is an $\epsilon$-approximator of $A$ if for any $C$ and any $i \in [n]$, we have $w_i(C, B) = (1 \pm \epsilon) w_i(C, A)$. We then need three sufficient conditions:

- Consistent total weights: For any subset $S \subseteq [n]$, $\sum_{i \in S} w_i(A, A) \leq \text{sum}(w)$, where $\text{sum}(w)$ is a finite upper bound on the sum of weights. When the weight is sensitivity, $\text{sum}(w)$ is simply the total sensitivity;

- Uniform sampling bound: Take any uniform subset $A' \subseteq A$, define the new weights $w_i'(A, A')$ as $w_i(A, A')$ if $a_i \in A'$ and $w_i(A, A' \cup \{a_i\})$ otherwise. Then $w_i'(A, A') \geq w_i(A, A)$ for all $i \in [n]$;

- Importance sampling bound: Suppose we sample according to $q_i = \min\{1, \alpha \cdot w_i(A, A)\}$ for some $\alpha \geq 1$, and reweight the sample by $1/q_i$, then with probability at least $1 - \delta$, the weighted sample is an $\epsilon$-approximator of $A$ of size at most $\alpha \cdot \text{sum}(w) \log(1/\delta)$.

Let $s = O(\alpha \cdot \text{sum}(w) \log(1/\delta))$. We obtain an algorithm that computes an $\epsilon$-coreset in the desired $\widetilde{O}(\sqrt{ns}) \cdot \mathcal{T}_{\text{sensitivity}}(s, d)$ time. Thus, by using weighted sampling with Lewis weights, we achieve a runtime of $\widetilde{O}_p(n^{0.5} d^{(0.5 \vee p/4)+1} (\epsilon^{-3} + d^{0.5}))$ for generating a coreset for $\ell_p$ regression. This improves upon the prior quantum algorithm for Lewis weights sampling that is based on *iterating leverage scores* (Apers et al., 2024), with a runtime of $\widetilde{O}_p(n^{0.5} d^7 \epsilon^{-3})$. Our algorithm provides a speedup for any $p \in (0, 2) \cup (2, 22]$ (which includes the popular $\ell_1$ regression), but it is worth noting that the main purpose of the work of Apers & Gribling (2024) is to estimate Lewis weights up to $p = O(\log n)$ as they use it as a subroutine for solving linear programs, so their algorithm has no $p$ dependence on $d$. Nevertheless, we provide a completely different sampling algorithm to construct an $\ell_p$ regression coreset that is particularly suitable for small $p$.

### 2.3 PURE-SAMPLING FRAMEWORK FOR LOW-RANK APPROXIMATION

Given $A \in \mathbb{R}^{d \times n}$, the rank-$k$ low-rank approximation problem seeks to find a pair of matrices $U \in \mathbb{R}^{d \times k}$, $V \in \mathbb{R}^{n \times k}$ such that $\|A - UV^\top\|_F^2 \leq (1 + \epsilon)\|A - A_k\|_F^2$, where $A_k$ is the best rank-$k$ approximation of $A$. Low-rank approximation is closely related to the $(k, 2)$-subspace approximation coreset: let $\mathcal{F}_k \subset \mathbb{R}^d$ be the set of all $k$-dimensional subspaces in $\mathbb{R}^d$, and define $\mathrm{cost}(a_i, x) = \|(I - P_x)a_i\|_2^2$ where $P_x$ is the orthogonal projection onto $x \in \mathcal{F}_k$. If we obtain a coreset $C$ for $A$, then we have for any $k$-dimensional orthogonal projection $P_x$, $\|(I - P_x)C\|_F^2 = (1 \pm \epsilon)\|(I - P_x)A\|_F^2$, which is sufficient to show that choosing $P_x$ as the projection onto $C_k$ will give the desired low-rank approximation (Cohen et al., 2017). Moreover, instead of $(k, 2)$-subspace sensitivities, one could sample according to the *ridge leverage scores*, which can be computed quickly. To adapt our weighted sampling framework, we need to identify the $\epsilon$-approximator for ridge leverage score, which is a coreset of $A$ and

$$(1 - \epsilon)AA^\top - \epsilon\lambda_{A_k}I \preceq CC^\top \preceq (1 + \epsilon)AA^\top + \epsilon\lambda_{A_k}I,$$

where $\lambda_{A_k} = \|A - A_k\|_F^2/k$. Thus, our framework gives an algorithm that runs in time $\widetilde{O}(n^{0.5}dk^{0.5}\epsilon^{-1})$.

While one might be satisfied with the ridge leverage score solution to low-rank approximation, more complexity arises if we aim to recover the solution through the subsampled columns. In particular, if we let $C \in \mathbb{R}^{d \times s}$ denote the weighted subset of columns of $A$ sampled by ridge leverage scores for $s = O(k \log k\epsilon^{-2})$, it is guaranteed that $\min_{X:\mathrm{rank}(X)\leq k} \|CX - A\|_F^2 \leq (1 + \epsilon)\|A_k - A\|_F^2$. Constructing an optimal $X$ would require computing $P_k(C^\dagger A)$ where $P_k$ is the projection onto the top-$k$ principal components. Directly computing $C^\dagger A$ is of course too expensive, and standard approaches mostly involve using an *oblivious subspace embedding* (OSE) matrix, a random matrix that approximates the cost of all regression problems. Matrices such as CountSketch (Charikar et al., 2002; Clarkson & Woodruff, 2013) could be applied in time $\mathrm{nnz}(A)$, but this is already too slow for our purpose. We address this with a pure-sampling framework for low-rank approximation: we demonstrate that it is possible to recover (or approximate) the solution $X$ via leverage score sampling.

In particular, for the regression problem $\min_{X:\mathrm{rank}(X)\leq k} \|CX - A\|_F^2$, one could sample according to the leverage score distribution of $C$ and solve the subsampled regression problem $\min_{X:\mathrm{rank}(X)\leq k} \|SCX - SA\|_F^2$. Standard leverage score guarantees ensure that the optimal solution to the subsampled regression closely approximates the original problem (Lemma A.13). Because of this fact, we can show that there exists a good solution $\widehat{X}$ in the row span of matrix $SA$; hence it is enough to solve the regression problem $\min_{Y:\mathrm{rank}(Y)\leq k} \|A - CYSA\|_F^2$, and we further speed up the algorithm by employing two leverage score sampling matrices $T_1$ and $T_2$ on the left and right accordingly. Consider the new subsampled regression problem: $\min_{Y:\mathrm{rank}(Y)\leq k} \|T_1AT_2 - T_1CYSAT_2\|_F^2$, and observe that we can compute the subsampled $A$ in *sublinear in* $n, d$ time, because $T_1AT_2$ and $SAT_2$ all amount to selecting a $\mathrm{poly}(k/\epsilon)$ subset of entries of $A$, which, assuming random access to the entries of $A$, can be done in the same order of time. This pure-sampling approach contrasts with OSE-based methods, which generally require reading all entries of $A$.

### 2.4 APPROXIMATE REGRESSION VIA SAMPLING RESPONSES

For matrix low-rank approximation and its variants, ridge leverage score sampling is the crucial tool to compute a good approximate solution. Can we extend the framework to solve *tensor* low-rank approximation? Unfortunately, even for a 3rd order tensor $A \in \mathbb{R}^{n \times n \times n}$, it is not always the case that it admits a low-rank approximation, due to the so-called border rank issue (De Silva & Lim, 2008). Even when the low-rank approximation exists, variants of Strong Exponential Time Hypothesis (SETH) rule out polynomial time algorithms to approximate the tensor rank of $A$ (Song et al., 2019). If one relaxes the problem by allowing the output to be a higher-rank solution (bicriteria solution) or a running time that depends exponentially on $k$ and $1/\epsilon$ (fixed-parameter tractable, i.e., FPT), then Song et al. (2019) provides algorithms with leading running time term being $\mathrm{nnz}(A)$. Their core algorithm is as follows: for tensor $A \in \mathbb{R}^{n \times n \times n}$, let $A_1, A_2, A_3 \in \mathbb{R}^{n \times n^2}$ be matrices such that the 1st, 2nd, and 3rd dimensions of the array are preserved, while the other 2 dimensions are collapsed and flattened into a dimension of size $n^2$. They then apply OSEs $S_1, S_2, S_3$ with only $\mathrm{poly}(k/\epsilon)$ columns to form $A_1S_1, A_2S_2$ and $A_3S_3$.

Although one might attempt to replace the OSEs $S_1, S_2$, and $S_3$ with leverage score matrices for $A_1, A_2$, and $A_3$, *this approach, unfortunately, does not work*. The argument of Song et al. (2019) is as follows: suppose the optimal rank-$k$ approximation $A_k$ exists, then $A_k = \sum_{i=1}^{k} U_i^* \otimes V_i^* \otimes W_i^*$. To reduce the problem dimension, the goal is to demonstrate that a good approximate solution exists in the column span of $A_1 S_1$ and $A_2 S_2$. In particular, suppose we have access to $V^*$ and $W^*$, set $Z_1 \in \mathbb{R}^{k \times n^2}$ where each row $i$ is $V_i^* \otimes W_i^*$, then it is not hard to see that the optimal $U^*$ could be recovered by solving $\min_{U \in \mathbb{R}^{n \times k}} \|U Z_1 - A_1\|_F^2$, as $\|U Z_1 - A_1\|_F^2 = \|\sum_{i=1}^{k} U_i \otimes V_i^* \otimes W_i^* - A\|_F^2$. The multiple response regression problem above can then be accelerated by applying an OSE on the right and instead solving $\min_U \|U Z_1 S_1 - A_1 S_1\|_F^2$, where the optimal solution has the closed form $\widehat{U} = A_1 S_1 (Z_1 S_1)^\dagger$. This establishes that $\widehat{U}$ is in the column span of $A_1 S_1$.

For sampling, this setup is more challenging. If we were to replace $S_1$ with the leverage score sampling matrix, we would require the *leverage score matrix of $Z_1$* in order to preserve the cost of the optimal solution. Thus, we could only argue for the correctness of this approach if $S_1$ is chosen according to the leverage score of an unknown matrix $Z_1$, which is unclear how to achieve. On the contrary, we do have access to the response matrix $A_1$, and one might wonder if sampling directly from $A_1$ is sufficient. However, a simple counterexample demonstrates that this approach fails: suppose $A_1$ is a single column equal to $e_n$, and the design matrix $Z_1$ is $e_i + e_n$ for $i$ randomly chosen from 1 to $n-1$. Any sampling scheme based on $A_1$ will likely sample the $n$-th entry but miss the $i$-th entry with high probability. This would lead to a solution on the original problem that has twice the optimal cost.

Surprisingly, we show that this 2-approximation is almost as bad as one can get: if one instead samples from the ridge leverage score distribution of $A_1$, then there exists a solution $\widehat{U}$ in the column span of $A_1 S_1$ ($S_1$ is the ridge leverage score sampling matrix of $A_1$) such that $\|\widehat{U} Z_1 - A_1\|_F^2 \leq (2 + \epsilon) \cdot \min_U \|U Z_1 - A_1\|_F^2$. This result is particularly surprising as one might expect an adversarial choice of $A_1$ that would disrupt ridge leverage score sampling. However, ridge leverage scores provide the so-called projection-cost preserving guarantee: for any rank-$k$ projection $P$, we have that $(1 - \epsilon)\|(I - P)A_1\|_F^2 \leq \|(I - P)A_1 S_1\|_F^2 \leq (1 + \epsilon)\|(I - P)A_1\|_F^2$, where setting $P_k$ as the projection onto the top-$k$ principal components of $A_1 S_1$ minimizes $\|(I - P_k)A_1 S_1\|_F^2$. Additionally, the optimal cost of the regression can be bounded by $\|[A_1]_k - A_1\|_F^2$, the best rank-$k$ approximation to $A_1$. Setting $\widehat{U} = P_k A_1 Z_1^\dagger$, we get

$$
\begin{aligned}
\|\widehat{U} Z_1 - A_1\|_F^2 &= \|P_k A_1 Z_1^\dagger Z_1 - A_1\|_F^2 \\
&= \|(P_k A_1 - A_1)(Z_1^\dagger Z_1) + A_1(I - Z_1^\dagger Z_1)\|_F^2 \\
&= \|(P_k A_1 - A_1)(Z_1^\dagger Z_1)\|_F^2 + \|A_1(I - Z_1^\dagger Z_1)\|_F^2 \\
&\leq \|(I - P_k)A_1\|_F^2 + \|A_1(I - Z_1^\dagger Z_1)\|_F^2 \\
&\leq (1 + \epsilon)\,\mathrm{OPT} + \mathrm{OPT} \\
&= (2 + \epsilon)\,\mathrm{OPT},
\end{aligned}
$$

where $\mathrm{OPT} := \min_U \|U Z_1 - A_1\|_F^2$, and we use the Pythagorean theorem in the proof, along with the fact that $\|A_1 - A_1 Z_1^\dagger Z_1\|_F^2$ is the optimal solution. To see $\widehat{U}$ is in the column span of $A_1 S_1$, it is enough to observe that $P_k$ is the projection onto the top-$k$ principal components of $A_1 S_1$, and hence $\widehat{U}$ is in the column span of $P_k$, a subset of the column span of $A_1 S_1$. This shows that as long as we sample according to the ridge leverage score distribution, we can still obtain a $(2 + \epsilon)$-approximate solution. Moreover, for 3rd order tensor low-rank approximation, we would only invoke ridge leverage score sampling on $A_1$ and $A_2$, as the components of the design matrix reside within the column span of both $A_1 S_1$ and $A_2 S_2$, making the problem tractable. We can, in turn, employ fast (classical) tensor leverage score sampling algorithms to achieve an overall approximation ratio of $(4 + \epsilon)$ with a significantly improved running time of $\widetilde{O}(n^2 k^{0.5}/\epsilon + n\,\mathrm{poly}(k/\epsilon))$ for dense tensors.

## 2.5 IMPROVED CORESET FOR CLUSTERING WITH APPLICATIONS

We also design an improved quantum algorithm for constructing an $\epsilon$-coreset of $(k, p)$-clustering. In contrast to the recursive sampling framework we developed in the preceding discussions, our algorithm could be viewed as a quantum implementation of Huang & Vishnoi (2020), where the idea

is to first compute a set of approximate $k$-centers, then perform sensitivity samplings on top of it. Why could our recursive sampling framework not be applied here? This is because the sensitivities of $(k, p)$-clustering can only be *overestimated*, and these overestimates in general do not satisfy the uniform sampling bound. In fact, a closer examination of our analysis shows that during the intermediate stages in the recursive sampling, we would need the sensitivities to be approximated in a *two-sided* fashion, i.e., let $s_i$ be the exact sensitivities. We require the approximate sensitivities $\widetilde{s}_i$ to satisfy $(1-\epsilon)s_i \leq \widetilde{s}_i \leq (1+\epsilon)s_i$. Nevertheless, we design a sensitivity sampling algorithm for $(k, p)$-clustering that is based on Huang & Vishnoi (2020), that computes a coreset of size $\widetilde{O}_p(k^5 \epsilon^{-5p-15})$ in time $\widetilde{O}_p(n^{0.5} dk^{2.5} \epsilon^{-2.5p-7.5})$. Compared to the previous work of Xue et al. (2023) where they obtain a coreset in $\widetilde{O}_p(n^{0.5} d^{1.5} k^{0.5} \epsilon^{-(p/2 \vee 1)})$ time,

- Our algorithm outputs a weighted subset of points $B \subseteq A$, as our coreset. In contrast, Xue et al. (2023) adapts an algorithm of Cohen-Addad et al. (2021), in which the coreset consists of weighted points from $A$ *and all bicriteria approximate centers*. Thus, composing the coreset from Xue et al. (2023) with any optimal-sized coreset algorithm (Huang et al., 2024) will also include points not in $A$;

- Our algorithm outputs a coreset of size $\widetilde{O}_p(k^5 \epsilon^{-5p-15})$, while Xue et al. (2023) outputs a coreset of size $\widetilde{O}_p(dk\epsilon^{-(2 \vee p)})$. This means to obtain an optimal-sized coreset of size $\widetilde{O}_p(k^{\frac{2p+2}{p+2}} \epsilon^{-2})$ by running the algorithm of Huang et al. (2024) on top of our coreset, we can achieve the result with an additional $\widetilde{O}_p(d \operatorname{poly}(k, \epsilon^{-p}))$ time, while Xue et al. (2023) would need $\widetilde{O}_p(d^2 \operatorname{poly}(k, \epsilon^{-p}))$ time.

As an application, we demonstrate that $(k, p)$-clustering can be used to bootstrap the construction of the data selection pipeline (Axiotis et al., 2024), as it enables the computation of approximate $k$-centers in sublinear time. Furthermore, we show that the quantum techniques developed for $(k, p)$-clustering can also be leveraged to obtain a sublinear-time quantum algorithm for data selection. We defer a more detailed discussion to Section G.

## 3 CONCLUSION

We present a quantum, sublinear-time algorithm for weighted sampling that yields a broad range of results in coreset construction. These include $(k, p)$-clustering, $\ell_p$ regression, $(k, p)$-subspace approximation, and low-rank approximation. For the low-rank approximation problem, we design specialized algorithms for multiple settings, including Frobenius norm error minimization, PSD low-rank approximation, kernel-based low-rank approximation, and tensor low-rank approximation. For $(k, p)$-clustering, we develop an improved quantum coreset construction that offers better dependence on the data dimension $d$, and we generalize this framework to address the data selection problem for foundation models. We highlight three major open problems arising from our work:

- **Two-sided approximation for clustering sensitivities.** Unlike regression and low-rank approximation, where coresets can be constructed efficiently via leverage scores or Lewis weights, the approximate sensitivities used in clustering are only known to be *upper bounds*. This asymmetry significantly limits the applicability of the recursive sampling framework to clustering. It remains an open question whether one can design algorithms that compute *two-sided* approximations to clustering sensitivities, thereby unifying clustering within our weighted sampling framework.

- **Quantum algorithms for Frobenius norm tensor low-rank approximation.** While we achieve a $(1 + \epsilon)$-approximation for matrix low-rank approximation in sublinear time, the scenario is more complex for tensors. As discussed in Section 2.4, for 3rd-order tensors, we obtain only a $(4 + \epsilon)$-approximation, and for general $q$-th order tensors, a $(2^{q-1} + \epsilon)$-approximation. A compelling open question is whether one can design a sublinear-time quantum algorithm—with potentially worse running time—that achieves a $(1 + \epsilon)$-multiplicative approximation for tensor low-rank approximation.

- **Query lower bounds for coreset construction.** In Section H, we establish a quantum query lower bound for computing additive-multiplicative spectral approximations, which are sufficient for low-rank approximation. An intriguing direction for future research is to generalize this lower bound to broader classes of coreset constructions.

## ETHICS STATEMENT

This paper studies novel quantum algorithm for sensitivity sampling which could lead to improvement to various machine learning problems. We don't believe there are significant ethics concerns need to be addressed.

## REPRODUCIBILITY STATEMENT

As this paper is theoretical in nature, we include complete proofs in the Appendix. In particular, we prove the generic sensitivity sampling result in Section B, low-rank approximation in Section C, kernel low-rank approximation in Section D, subspace approximation in Section E, tensor low-rank approximation in Section F, clustering and data selection pipeline in Section G and lower bound in Section H.

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

## A  PRELIMINARIES

### A.1  NOTATION

For any $n \in \mathbb{N}$, let $[n]$ denote the set $\{1, 2, \ldots, n\}$. We use $\widetilde{O}(\cdot)$ to hide polylogarithmic factors in $n$, $d$, $1/\epsilon$, $1/\delta$, and other problem-related parameters, such as $k$ and $p$. For two numbers $a$ and $b$, we use $a \vee b$ as a shorthand for $\max\{a, b\}$. We use $a = (1 \pm \epsilon)b$ to denote $a \in [(1 - \epsilon)b, (1 + \epsilon)b]$.

For a matrix $A$, we use $\|A\|_2$ or simply $\|A\|$ to denote the spectral norm of $A$. For a tensor $A$, let $\|A\|$ and $\|A\|_2$ (used interchangeably) denote the spectral norm of tensor $A$,

$$\|A\| = \sup_{x, y, z \neq 0} \frac{|A(x, y, z)|}{\|x\| \cdot \|y\| \cdot \|z\|}.$$

Let $A \in \mathbb{R}^{n \times d}$ and $k \leq \min\{n, d\}$. We will use $A_k$ or $[A]_k$ to denote its best rank-$k$ approximation. Let $\|A\|_F$ denote the Frobenius norm of a matrix/tensor $A$, i.e., $\|A\|_F$ is the square root of the sum of squares of all entries of $A$. For $1 \leq p < 2$, we use $\|A\|_p$ to denote the entry-wise $\ell_p$-norm of a matrix/tensor $A$, i.e., $\|A\|_p$ is the $p$-th root of the sum of $p$-th powers of the absolute values of the entries of $A$. $\|A\|_1$ will be an important special case of $\|A\|_p$, representing the sum of the absolute values of all entries.

Let $\mathrm{nnz}(A)$ denote the number of nonzero entries of $A$. Let $\det(A)$ denote the determinant of a square matrix $A$. Let $A^\top$ denote the transpose of $A$. Let $A^\dagger$ denote the Moore-Penrose pseudoinverse of $A$. Let $A^{-1}$ denote the inverse of a full-rank square matrix.

For a 3rd order tensor $A \in \mathbb{R}^{n \times n \times n}$, we use $A_{i,j,l}$ to denote its $(i, j, l)$-th element, $A_{i,*,l}$ to denote its $i$-th row, and $A_{i,j,*}$ to denote its $j$-th column.

A tensor $A$ is symmetric if and only if for any $i, j, k$, $A_{i,j,k} = A_{i,k,j} = A_{j,i,k} = A_{j,k,i} = A_{k,i,j}$.

For a tensor $A \in \mathbb{R}^{n_1 \times n_2 \times n_3}$, we use $\top$ to denote rotation (3-dimensional transpose) so that $A^\top \in \mathbb{R}^{n_3 \times n_1 \times n_2}$. For a tensor $A \in \mathbb{R}^{n_1 \times n_2 \times n_3}$ and matrix $B \in \mathbb{R}^{n_3 \times k}$, we define the tensor-matrix dot product to be $A \cdot B \in \mathbb{R}^{n_1 \times n_2 \times k}$.

### A.2  SENSITIVITY AND CORESET

Throughout this paper, we will extensively work with sensitivity and coreset. Let $X$ be some universe of elements. Our main focus is the cost function: $\mathrm{cost} : \mathbb{R}^d \times X \to \mathbb{R}_{\geq 0}$, which measures the cost of an element $x \in X$ with respect to the first argument. We then define the notion of strong and weak coresets.

**Definition A.1** ((Strong) Coreset). *Let $B \subseteq A$ and $\epsilon \in (0, 1)$. We say that $B$ is an $\epsilon$-strong coreset or $\epsilon$-coreset of $A$ if there exists a nonnegative weight vector $w \in \mathbb{R}_{\geq 0}^{|B|}$ such that for all $x \in X$,*

$$\sum_{b \in B} w_b \cdot \mathrm{cost}(b, x) = (1 \pm \epsilon) \cdot \mathrm{cost}(A, x).$$

Strong coreset preserves the cost over all possible $x \in X$, but sometimes we only need the optimal cost preserved. We also introduce the notion of weak coreset.

**Definition A.2** (Weak Coreset). *Let $B \subseteq A$ and $\epsilon \in (0, 1)$. We say that $B$ is an $\epsilon$-weak coreset if there exists a nonnegative weight vector $w \in \mathbb{R}_{\geq 0}^{|B|}$ such that*

$$\min_{x \in X} \sum_{b \in B} w_b \cdot \mathrm{cost}(b, x) = (1 \pm \epsilon) \cdot \mathrm{OPT},$$

*where $\mathrm{OPT} = \min_{x \in X} \mathrm{cost}(A, x)$.*

**Remark A.3.** *Oftentimes, given a weighted subset $(B, w)$, we will use $\mathrm{cost}(B, x)$ as an abbreviation for $\sum_{b \in B} w_b \cdot \mathrm{cost}(b, x)$, as our analysis and algorithms on the subset of points work in both unweighted and weighted settings. Hence, when the weight is clear from context, we will abuse notation and use $\mathrm{cost}(b, x)$ to denote $w_b \cdot \mathrm{cost}(b, x)$.*

**Definition A.4** (Sensitivity and Generalized Sensitivity). *Let $A = \{a_1, \ldots, a_n\} \subset \mathbb{R}^d$. We define the* sensitivity *of $a_i$ as*

$$s_i(A, A) = \max_{x \in X} \frac{\text{cost}(a_i, x)}{\text{cost}(A, x)}.$$

*Let $B \subset \mathbb{R}^d$. We define the* sensitivity *of $a_i$ with respect to $B$ as*

$$s_i(A, B) = \max_{x \in X, \text{cost}(B, x) \neq 0} \frac{\text{cost}(a_i, x)}{\text{cost}(B, x)}.$$

### A.3 LEVERAGE SCORE, RIDGE LEVERAGE SCORE, AND LEWIS WEIGHTS

**Definition A.5** (Statistical Dimension). *For real value $\lambda \geq 0$ and a rank-d matrix $A \in \mathbb{R}^{n \times d}$ with singular values $\sigma_i(A)$, the quantity $s_d^\lambda(A) := \sum_{i=1}^d \frac{1}{\sqrt{1 + \lambda/\sigma_i^2(A)}}$ is the statistical dimension of the ridge regression problem with regularizing weight $\lambda$.*

**Definition A.6** (Leverage Score). *Given matrix $A \in \mathbb{R}^{n \times d}$, leverage score can be defined as follows:*

$$\tau_i(A) := a_i^\top (A^\top A)^\dagger a_i,$$

*where $a_i^\top$ is the $i$-th row of $A$ for all $i \in [n]$.*

**Definition A.7** (Ridge Leverage Score). *Given matrix $A \in \mathbb{R}^{n \times d}$, we denote the $i$-th ridge leverage score, for $i \in [n]$, as follows:*

$$\overline{\tau}_i(A, \lambda_{A_k}) := a_i^\top (A^\top A + \lambda_{A_k} I)^{-1} a_i,$$

*where $\lambda_{A_k} = \|A - A_k\|_F^2 / k$ and $I \in \mathbb{R}^{d \times d}$ is the identity matrix. When the rank $k$ is clear from context, we may abbreviate $\overline{\tau}_i(A)$ as $\overline{\tau}_i(A, \lambda_{A_k})$.*

**Definition A.8** (Generalized Ridge Leverage Score). *Let $A \in \mathbb{R}^{n \times d}$, $C \in \mathbb{R}^{n \times d'}$, and $i \in [d]$. We define the $i$-th generalized ridge leverage score of $A \in \mathbb{R}^{n \times d}$ with respect to $C \in \mathbb{R}^{n \times d'}$ as follows:*

$$\overline{\tau}_i(A, C, \lambda_{C_k}) = \begin{cases} a_i^\top (CC^\top + \lambda_{C_k} I_n)^\dagger a_i, & \text{if } a_i \in \text{span}(CC^\top + \lambda I_n); \\ \infty, & \text{otherwise.} \end{cases}$$

*When the rank $k$ is clear from context, we may use $\overline{\tau}_i(A, C)$ as shorthand for $\overline{\tau}_i(A, C, \lambda_{C_k})$.*

**Definition A.9** (Lewis Weights). *Let $p \in (0, \infty)$ and $A \in \mathbb{R}^{n \times d}$. We define the $\ell_p$ Lewis weights of $A$, denoted by $w_A$, as*

$$w_{A,i} = \tau_i(W_A^{1/2 - 1/p} A),$$

*or equivalently,*

$$w_{A,i}^{2/p} = a_i^\top (A^\top W_A^{1 - 2/p} A)^{-1} a_i.$$

### A.4 MATRIX APPROXIMATIONS

**Definition A.10** (Subspace Embedding in Sarlos (2006)). *Let $\epsilon, \delta \in (0, 1)$ and $n > d$. Given a matrix $U \in \mathbb{R}^{n \times d}$ which is orthonormal (i.e., $U^\top U = I_d$), we say $S \in \mathbb{R}^{m \times n}$ is an $\mathsf{SE}(\epsilon, \delta, n, d)$ subspace embedding for fixed $U$ if*

$$(1 - \epsilon)\|Ux\|_2^2 \leq \|SUx\|_2^2 \leq (1 + \epsilon)\|Ux\|_2^2$$

*holds with probability $1 - \delta$. This is equivalent to*

$$\|U^\top S^\top S U - U^\top U\| \leq \epsilon.$$

**Definition A.11** (Weak $\epsilon$-Affine Embedding, Theorem 39 in Clarkson & Woodruff (2013)). *Let matrices $A \in \mathbb{R}^{n \times r}$ and $B \in \mathbb{R}^{n \times d}$. Given matrix $S \in \mathbb{R}^{t \times n}$, we say $S$ is weak $\epsilon$-affine embedding if the following conditions hold: let $\widehat{X} = \arg\min_X \|AX - B\|_F^2$ and $\widehat{B} = A\widehat{X} - B$ and then*

$$\|S(AX - B)\|_F^2 - \|S\widehat{B}\|_F^2 = (1 \pm \epsilon)\|AX - B\|_F^2 - \|\widehat{B}\|_F^2$$

## A.5    PROPERTIES OF LEVERAGE SCORE

Sampling according to leverage score distribution yields a weak affine embedding property; additionally, solving the subsampled problem results in an optimal solution whose cost is close to the original optimal cost.

**Lemma A.12** (Theorem 42 in Clarkson & Woodruff (2013)). *Let matrix $A \in \mathbb{R}^{n \times r}$ with rank at most $k$, and let $B \in \mathbb{R}^{n \times d}$. If $S \in \mathbb{R}^{n \times n}$ is a sampling and rescaling diagonal matrix according to the leverage scores of $A$, let $m = O(\epsilon^{-2} k \log k)$ denote the number of nonzero entries on the diagonal of $S$. Then for all $X \in \mathbb{R}^{r \times d}$, we have:*

- *$S$ is a weak $\epsilon$-affine embedding (see Definition A.11);*

- *equivalently, if $\widehat{X} = \arg\min_X \|AX - B\|_F^2$, $\widehat{B} = A\widehat{X} - B$, and $C := \|S\widehat{B}\|_F^2 - \|\widehat{B}\|_F^2$, then*

$$(1 - \epsilon) \cdot \|AX - B\|_F^2 + C \leq \|S(AX - B)\|_F^2 \leq (1 + \epsilon) \cdot \|AX - B\|_F^2 + C.$$

**Lemma A.13** (Leverage Score Preserves Optimal Cost, Lemma C.31 of Song et al. (2019)). *Let $A \in \mathbb{R}^{n \times r}$ be a matrix with rank at most $k$, and let $B \in \mathbb{R}^{n \times d}$. If we sample $O(k \log k + k/\epsilon)$ rows of $A$ and $B$ proportional to the leverage scores of $A$ to obtain a sampling matrix $S$, then with probability at least $1 - \delta$,*

$$\|AY_* - B\|_F^2 \leq (1 + \epsilon) \cdot \min_Y \|AY - B\|_F^2,$$

*where $Y_* = \arg\min_Y \|SAY - SB\|_F^2$.*

## A.6    QUANTUM PRIMITIVES

Our core quantum primitive is a sampling algorithm based on Grover search.

**Lemma A.14** (Claim 3 in Apers & De Wolf (2022)). *Let $n$ be a positive integer and let $p_i$ for all $i \in [n]$ with $p_i \in [0, 1]$. There is a quantum algorithm that generates a list of indices with $i$ sampled with probability $p_i$ independently, in time $\widetilde{O}(\sqrt{n \sum_{i=1}^n p_i}) \cdot \mathcal{T}$, where $\mathcal{T}$ is the time to compute $p_i$.*

We note that this runtime bound could also be achieved via quantum rejection sampling (Ozols et al., 2012). Let $P = \sum_{i=1}^n p_i$, then $p_i/P$ for all $i \in [n]$ induces a probability distribution, which we denote by $\sigma$. Recall that the rejection sampling aims to generate one sample from the target distribution $\sigma$ (where $\sigma_i = p_i/P$) using a uniform proposal distribution $\pi$ (where $\pi_i = 1/n$), the query complexity is $\widetilde{O}(\max_{i \in [n]} \sqrt{\sigma_i/\pi_i}) \cdot \mathcal{T}$, as each $p_i \leq 1$, the ratio can be upper bounded by $\max_i (p_i/P)/(1/n) \leq n/P$, thus, the complexity to generate *one sample* is $\widetilde{O}(\sqrt{n/P}) \cdot \mathcal{T}$. As $\sum_{i=1}^n p_i = P$, if we choose each index $i$ with probability $p_i$ independently, then the expected size is $P$, hence the total expected complexity is $\widetilde{O}(P\sqrt{n/P}) \cdot \mathcal{T} = \widetilde{O}(\sqrt{nP}) \cdot \mathcal{T}$, as desired.

Throughout the paper, we will use the notation $\text{QLS}(A, s, \delta)$ to denote the procedure of sampling $s$ rows or columns from $A$ according to the leverage score distribution of $A$, with probability at least $1 - \delta$ that these leverage scores are constant factor approximations to the exact leverage scores. The time for this procedure is $\sqrt{ns} \cdot \mathcal{T}$, where $\mathcal{T}$ is the time to compute a single score. Similarly, we use $\text{QGRLS}(A, C, \epsilon, \delta, \lambda)$ to denote the procedure of sampling according to the generalized ridge leverage score distribution $\overline{\tau}_i(A, C, \lambda)$.

# B    A QUANTUM RECURSIVE SAMPLING FRAMEWORK FOR CORESET

Throughout this section, let us consider $A = \{a_1, \ldots, a_n\} \subset \mathbb{R}^d$ to be a set of $n$ points in $\mathbb{R}^d$, and $X$ to be a set. Let $\text{cost} : \mathbb{R}^d \times X \to \mathbb{R}_{\geq 0}$ be a cost function, and for $x \in X$, let $\text{cost}(A, x) = \sum_{i=1}^n \text{cost}(a_i, x)$. The main objective of this section is to develop a framework for sampling a weighted subset of $A$ that approximates the cost of $A$. To do so, we prove that if the weights satisfying certain assumptions, then a generic recursive sampling framework could construct a coreset from these weights. The assumptions are listed in the following.

**Assumption B.1.** *Given two finite subsets $A, B \subseteq \mathbb{R}^d$, let $w(A, B) \in \mathbb{R}^{|A|}$ be a nonnegative weight vector where $w_i(A, B)$ is the weight of $a_i$ with respect to $B$. We assume $w$ satisfies the following conditions:*

- *Consistent total weights: for any subset $S \subseteq [n]$, $\sum_{i \in S} w_i(A_S, A_S) \leq \text{sum}(w)$ where $\text{sum}(w)$ is a finite upper bound on the total weights;*

- *Uniform sampling bound: let $A'$ be a uniform subset of $A$ with size $m$ and let $w'(A, A') \in \mathbb{R}^n$ be defined as $w'_i(A, A') := \begin{cases} w_i(A, A'), & \text{if } a_i \in A', \\ w_i(A, A' \cup \{a_i\}), & \text{otherwise;} \end{cases}$, then $w'_i(A, A') \geq w_i(A, A)$ for all $i \in [n]$;*

- *Importance sampling bound: let $u_i$ be an overestimate of $w_i(A, A)$ and suppose we sample according to $q_i = \min\{1, g(\epsilon, n, d) \cdot u_i\}$, yielding a weighted subset $B \subseteq A$ of size $g(\epsilon, n, d) \cdot \|u\|_1$, then with high probability, $B$ is an $\epsilon$-coreset of $A$ with size $g(\epsilon, n, d) \cdot \|u\|_1$;*

- *Coreset preserves weights: let $B$ be an $\epsilon$-coreset of $A$, then $w_i(C, B) = (1 \pm \epsilon) \cdot w_i(C, A)$ for any fixed $C$ and for all $i \in [n]$.*

---

**Algorithm 1** Quantum recursive sampling for coreset.

---

1: **procedure** QRecurseSample($A \in \mathbb{R}^{n \times d}, \epsilon$)
2:     **if** $n \leq g(\epsilon, n, d) \cdot \text{sum}(w)$ **then**
3:         **return** $(A, I_n)$
4:     **end if**
5:     $c \leftarrow 1000$
6:     $A' \subset_{1/2} A$
7:     $s \leftarrow g(\epsilon, n, d) \cdot \text{sum}(w)$
8:     $(C', D') \leftarrow$ QRecurseSample($A', \epsilon$)
9:     Implement a classical oracle for $w'_i(A, C')$
10:                                              $\triangleright\ p_i = \min\{1, c \cdot g(\epsilon, n, d) \cdot w'_i(A, C')\}$
11:     $D \leftarrow$ QSample($p$)
12:     $C \leftarrow D^\top A$
13:     **return** $(C, D)$
14: **end procedure**

---

Before presenting our most general result, we first show that if $B$ is a coreset of $A$, then $B \cup \{p\}$ is also a coreset of $A \cup \{p\}$ for any $p \notin A$.

**Lemma B.2.** *Let $B$ be an $\epsilon$-coreset of $A$ and let $p \notin A$, then $B \cup \{p\}$ is an $\epsilon$-coreset of $A \cup \{p\}$.*

*Proof.* Since $B$ is an $\epsilon$-coreset of $A$, we know that for any $x \in X$, $\text{cost}(B, x) = (1 \pm \epsilon) \cdot \text{cost}(A, x)$ with high probability. Conditioning on this event, we note that

$$\begin{aligned} \text{cost}(B \cup \{p\}, x) &= \text{cost}(B, x) + \text{cost}(\{p\}, x) \\ &\leq (1 + \epsilon) \cdot \text{cost}(A, x) + \text{cost}(\{p\}, x) \\ &\leq (1 + \epsilon) \cdot \text{cost}(A \cup \{p\}, x), \end{aligned}$$

the lower bound can be established similarly. $\qquad\square$

**Theorem B.3.** *Let $A \in \mathbb{R}^{n \times d}$. Then, there exists a quantum algorithm that constructs an $\epsilon$-coreset $C$ of expected size $s := O(g(\epsilon, n, d) \cdot \text{sum}(w))$ Moreover, if a classical oracle for $w_i(X, Y)$ can be implemented with*

- *Preprocessing in time $\mathcal{T}_{\text{prep}}(|Y|, d)$;*

- *Query time $\mathcal{T}_{\text{query}}(|Y|, d)$ for computing $w_i(X, Y)$ for any $i \in X$,*

*the algorithm runs in time*

$$\mathcal{T}_{\text{prep}}(s', d) + \widetilde{O}(\sqrt{ns} \cdot \mathcal{T}_{\text{query}}(s', d)),$$

*where $s' = O(g(0.01, n, d) \cdot \text{sum}(w))$ and uses $\widetilde{O}(\sqrt{ns})$ queries to the rows or columns of $A$.*

---

**Algorithm 2** Quantum iterative sampling for coreset.

---

1: **procedure** QITERATESAMPLE($A \in \mathbb{R}^{n \times d}, \epsilon$)
2:     $c \leftarrow 1000$
3:     $s \leftarrow 4c \cdot g(\epsilon, n, d) \cdot \text{sum}(w)$
4:     $T \leftarrow \log(n/s)$
5:     $\epsilon_0 \leftarrow 0.01$
6:     $s' \leftarrow 4c \cdot g(\epsilon', n, d) \cdot \text{sum}(w)$
7:     $A_0 \subset_{1/2} A_1 \subset_{1/2} \ldots \subset_{1/2} A_{T-1} \subset_{1/2} A_T = A$
8:     $C_0 \leftarrow A_0$
9:     **for** $t = 1 \rightarrow T - 1$ **do**
10:         Implement a classical oracle for $w'_i(A_t, C_{t-1})$ for all $a_i \in A_t$
11:                                        $\triangleright p_i = \min\{1, c \cdot g(\epsilon', n, d) \cdot w'_i(A_t, C_{t-1})\}$
12:         $D_t \leftarrow$ QSAMPLE($p$)                    $\triangleright \|D_t\|_0 = s'$
13:         $C_t \leftarrow D_t^\top A_t$                    $\triangleright C_t \in \mathbb{R}^{s' \times d}$
14:     **end for**
15:     Implement a classical oracle for $w'_i(A_T, C_{T-1})$ for all $a_i \in A_T$
16:                                        $\triangleright p_i = \min\{1, c \cdot g(\epsilon, n, d) \cdot w'_i(A_T, C_{T-1} \cup \{a_i\})\}$
17:     $D_T \leftarrow$ QSAMPLE($p$)                    $\triangleright \|D_T\|_0 = s$
18:     $C_T \leftarrow D_T^\top A_T$                    $\triangleright C_T \in \mathbb{R}^{s \times d}$
19:     **return** $(C_T, D_T)$
20: **end procedure**

---

*Proof.* As the algorithm is recursive, we will prove by induction on $n$. For the base case, we have $n \leq g(\epsilon, n, d) \cdot \text{sum}(w)$; in this case, we could simply take the coreset as $A$, as it satisfies the size guarantee with exact approximation.

For the inductive step, we assume it holds for $n/2$ as our algorithm uniformly samples half of the points. This means that $C'$ is an $\epsilon$-coreset for $A'$ and by the importance sampling bound of Assumption B.1, we have $w_i(A, C') = (1 \pm \epsilon) \cdot w_i(A, A')$ with high probability. Now, we consider two cases: if $a_i \in A'$, then $w'_i(A, A') = w_i(A, A')$ and $w'_i(A, C') = w_i(A, C') = (1 \pm \epsilon)w_i(A, A') = (1 \pm \epsilon)w'_i(A, A')$. If $a_i \notin A'$, then $w'_i(A, A') = w'_i(A, A' \cup \{a_i\}) = (1 \pm \epsilon)w_i(A, C' \cup \{a_i\}) = (1 \pm \epsilon)w'_i(A, C')$ by Lemma B.2.

Next, we prove that for any uniform subset $S \subseteq [n]$ with $|S| = m$, we have

$$\mathbb{E}[\|w'(A, SA)\|_1] \leq \frac{n}{m} \cdot \|w(A, A)\|_1.$$

Let us denote $S^{(i)}$ as the diagonal indicator matrix for $S \cup \{i\}$. Then, note

$$\sum_{i=1}^n w'_i(A, SA) = \sum_{i \in S} w_i(A, SA) + \sum_{i \notin S} w_i(A, S^{(i)}A)$$

$$= \|w(SA, SA)\|_1 + \sum_{i \notin S} w_i(A, S^{(i)}A)$$

$$\leq \|w(A, A)\|_1 + \sum_{i \notin S} w_i(A, S^{(i)}A),$$

to bound the second term, note that it is generated via the following random process: first selecting $S$, then selecting a random $i \notin S$ and returning $w_i(A, S^{(i)}A)$. Since there are $n - m$ points not in $SA$, the expected value of this process is $\frac{1}{n-m} \mathbb{E}[\sum_{i \notin S} w_i(A, S^{(i)}A)]$. The key observation is that this process is equivalent to another process: pick a random subset $S' \subset [n]$ of size $m + 1$, then randomly pick a point $a_i \in S'A$ and return $w_i(A, S'A)$. In expectation, this is equal to the average weight over $S'A$. Since $S'A$ contains $m + 1$ points and by the consistent total weights assumption, the average weight is at most $\frac{\|w(A,A)\|_1}{m+1}$. Therefore,

$$\mathbb{E}[\sum_{i \notin S} w_i(A, S^{(i)}A)] \leq (n - m) \cdot \frac{\|w(A, A)\|_1}{m+1},$$

combining these results, we obtain the following expectation bound:

$$\mathbb{E}[\sum_{i=1}^{n} w'_i(A, SA)] \leq \|w(A,A)\|_1 + (n-m) \cdot \frac{\|w(A,A)\|_1}{m+1}$$

$$\leq \frac{n+1}{m+1} \cdot \|w(A,A)\|_1$$

$$\leq \frac{n}{m} \cdot \|w(A,A)\|_1.$$

Hence, since $A'$ is a uniform subset of $A$ with size $n/2$, we know that $\mathbb{E}[\|w'(A,A')\|_1] \leq 2\|w(A,A)\|_1$ and $w'_i(A,A') \geq w_i(A,A)$ by the uniform sampling bound. Therefore, if we simply scale $w'_i(A,C')$ by a factor of $\frac{1}{1-\epsilon}$, then we have

$$w'_i(A,C') \geq w'_i(A,A') \geq w_i(A,A)$$

and moreover

$$\mathbb{E}[\|w'(A,C')\|_1] \leq (1+3\epsilon)\mathbb{E}[\|w'(A,A')\|_1]$$

$$\leq 4\|w(A,A)\|_1$$

$$\leq 4 \cdot \mathrm{sum}(w)$$

consequently, if we sample according to $c \cdot g(\epsilon, n, d) \cdot w'_i(A, C')$, then the expected size of $C$ is at most $c' \cdot g(\epsilon, n, d) \cdot \mathrm{sum}(w)$ for $c' = 4c$, and the coreset guarantee follows naturally from the importance sampling bound of Assumption B.1.

Regarding the running time, we analyze an iterative version of the algorithm that achieves the same effect, illustrated in Algorithm 2. One key difference is that for the intermediate steps, we use a constant approximation to improve the runtime. We divide the proof into steps.

- To uniformly subsample half of the points, we follow the approach of Apers & Gribling (2024), which takes $\widetilde{O}(\log(n/s))$ time;

- For each iteration, we first prepare a classical oracle for $w'_i(A_t, C_{t-1})$ in $\mathcal{T}_{\mathrm{prep}}(s', d)$ time;

- Next, we need to sample according to $p_i = \min\{1, g(\epsilon', n, d) \cdot w'_i(A_t, C_{t-1})\}$ with

$$\mathbb{E}[\sum_{i \in A_t} p_i] \leq c \cdot g(\epsilon', n, d) \cdot \mathbb{E}[\sum_{i \in A_t} w'_i(A_t, C_{t-1})]$$

$$\leq 2c \cdot g(\epsilon', n, d) \cdot \mathbb{E}[\sum_{i \in A_t} w'_i(A_t, A_{t-1})]$$

$$\leq 4c \cdot g(\epsilon', n, d) \cdot \sum_{i \in A_t} w_i(A_t, A_t)$$

$$\leq 4c \cdot g(\epsilon', n, d) \cdot \mathrm{sum}(w)$$

$$= s',$$

using Lemma A.14, this step can be implemented in time

$$\widetilde{O}(\sqrt{ns'}) \cdot \mathcal{T}_{\mathrm{query}}(s', d);$$

- Forming $C_t$ requires selecting and weighting $s'$ points, which can be done in $O(s')$ time;

- Finally, we do a resampling with $\epsilon$ to form the final coreset, which takes

$$\mathcal{T}_{\mathrm{prep}}(s, d) + \widetilde{O}(\sqrt{ns} \cdot \mathcal{T}_{\mathrm{query}}(s, d))$$

time, as desired. $\qquad\square$

While Theorem B.3 provides both approximation guarantees in terms of coreset and runtime, in applications it is more convenient to craft an algorithm that takes the size of the coreset as a parameter.

---

**Algorithm 3** Quantum iterative sampling for coreset: fixed size.

---

1: **procedure** QITERATEFIXEDSIZE($A \in \mathbb{R}^{n \times d}, s, s'$)
2:     $c \leftarrow 1000 \cdot s/\|w(A, A)\|_1$
3:     $c' \leftarrow 1000 \cdot s'/\|w(A, A)\|_1$
4:     $T \leftarrow \log(n/s)$
5:     $A_0 \subset_{1/2} A_1 \subset_{1/2} \ldots \subset_{1/2} A_{T-1} \subset_{1/2} A_T = A$
6:     $C_0 \leftarrow A_0$
7:     **for** $t = 1 \rightarrow T - 1$ **do**
8:        Implement a classical oracle for $w_i'(A_t, C_{t-1})$ for all $a_i \in A_t$
9:                                     $\triangleright \ p_i = \min\{1, c' \cdot w_i'(A_t, C_{t-1})\}$
10:       $D_t \leftarrow$ QSAMPLE($p$)                         $\triangleright \ \|D_t\|_0 = s'$
11:       $C_t \leftarrow D_t^\top A_t$                           $\triangleright \ C_t \in \mathbb{R}^{s' \times d}$
12:     **end for**
13:     Implement a classical oracle for $w_i'(A_T, C_{T-1})$ for all $a_i \in A_T$
14:                                   $\triangleright \ p_i = \min\{1, c \cdot w_i'(A_T, C_{T-1})\}$
15:       $D_T \leftarrow$ QSAMPLE($p$)                        $\triangleright \ \|D_T\|_0 = s$
16:       $C_T \leftarrow D_T^\top A_T$                         $\triangleright \ C_T \in \mathbb{R}^{s \times d}$
17:     **return** $(C_T, D_T)$
18: **end procedure**

---

**Corollary B.4.** *Let $A \in \mathbb{R}^{n \times d}$ and $s, s' \in [n]$. Then, there exists a quantum algorithm that constructs a coreset $C$ of $A$ with expected size $s$. Assuming access to a classical oracle for $w_i(X, Y)$ with:*

- *Preprocessing time $\mathcal{T}_{\mathrm{prep}}(|Y|, d)$;*

- *Query time $\mathcal{T}_{\mathrm{query}}(|Y|, d)$ for computing $w_i(X, Y)$ for any $i \in X$,*

*the algorithm runs in time*

$$\mathcal{T}_{\mathrm{prep}}(s', d) + \widetilde{O}(\sqrt{ns} \cdot \mathcal{T}_{\mathrm{query}}(s', d)),$$

*and uses $\widetilde{O}(\sqrt{ns})$ queries to the rows or columns of $A$.*

Our main contribution is to prove that *sensitivity sampling* satisfies Assumption B.1.

**Definition B.5.** *Let $A = \{a_1, \ldots, a_n\} \subset \mathbb{R}^d$ and let $\mathrm{cost} : \mathbb{R}^d \times X \to \mathbb{R}_{\geq 0}$ be a cost function. We define the sensitivity of $a_i$ with respect to $B$, denoted by $s_i(A, B)$, as*

$$s_i(A, B) = \max_{x \in X, \mathrm{cost}(B,x) \neq 0} \frac{\mathrm{cost}(a_i, x)}{\mathrm{cost}(B, x)}.$$

We also need to define the dimension of a system $(A, w, X, \mathrm{cost})$:

**Definition B.6.** *Given a point set $A = \{a_1, \ldots, a_n\} \subset \mathbb{R}^d$, nonnegative weights $w \in \mathbb{R}_{\geq 0}^{|A|}$, a space $X$ and a cost function $\mathrm{cost} : \mathbb{R}^d \times X \to \mathbb{R}_{\geq 0}$, let $r \in [0, \infty)$ and let $X(A_S)$ be a function that inputs a subset of points from $A$ and outputs a set of $x \in X$ associated with $A_S$. We define*

$$\mathrm{range}(x, r) = \{a_i \in A : w_i \cdot \mathrm{cost}(a_i, x) \leq r\}.$$

*The dimension of $(A, w, X, \mathrm{cost})$ is the smallest integer $\dim$ such that for any subset $S \subseteq [n]$ we have*

$$|\{\mathrm{range}(x, r) : x \in X(A_S), r \in [0, \infty)]\}| \leq |S|^{\dim}.$$

**Lemma B.7** (Theorem 2.7 of Braverman et al. (2022)). *Let $\dim$ be the dimension of $(A, w, X, \mathrm{cost})$ (Def. B.6), let $q_i := \min\{1, w_i \cdot s_i(A, A)\}$ and $t \geq \sum_{i=1}^n q_i$, let $\epsilon, \delta \in (0, 1)$. Let $c \geq 1$ be a sufficiently large constant, and let $S$ be a sample generated by sampling according to $q_i$. Then, with probability at least $1 - \delta$, we can generate a subset $S \subseteq [n]$ such that for all $x \in X(S)$,*

$$|\mathrm{cost}(A, x) - \sum_{i \in S} \frac{w_i}{|S| \cdot q_i} \mathrm{cost}(a_i, x)| \leq \epsilon \cdot \mathrm{cost}(A, x),$$

*moreover, the size of $S$ is*

$$\frac{ct}{\epsilon^2} \cdot (\dim \cdot \log t + \log(1/\delta)).$$

**Theorem B.8.** *Let $A = \{a_1, \ldots, a_n\} \subset \mathbb{R}^d$ and let $\mathrm{cost} : \mathbb{R}^d \times X \to \mathbb{R}_{\geq 0}$. Moreover, suppose the total sensitivity has a finite upper bound, i.e., there exists some $\mathrm{sum}(s) < \infty$ such that for any finite subset $C \subset \mathbb{R}^d$, $\sum_{i \in C} s_i(C, C) \leq \mathrm{sum}(s)$. Then, the sensitivity of $A$ with respect to $B$, $s(A, B)$ (Def. B.5) satisfies Assumption B.1.*

*Proof.* We need to prove $s(A, B)$ satisfies the four items in Assumption B.1.

- Consistent total weights: by assumption, we have that for any $S \subseteq [n]$, $\sum_{i \in S} s_i(A_S, A_S) \leq \mathrm{sum}(s)$ with $\mathrm{sum}(s)$ being finite.

- Uniform sampling bound: we analyze by cases. For the first case, where $a_i \in A'$, we have $w_i'(A, A') = s_i(A, A')$. Let $x_1, x_2$ be the two points that realize $s_i(A, A')$ and $s_i(A, A)$, respectively. Suppose $\mathrm{cost}(A', x_2) \neq 0$, then

$$
\frac{\mathrm{cost}(a_i, x_1)}{\mathrm{cost}(A', x_1)} \geq \frac{\mathrm{cost}(a_i, x_2)}{\mathrm{cost}(A', x_2)}
$$

$$
\geq \frac{\mathrm{cost}(a_i, x_2)}{\mathrm{cost}(A', x_2) + \mathrm{cost}(A \setminus A', x_2)}
$$

$$
= \frac{\mathrm{cost}(a_i, x_2)}{\mathrm{cost}(A, x_2)}
$$

  where we use the fact that $\mathrm{cost}$ is nonnegative, therefore increasing the denominator will only decrease the fraction. On the other hand, if $\mathrm{cost}(A', x_2) = 0$, then it must be that $\mathrm{cost}(a_i, x_2) = 0$ due to the nonnegativity of cost. Hence, $s_i(A, A) = 0$, and consequently $s_i(A, A') = 0$ as otherwise we could pick $x_1$ for $s_i(A, A)$.

  For the next case, where $a_i \notin A'$, we have $w_i'(A, A') = s_i(A, A' \cup \{a_i\})$. Again, let $x_1, x_2$ be the two points that realize $s_i(A, A' \cup \{a_i\})$ and $s_i(A, A)$. The argument is similar: suppose $\mathrm{cost}(A', x_2) \neq 0$, then

$$
\frac{\mathrm{cost}(a_i, x_1)}{\mathrm{cost}(A', x_1) + \mathrm{cost}(a_i, x_1)} \geq \frac{\mathrm{cost}(a_i, x_2)}{\mathrm{cost}(A', x_2) + \mathrm{cost}(a_i, x_2)}
$$

$$
\geq \frac{\mathrm{cost}(a_i, x_2)}{\mathrm{cost}(A', x_2) + \mathrm{cost}(a_i, x_2) + \mathrm{cost}(A \setminus (A' \cup \{a_i\}), x_2)}
$$

$$
= \frac{\mathrm{cost}(a_i, x_2)}{\mathrm{cost}(A, x_2)}.
$$

  If $\mathrm{cost}(A', x_2) = 0$, then we claim that in fact, $x_1 = x_2$. This follows because

$$
\frac{\mathrm{cost}(a_i, x_2)}{\mathrm{cost}(A', x_2) + \mathrm{cost}(a_i, x_2)} = \frac{\mathrm{cost}(a_i, x_2)}{\mathrm{cost}(a_i, x_2)}
$$

$$
= 1,
$$

  by the definition of sensitivity, the max sensitivity is 1, therefore in this case it must be $x_1 = x_2$ and $s_i(A, A) \leq 1 = s_i(A, A' \cup \{a_i\})$.

- Importance sampling bound: this can be achieved via Lemma B.7, by taking $m = O(\epsilon^{-2}\|u\|_1 \cdot (\dim \cdot \log(\|u\|_1) + \log(1/\delta)))$ samples.

- Coreset preserves weights: let $B$ be an $\epsilon$-coreset of $A$. Then, we know that for any $x \in X$, $\mathrm{cost}(B, x) = (1 \pm \epsilon) \cdot \mathrm{cost}(A, x)$. Now, let $C \subset \mathbb{R}^d$ be any fixed set of points, and let $x_1, x_2 \in X$ be the points that achieve $s_i(C, A)$ and $s_i(C, B)$. We have:

$$
w_i(C, B) = s_i(C, B)
$$

$$
= \frac{\mathrm{cost}(c_i, x_2)}{\mathrm{cost}(B, x_2)}
$$

$$
\leq (1 + \epsilon) \cdot \frac{\mathrm{cost}(c_i, x_2)}{\mathrm{cost}(A, x_2)}
$$

$$\leq (1 + \epsilon) \cdot \frac{\text{cost}(c_i, x_1)}{\text{cost}(A, x_1)}$$

$$= (1 + \epsilon) \cdot s_i(C, A),$$

we could similarly establish that $s_i(C, B) \geq (1-\epsilon) \cdot s_i(C, A)$. This proves the assertion. $\square$

In what follows, we demonstrate how to concretely implement sensitivity sampling for various cost functions, such as $\ell_p$ sensitivity and $k$-subspace sensitivity.

### B.1 $\ell_2$ SENSITIVITY AND LEVERAGE SCORE

Let $X = \mathbb{R}^d$ and $\text{cost}(a_i, x) = (a_i^\top x)^2$. In this case, the $\ell_2$ sensitivity defined as

$$s_i(A, B) = \max_{x \in \mathbb{R}^d, Bx \neq 0} \frac{(a_i^\top x)^2}{\|Bx\|_2^2}$$

is, in fact, the leverage score $\tau_i(A)$. The leverage score has many favorable structures: for example, to obtain an $\epsilon$-coreset, it is sufficient to sample $O(\epsilon^{-2} d \log d)$ points, and one could sample with $w_i(A, A')$ instead of $w_i'(A, A')$.

---

**Algorithm 4** Classical oracle for leverage score.

1: **data structure** LEVERAGESCORE
2: **members**
3:     $A \in \mathbb{R}^{n \times d}$
4:     $C \in \mathbb{R}^{s \times d}$
5:     $M \in \mathbb{R}^{O(\log n) \times d}$
6: **end members**
7:
8: **procedure** PREPROCESS($A \in \mathbb{R}^{n \times d}, C \in \mathbb{R}^{s \times d}$)
9:     $c \leftarrow 1000$
10:     Compute the thin SVD of $C$: $C = U\Sigma V^\top$          $\triangleright V \in \mathbb{R}^{d \times s}$
11:     Let $G \in \mathbb{R}^{c \log n \times s}$ be a random Gaussian matrix
12:     $M \leftarrow (GV)(\Sigma^\dagger V^\top)$          $\triangleright M \in \mathbb{R}^{c \log n \times d}$
13: **end procedure**
14:
15: **procedure** QUERY($i \in [n]$)
16:     **return** $\|Ma_i\|_2^2$
17: **end procedure**
18: **end data structure**

---

**Lemma B.9.** *Let $A = \{a_1, \ldots, a_n\} \subset \mathbb{R}^d$ and define $\text{cost} : \mathbb{R}^d \times \mathbb{R}^d \to \mathbb{R}_{\geq 0}$ by $\text{cost}(a_i, x) = (a_i^\top x)^2$, and let $w(A, B)$ be defined as*

$$w_i(A, B) = \begin{cases} a_i^\top (B^\top B)^\dagger a_i, & \text{if } a_i \in \text{span}(B^\top B); \\ \infty, & \text{otherwise.} \end{cases}$$

*Then, the weights $w$ satisfy Assumption B.1. Moreover, there exists a randomized algorithm (Algorithm 4) that implements PREPROCESS and QUERY procedures, with*

- *$\mathcal{T}_{\text{prep}}(s, d) = \widetilde{O}(sd^{\omega-1})$;*

- *$\mathcal{T}_{\text{query}}(s, d) = \widetilde{O}(d)$.*

*Proof.* While leverage score is $\ell_2$ sensitivity and we could directly use Theorem B.8, we include a proof that utilizes the structure of leverage score for completeness.

- Consistent total weights: first note that

$$\sum_{i=1}^n w_i(A, A) = \sum_{i=1}^n a_i^\top (A^\top A)^\dagger a_i$$

$$= \text{tr}[(A^\top A)^\dagger A^\top A]$$
$$= \text{rank}(A)$$
$$\leq d$$

hence we have $\text{sum}(w) = d$. Let $S \subset [n]$ with $|S| \geq d$, then

$$\sum_{i \in S} w_i(A_S, A_S) = \sum_{i \in S} a_i^\top (A_S^\top A_S)^\dagger a_i$$
$$= \text{tr}[(A_S^\top A_S)^\dagger (A_S^\top A_S)]$$
$$= \text{rank}(A_S)$$
$$\leq d.$$

- Uniform sampling bound: the proof closely follows that of (Cohen et al., 2015b, Theorem 1), and we analyze by cases. Let $S$ be an indicator matrix with $A' = SA$ and let $S^{(i)}A$ be the indicator matrix for $S \cup \{i\}$. We will show that $w_i'(A, A') = w_i(A, S^{(i)}A)$ via case analysis. If $a_i \in A'$, then $w_i'(A, A') = w_i(A, A')$ and $S = S^{(i)}$, consequently $w_i(A, SA) = w_i(A, S^{(i)}A)$. If $a_i \notin A'$, then $w_i'(A, A') = w_i(A, A' \cup \{i\}) = w_i(A, S^{(i)}A)$. This completes the proof. To show the overestimate, observe that $S^{(i)}$ is an indicator matrix for the sample and thus $S^{(i)} \preceq I_n$, we can then conclude

$$A^\top (S^{(i)})^2 A \preceq A^\top A$$

and

$$w_i'(A, A') = a_i^\top (A^\top (S^{(i)})^2 A)^\dagger a_i$$
$$\geq a_i^\top (A^\top A)^\dagger a_i$$
$$= w_i(A, A).$$

- Importance sampling bound: this is standard as $w_i(A, A)$ is the leverage score of matrix $A$. The proof follows from a standard matrix Chernoff bound (by sampling $O(\epsilon^{-2} d \log d)$ points) and we refer readers to (Cohen et al., 2015b, Lemma 4).

- Coreset preserves weights: because $B$ is an $\epsilon$-coreset of $A$, we know that for any $x \in \mathbb{R}^d$, $\|Bx\|_2^2 = (1 \pm \epsilon) \cdot \|Ax\|_2^2$. Expanding yields

$$(1 - \epsilon) \cdot x^\top A^\top Ax \preceq x^\top B^\top Bx \preceq (1 + \epsilon) \cdot x^\top A^\top Ax,$$

this implies that $B^\top B$ is a spectral approximation to $A^\top A$ and $\ker(A) = \ker(B)$, and the same holds for $(B^\top B)^\dagger$ with respect to $(A^\top A)^\dagger$. Let $C \subset \mathbb{R}^d$ be any fixed subset of $\mathbb{R}^d$. We conclude the proof by spectral approximation:

$$(1 - \epsilon) \cdot c_i^\top (A^\top A)^\dagger c_i \preceq c_i^\top (B^\top B)^\dagger c_i \preceq (1 + \epsilon) \cdot c_i^\top (A^\top A)^\dagger c_i.$$

Now, we turn to the runtime analysis of Algorithm 4. Let $C = U\Sigma V^\top$. Then we have $(C^\top C)^\dagger = (V\Sigma^2 V^\top)^\dagger = V(\Sigma^\dagger)^2 V^\top$. By definition,

$$w_i(A, C) = a_i^\top (C^\top C)^\dagger a_i$$
$$= a_i^\top V(\Sigma^\dagger)^2 V^\top a_i$$
$$= \|\Sigma^\dagger V^\top a_i\|_2^2,$$

using a standard Johnson-Lindenstrauss trick (Spielman & Srivastava, 2011), it is sufficient to apply a JL matrix $G$ and prepare the matrix $G\Sigma^\dagger V^\top$. Then, with high probability, all $w_i(A, C)$ can be approximated within a factor of $1 \pm \epsilon$. By a simple scaling argument, this gives an overestimate. Thus, Algorithm 4 gives the correct overestimates of leverage scores. It remains to analyze the runtime.

- Computing the thin SVD of $C$ takes $O(sd^{\omega-1})$ time;

- Computing $GV$ takes $\widetilde{O}(sd)$ time and then we multiply $GV$ with $\Sigma^\dagger V^\top$ which takes $\widetilde{O}(sd)$ time as well;

- For query, note that $M \in \mathbb{R}^{\log n \times d}$, and thus computing $\|Ma_i\|_2^2$ takes $\widetilde{O}(d)$ time.

This completes the proof of the assertion. □

**Remark B.10.** *If we faithfully execute the framework of Theorem B.8, then we would need to compute the $w_i(A, C \cup \{a_i\})$ instead of $w_i(A, C)$. Instead, we only need to sample with $w_i(A, C)$. This is a key feature for leverage score and related notions, which we summarize below.*

**Lemma B.11** (Theorem 4 of Cohen et al. (2015b)). *Let $A = \{a_1, \ldots, a_n\} \subset \mathbb{R}^d$. Suppose we sample points uniformly and independently with probability $\frac{m}{n}$ to obtain $SA$. Let $q_i = \min\{1, w_i(A, SA)\}$ and sample points of $A$ according to $q$ and reweight them accordingly to obtain a weighted subset $B$. Then, with high probability, $B$ is an $\epsilon$-coreset of $A$ with size $O(\frac{nd \log d}{\epsilon^2 m})$.*

Setting $m = n/2$, Lemma B.11 itself is sufficient to prove Theorem B.3, without resorting to use $w_i(A, C \cup \{a_i\})$. Our following theorem recovers the main result of Apers & Gribling (2024).

**Theorem B.12.** *Let $A = \{a_1, \ldots, a_n\} \subset \mathbb{R}^d$ and $\epsilon, \delta \in (0, 1)$. Then, there exists a quantum algorithm that with probability $1 - \delta$, constructs an $\epsilon$-coreset $B$ of $A$ of size $O(\epsilon^{-2}d\log(d/\delta))$, in time $\widetilde{O}(\epsilon^{-1}n^{0.5}d^{1.5} + d^\omega)$ and $\widetilde{O}(\epsilon^{-1}n^{0.5}d^{0.5})$ queries to points in $A$.*

*Furthermore, if we wish to construct a fixed-size sample of size $s$, we use $\mathrm{QLS}(A, s, \delta)$ to denote this algorithm. This variant succeeds with probability at least $1 - \delta$ to sample $s$ weighted points, in time $\widetilde{O}(n^{0.5}s^{0.5}d + sd^{\omega-1})$ and $\widetilde{O}(n^{0.5}s^{0.5})$ queries to points in $A$.*

*Proof.* The proof follows by observing that we could replace condition 2 and 3 of Assumption B.1 by Lemma B.11, and then we could integrate Lemma B.9 into Theorem B.3. To achieve the desired $\epsilon$-coreset guarantee, we choose

- $s = O(\epsilon^{-2}d\log(d/\delta))$;

- $s' = O(d\log(d/\delta))$.

Plugging in the choices of $s, s'$ into Lemma B.9 and Theorem B.3 yields an overall runtime of

$$\widetilde{O}(\epsilon^{-1}n^{0.5}d^{1.5} + d^\omega). \qquad \square$$

### B.2 $\ell_p$ SENSITIVITY AND LEWIS WEIGHTS

To preserve $\ell_p$ subspace, one could sample according to $\ell_p$ sensitivity: let us define $\mathrm{cost}(a_i, x) = |a_i^\top x|^p$ for $p \in (0, \infty)$, then the $\ell_p$ sensitivity is

$$s_i(A, B) = \max_{x \in \mathbb{R}^d, Bx \neq 0} \frac{|a_i^\top x|^p}{\|Bx\|_p^p},$$

and a computationally efficient proxy for $\ell_p$ sensitivity is $\ell_p$ Lewis weights, defined as the unique nonnegative weight vector $w_A \in \mathbb{R}^n$ with

$$w_{A,i}^{2/p} = a_i^\top (A^\top W_A^{1-2/p} A)^{-1} a_i,$$

where $W_A \in \mathbb{R}^{n \times n}$ is the diagonal matrix of $w_A$. Naturally, we define our weights as

$$w_i(A, B) = (a_i^\top (B^\top W_B^{1-2/p} B)^{-1} a_i)^{p/2}.$$

To implement recursive sampling according to Lewis weights, we need a stronger notion of approximation for $\epsilon$-coreset, as beyond sensitivity, the weights might not be preserved by an $\epsilon$-coreset. We explicitly define the notion of an $\epsilon$-*approximator*, a weighted subset of points that preserves the weights. Note that an $\epsilon$-approximator is not necessarily an $\epsilon$-coreset.

**Definition B.13.** *Let $A = \{a_1, \ldots, a_n\} \subset \mathbb{R}^d$. We say a weighted subset $B$ of $A$ is an $\epsilon$-approximator if for any fixed $C$ and for any $i \in [n]$,*

$$w_i(C, B) = (1 \pm \epsilon) \cdot w_i(C, A).$$

For $\ell_p$ Lewis weights, it might be simpler to talk about approximating the $2/p$-th power of $w$; in this case, we have that $B^\top W_B^{1-2/p} B$ is a $1 \pm \epsilon$ spectral approximation to $A^\top W_A^{1-2/p} A$. Cohen & Peng (2015) proves an analogous result to Lemma B.11 for $\ell_p$ Lewis weights, and in turn this satisfies the requirements of Theorem B.3.

**Lemma B.14** (Lemma 6.2 of Cohen & Peng (2015)). *Let $A = \{a_1, \ldots, a_n\} \subset \mathbb{R}^d$. Suppose we sample points uniformly and independently with probability $\frac{1}{2}$ to obtain $SA$. Let $q_i = \min\{1, w_i(A, SA)\}$ and sample points of $A$ according to $q$ and reweight them accordingly to obtain a weighted subset $B$. Then, with high probability, $B$ is an $\epsilon$-approximator of $A$ with expected size $O_p(\epsilon^{-(2 \vee p)} d^{(p/2 \vee 1)+1} \log d)$.*

We also need the following result due to Fazel et al. (2022) that approximates $\ell_p$ Lewis weights in nearly exact leverage score time:

**Lemma B.15** (Theorem 2 of Fazel et al. (2022)). *Let $A = \{a_1, \ldots, a_n\} \subset \mathbb{R}^d$, $p \in (0, \infty)$ and $\epsilon \in (0, 1)$. Then, there exists a deterministic algorithm that outputs a vector $\widetilde{w}_A \in \mathbb{R}^n$ such that for any $i \in [n]$, $\widetilde{w}_{A,i} = (1 \pm \epsilon) \cdot w_{A,i}$. Moreover, the runtime of this algorithm is*

$$O_p(nd^{\omega-1} \log(np/\epsilon)).$$

---

**Algorithm 5** Classical oracle for Lewis weights.

1: **data structure** LewisWeights
2: **members**
3:     $A \in \mathbb{R}^{n \times d}$
4:     $C \in \mathbb{R}^{s \times d}$
5:     $M \in \mathbb{R}^{O(p^2 \log n) \times d}$
6: **end members**
7:
8: **procedure** PREPROCESS($A \in \mathbb{R}^{n \times d}, C \in \mathbb{R}^{s \times d}$)
9:     $c \leftarrow 1000p^2$
10:     Generate $\widetilde{W}_C$ via Lemma B.15 on $C$            $\triangleright \widetilde{W}_C \in \mathbb{R}^{s \times s}$
11:     Compute the thin SVD of $\widetilde{W}_C^{1/2-1/p} C$: $\widetilde{W}_C^{1/2-1/p} C = U\Sigma V^\top$
12:     Let $G \in \mathbb{R}^{c \log n \times s}$ be a random Gaussian matrix
13:     $M \leftarrow (GV)(\Sigma^{-1} V^\top)$                 $\triangleright M \in \mathbb{R}^{c \log n \times d}$
14: **end procedure**
15:
16: **procedure** QUERY($i \in [n]$)
17:     **return** $\|Ma_i\|_2^p$
18: **end procedure**
19: **end data structure**

---

Note the striking similarity between Algorithm 5 and Algorithm 4, as Lewis weights are leverage scores of $A$ after appropriate reweighting.

**Lemma B.16.** *Let $A = \{a_1, \ldots, a_n\} \subset \mathbb{R}^d$, $p \in (0, \infty)$, $\epsilon, \delta \in (0, 1)$, and define $w(A, B)$ as*

$$w_i(A, B)^{p/2} = a_i^\top (B^\top W_B^{1-2/p} B)^{-1} a_i,$$

*Then, the weights $w$ satisfy the requirements for Theorem B.3 for an $\epsilon$-approximator. Moreover, there exists a randomized algorithm (Algorithm 5) that implements PREPROCESS and QUERY procedures with*

- $\mathcal{T}_{\mathrm{prep}}(s, d) = \widetilde{O}_p(sd^{\omega-1})$;

- $\mathcal{T}_{\mathrm{query}}(s, d) = \widetilde{O}_p(d)$.

*Proof.* The proof is similar to Theorem B.12 by observing that we can replace condition 2 and 3 of Assumption B.1 by Lemma B.14. It remains to verify condition 1 and 4.

- Consistent total weights: observe that we can alternatively define $w_A$ as $w_{A,i} = \tau_i(W_A^{1/2-1/p}A)$, i.e., it is the leverage score of $W_A^{1/2-1/p}A$. Since the sum of the leverage scores is at most the rank, we have $\text{sum}(w) = d$.

- Coreset preserves weights: instead of a coreset, we will be generating a sequence of $\epsilon$-approximators, so we will instead prove that: if $B$ is an $\epsilon$-approximator of $A$, then for any fixed $C$ and any $i \in [n]$, $w_i(C,B) = (1 \pm \epsilon) \cdot w_i(C,A)$. By definition, if $B$ is an $\epsilon$-approximator of $A$, then we have the following:

$$c_i^\top (B^\top W_B B)^{-1} c_i = (1 \pm \epsilon) \cdot c_i (A^\top W_A A)^{-1} c_i,$$

  however, this shows that $w_i(C,B)^{2/p} = (1 \pm \epsilon) \cdot w_i(C,A)^{2/p}$. By raising both sides to the appropriate power, we see that $w_i(C,B) = (1 \pm \epsilon)^{p/2} \cdot w_i(C,A) = (1 \pm p\epsilon/2) \cdot w_i(C,A)$. What we have just shown is that an $\epsilon$-approximator preserves weights up to a factor of $1 \pm O(p\epsilon)$, so to achieve $(1 \pm \epsilon)$ factor approximation for the weights, we would need an $\epsilon/p$-approximator.

Since in the end, we will do a final resampling using the approximated Lewis weights, we will stick to obtaining an $\epsilon/p$-approximator.

To analyze Algorithm 5, we first consider a variant where the Johnson-Lindenstrauss transformation is not applied. We compute $\widetilde{W}_C$ using Lemma B.15 which is a $1 \pm \epsilon$ spectral approximation to $W_C$, then we know that $\widetilde{W}_C^{1-2/p}$ is a $(1 \pm \epsilon)^{|1-2/p|}$ spectral approximation to $W_C^{1-2/p}$, and this approximation guarantee propagates to $C^\top \widetilde{W}_C^{1-2/p} C$ and $(C^\top \widetilde{W}_C^{1-2/p} C)^{-1}$. So far, we have established that for any $a_i$, $a_i^\top (C^\top \widetilde{W}_C^{1-2/p} C)^{-1} a_i = (1 \pm \epsilon)^{|1-2/p|} \cdot a_i^\top (C^\top W_C^{1-2/p} C)^{-1} a_i$, and our final output is the $(p/2)$-th power of the quantity, hence the approximate weight is a $(1 \pm \epsilon)^{|p/2-1|}$ approximation to the true weight. Hence, for $p \in (0,2)$, our output is already a $1 \pm O(\epsilon)$ approximation to the true quantity, and for $p \geq 2$, we are outputting a $1 \pm p\epsilon/2$ approximation. To obtain the correct $1 \pm \epsilon$ approximation, we need to set the correct approximation factor.

When applying the Johnson-Lindenstrauss transformation, we are effectively approximating $a_i^\top (C^\top \widetilde{W}_C^{1-2/p} C)^{-1} a_i$, and by the same argument, we could use a $1 \pm O(1/p)$ approximation for Johnson-Lindenstrauss, resulting in a dimension of $O(p^2 \log n)$. Let us analyze the runtime.

- PREPROCESS: to compute $\widetilde{W}_C$, we need to set the $\epsilon$ parameter in Lemma B.15 to $O(1/p)$, and it runs in time $\widetilde{O}_p(sd^{\omega-1})$. Computing the SVD takes $O(sd^{\omega-1})$ time, and applying the random Gaussian matrix takes $\widetilde{O}_p(sd)$ time.

- QUERY: note that $M \in \mathbb{R}^{\widetilde{O}_p(1) \times d}$, hence answering one query takes time $\widetilde{O}_p(d)$.

This completes the proof. $\qquad\square$

Lemma B.16 gives an approach to compute an $\epsilon$-approximator for $A$, but our ultimate goal is to compute an $\epsilon$-coreset for $A$, which has a different objective. The following result states that sampling according to the appropriate scaling of overestimates of Lewis weights indeed yields an $\epsilon$-coreset:

**Lemma B.17** (Theorem 1.3 of Woodruff & Yasuda (2023)). *Let $A = \{a_1, \ldots, a_n\} \subset \mathbb{R}^d$, $\epsilon, \delta \in (0,1)$ and $p \in (0, \infty)$ and let $u \in \mathbb{R}^n$ be an overestimate of $w_A$ with $\|u\|_1 \leq O(d)$. Consider the sampling scheme where each point is sampled with probability $q_i = \min\{1, \alpha \cdot u_i\}$ where*

- $\alpha = \epsilon^{-2}(\log^3 d + \log(1/\delta))$ *for $p \in (0,1)$;*

- $\alpha = \epsilon^{-2} \log(n/\delta)$ *for $p = 1$;*

- $\alpha = \epsilon^{-2}(\log^2 d \log n + \log(1/\delta))$ *for $p \in (1,2)$;*

- $\alpha = \epsilon^{-2} d^{p/2-1}(\log^2 d \log n + \log(1/\delta))$ *for $p \geq 2$.*

*Set $s_i = q_i^{-1/p}$. Then, with probability at least $1 - \delta$, $SA$ is an $\epsilon$-coreset of $A$, and the number of samples is at most $\alpha \cdot \|u\|_1$.*

We are now ready to state our main result for constructing an $\epsilon$-coreset. Due to Lemma B.17, we only need an overestimate for Lewis weights, so we will obtain an $O(1/p)$-approximator first, and then use it to generate approximate Lewis weights.

**Theorem B.18.** *Let $A = \{a_1, \ldots, a_n\} \subset \mathbb{R}^d$, $\epsilon, \delta \in (0, 1)$ and $p \in (0, \infty)$. There exists a quantum algorithm that with probability at least $1 - \delta$, constructs an $\epsilon$-coreset of $A$ with size $\alpha \cdot d$, for $\alpha$ given in Lemma B.17. The runtimes for generating the coreset are*

- $\widetilde{O}_p(d^{\omega+1} + \epsilon^{-2}d^3) + \widetilde{O}_p(n^{0.5}d^{1.5}(\epsilon^{-3} + d^{0.5}))$ *for* $p \in (0, 2)$;

- $\widetilde{O}_p(d^{p/2}(d^\omega + \epsilon^{-2}d^2)) + \widetilde{O}_p(n^{0.5}d^{p/4+1}(\epsilon^{-3} + d^{0.5}))$ *for* $p \geq 2$.

*The number of queries to the points in A for generating the coresets are*

- $\widetilde{O}_p(\epsilon^{-1}n^{0.5}d^{0.5})$ *for* $p \in (0, 2)$;

- $\widetilde{O}_p(\epsilon^{-1}n^{0.5}d^{p/4})$ *for* $p \geq 2$.

*Proof.* Our strategy will be to first construct an $O(1/p)$-approximator of $A$, which in turn gives an $O(1)$-approximation to $w_A$, then we will sample according to these approximations, in conjunction with Lemma A.14.

- Stage 1: constructing an $O(1/p)$-approximator of $A$. The proof follows by combining Lemma B.16 and Theorem B.3, with $s = s'$ and
$$s = \widetilde{O}_p(d^{(p/2\vee 1)+1}),$$
and the time to generate such an $O(1/p)$-approximator is
$$\widetilde{O}_p(d^{(p/2\vee 1)+\omega}) + \widetilde{O}_p(n^{0.5}d^{(p/2\vee 1)/2+1.5}).$$
We let $\widetilde{B}$ denote the resulting approximator. Note that the size of $\widetilde{B}$ is $\widetilde{O}_p(d^{(p/2\vee 1)+1})$.

- Stage 2: constructing an $\epsilon$-coreset of $A$. Observe that $\widetilde{B}$ gives an $O(1)$-approximation to $w_A$, as for any $a_i$,
$$(a_i^\top(\widetilde{B}^\top W_{\widetilde{B}}^{1-2/p}\widetilde{B})^{-1}a_i)^{p/2} = O(1) \cdot (a_i^\top(A^\top W_A^{1-2/p}A)^{-1}a_i)^{p/2}$$
$$= O(1) \cdot w_{A,i},$$
and after appropriately rescaling this yields the desired oversampling vector $u$. Note that
$$\|u\|_1 = \sum_{i=1}^n (a_i^\top(\widetilde{B}^\top W_{\widetilde{B}}^{1-2/p}\widetilde{B})^{-1}a_i)^{p/2}$$
$$= O(1) \cdot (a_i^\top(A^\top W_A^{1-2/p}A)^{-1}a_i)^{p/2}$$
$$= O(d),$$
and we could invoke Lemma B.17 to generate the desired $\epsilon$-coreset. We could still use Algorithm 5 as our oracle to supply the sampling probability, except we need to use a Johnson-Lindenstrauss transformation that gives $(1 \pm \epsilon/p)$-approximation. Given $\widetilde{B}$, generating $\widetilde{W}_{\widetilde{B}}$ takes $\widetilde{O}_p(d^{(p/2\vee 1)+\omega})$ time, and the next time-consuming operation is applying the JL. Note that the JL has dimension $\widetilde{O}_p(\epsilon^{-2})$, hence applying the JL takes time $\widetilde{O}_p(\epsilon^{-2}d^{(p/2\vee 1)+2})$. For query, note that the dimension of $M$ is $\widetilde{O}_p(\epsilon^{-2}) \times d$, and each query can be implemented in $\widetilde{O}_p(\epsilon^{-2}d)$ time. All in all, we obtain the following (simplified) runtime for generating the $\epsilon$-coreset:

  - For $p \in (0, 2)$, it takes time $\widetilde{O}_p(d^{\omega+1} + \epsilon^{-2}d^3 + \epsilon^{-3}n^{0.5}d^{1.5})$;
  - For $p \geq 2$, it takes time $\widetilde{O}_p(d^{p/2+\omega} + \epsilon^{-2}d^{p/2+2} + \epsilon^{-3}n^{0.5}d^{p/4+1})$.

This concludes the proof. $\qquad\square$

## B.3 $k$-SUBSPACE SENSITIVITY AND RIDGE LEVERAGE SCORE

Let $\mathcal{F}_k$ be the set of all $k$-dimensional subspaces in $\mathbb{R}^d$. We can define the cost with respect to a subspace by identifying $X = \mathcal{F}_k$ and $\mathrm{cost} : \mathbb{R}^d \to \mathcal{F}_k \to \mathbb{R}_{\geq 0}$ as

$$\mathrm{cost}(a_i, x) = \|a_i^\top (I - P_x)\|_2^2,$$

where $P_x$ is the orthogonal projection onto $x$. Then, the $k$-subspace sensitivity is

$$s_i(A, B) = \max_{x \in X} \frac{\|a_i^\top (I - P_x)\|_2^2}{\|B(I - P_x)\|_F^2}.$$

Similar to $\ell_p$ sensitivity, $k$-subspace sensitivity can be overestimated by *ridge leverage score*, defined as

$$\overline{\tau}_i(A) = a_i^\top (A^\top A + \lambda_{A_k} I)^{-1} a_i$$

where $\lambda_{A_k} = \|A - A_k\|_F^2 / k$. We then define the weights similar to leverage score:

$$w_i(A, B) = \begin{cases} a_i^\top (B^\top B + \lambda_{B_k} I)^\dagger a_i, & \text{if } a_i \in \mathrm{span}(B^\top B + \lambda_{B_k} I), \\ \infty, & \text{otherwise.} \end{cases}$$

We will explicitly define the notion of $\epsilon$-approximator:

**Definition B.19.** *Let $A = \{a_1, \ldots, a_n\} \subset \mathbb{R}^d$, $\epsilon \in (0, 1)$ and $1 \leq k \leq d$. We say $B$ is an $\epsilon$-approximator of $A$ if*

- *$B$ is an $\epsilon$-coreset of $A$;*

- *The following additive-multiplicative spectral approximation guarantee holds:*

$$(1 - \epsilon) B^\top B - \epsilon \lambda_{A_k} I \preceq A^\top A \preceq (1 + \epsilon) B^\top B + \epsilon \lambda_{A_k} I.$$

The following two results due to Cohen et al. (2017) illustrate that an $\epsilon$-approximator indeed preserves all weights, and uniform sampling gives sufficiently good approximation.

**Lemma B.20** (Lemma 12 of Cohen et al. (2017)[2]). *Let $A = \{a_1, \ldots, a_n\} \subset \mathbb{R}^d$ and $\epsilon \in (0, 1)$. If $B$ is an $\epsilon$-approximator of $A$, then for any fixed $C$ and for all $i \in [n]$, $w_i(C, B) = (1 \pm \epsilon) \cdot w_i(C, A)$.*

**Lemma B.21** (Theorem 14 of Cohen et al. (2017)). *Let $A = \{a_1, \ldots, a_n\} \subset \mathbb{R}^d$. Suppose we sample points uniformly and independently with probability $\frac{1}{2}$ to obtain $SA$. Let $q_i = \min\{1, w_i(A, SA)\}$ and sample points of $A$ according to $q$ and reweight them accordingly to obtain a weighted subset $B$. Then, with high probability, $B$ is an $\epsilon$-approximator of $A$ with expected size $O(\epsilon^{-2} k \log k)$.*

**Lemma B.22.** *Let $A = \{a_1, \ldots, a_n\} \subset \mathbb{R}^d$, $k \leq d$, $\epsilon, \delta \in (0, 1)$, and define $w(A, B)$ as*

$$w_i(A, B) = \begin{cases} a_i^\top (B^\top B + \lambda_{B_k} I)^\dagger a_i, & \text{if } a_i \in \mathrm{span}(B^\top B + \lambda_{B_k} I), \\ \infty, & \text{otherwise.} \end{cases}$$

*Then, the weights $w$ satisfy the requirements for Theorem B.3 for an $\epsilon$-approximator. Moreover, there exists a randomized algorithm (Algorithm 6) that implements PREPROCESS and QUERY procedures with*

- $\mathcal{T}_{\mathrm{prep}}(s, d) = \widetilde{O}(d s^{\omega - 1})$;

- $\mathcal{T}_{\mathrm{query}}(s, d) = \widetilde{O}(d)$.

*Proof.* We only need to derive a total weights upper bound, as other conditions of Assumption B.1 are already satisfied by Lemma B.21. Let $A = U \Sigma V^\top$ be the SVD of $A$. Then,

$$\sum_{i=1}^n w_i(A, A) = \sum_{i=1}^n a_i^\top (A^\top A + \lambda_{A_k} I)^\dagger a_i$$

---

[2]Note that while the original Lemma in Cohen et al. (2017) states the result in terms of ridge leverage score, their proof essentially shows that $B^\top B + \lambda_{B_k} I$ is a $1 \pm \epsilon$ spectral approximation of $A^\top A + \lambda_{A_k} I$, which gives the desired approximator guarantee.

---

**Algorithm 6** Classical oracle for ridge leverage score.

---

1: **data structure** RIDGELEVERAGESCORE
2: **members**
3: $A \in \mathbb{R}^{n \times d}$
4: $C \in \mathbb{R}^{s \times d}$
5: $M \in \mathbb{R}^{O(\log n) \times d}$
6: **end members**
7:
8: **procedure** PREPROCESS($A \in \mathbb{R}^{n \times d}, C \in \mathbb{R}^{s \times d}$)
9: $c \leftarrow 1000$
10: Compute the thin SVD of $C$: $C = U\Sigma V^\top$      $\triangleright V \in \mathbb{R}^{d \times s}$
11: $\lambda \leftarrow \sum_{i=k+1}^{d} \sigma_i$
12: Let $G \in \mathbb{R}^{c \log n \times s}$ be a random Gaussian matrix
13: $M \leftarrow (GV)(\Sigma^\dagger V^\top + \frac{1}{\sqrt{\lambda}} V^\top)$      $\triangleright M \in \mathbb{R}^{c \log n \times d}$
14: **end procedure**
15:
16: **procedure** QUERY($i \in [n]$)
17: **return** $\|Ma_i\|_2^2$
18: **end procedure**
19: **end data structure**

---

$$= \text{tr}[A^\top A (A^\top A + \lambda_{A_k} I)^\dagger]$$

$$= \text{tr}[V\Sigma^2 V^\top (V(\Sigma^2)^\dagger V^\top + \frac{1}{\lambda_{A_k}} VV^\top)]$$

$$= \sum_{i=1}^{n} \frac{\sigma_i^2}{\sigma_i^2 + \frac{\|A - A_k\|_F^2}{k}}$$

$$\leq k + \sum_{i=k+1}^{n} \frac{\sigma_i^2}{\sigma_i^2 + \frac{\|A - A_k\|_F^2}{k}}$$

$$\leq k + \sum_{i=k+1}^{n} \frac{\sigma_i^2}{\frac{\|A - A_k\|_F^2}{k}}$$

$$= k + k \cdot \frac{\|A - A_k\|_F^2}{\|A - A_k\|_F^2}$$

$$\leq 2k$$

where for the fifth step, we upper bound $\frac{\sigma_i^2}{\sigma_i^2 + \frac{\|A - A_k\|_F^2}{k}}$ by 1 for $i \leq k$. The runtime analysis is identical to that of Lemma B.9. $\qquad\square$

One of the key features of the $\epsilon$-approximator for $k$-subspace approximation is that it is also an $\epsilon$-coreset.

**Theorem B.23.** *Let $A = \{a_1, \ldots, a_n\} \subset \mathbb{R}^d$, $\epsilon, \delta \in (0, 1)$ and $k \in [d]$. There exists a quantum algorithm QRLS($A, k, \epsilon, \delta$) that with probability at least $1 - \delta$, constructs an $\epsilon$-coreset of $A$ with size $O(\epsilon^{-2} k \log(k/\delta))$, in time $\widetilde{O}(\epsilon^{-1} n^{0.5} dk^{0.5} + dk^{\omega - 1})$ and $\widetilde{O}(\epsilon^{-1} n^{0.5} k^{0.5})$ queries to the points in $A$.*

*Proof.* The proof is almost identical to the proof of Theorem B.12, except that the sizes $s$ and $s'$ are

- $s = \widetilde{O}(\epsilon^{-2} k)$;

- $s' = \widetilde{O}(k)$.

Plugging these choices into Lemma B.22 and Theorem B.3 gives a runtime of

$$\widetilde{O}(\epsilon^{-1}n^{0.5}dk^{0.5} + dk^{\omega-1}). \qquad \square$$

# C  QUANTUM COLUMN SUBSET SELECTION AND LOW-RANK APPROXIMATION

In this section, we present the first application of the generic sampling framework developed in Section B. In particular, when the cost is the $k$-subspace cost defined as $\mathrm{cost}(A, x) = \|A(I - P_x)\|_F^2$ where $x \in \mathcal{F}_k$, then an $\epsilon$-coreset of $A$ can be used to compute a Frobenius norm low-rank approximation. Let $A \in \mathbb{R}^{d \times n}$, we let the set of points be $\{a_1, \ldots, a_n\} \subset \mathbb{R}^d$, and the goal is to compute a weighted subset of columns of $A$.

**Lemma C.1** (Lemma 3 of Cohen et al. (2015a)). *Let $A = \{a_1, \ldots, a_n\} \subset \mathbb{R}^d$, $\epsilon \in (0, 1)$, $k \in [\min\{n, d\}]$ and let $B \subset A$ be an $\epsilon$-coreset of $A$ with respect to the $k$-subspace cost. Then, the projection onto the top-$k$ left singular vectors of $B$, denoted by $P_{B_k}$, satisfies*

$$\|A - P_{B_k}P_{B_k}^\top A\|_F^2 = (1 \pm \epsilon)\|A - A_k\|_F^2.$$

Cohen et al. (2017) is the first to observe that ridge leverage score is in fact an overestimate of $k$-subspace sensitivity, and sampling according to ridge leverage score gives in fact a stronger $\epsilon$-approximator (see Section B.3), which is an $\epsilon$-coreset. We hence summarize the result below.

**Corollary C.2.** *Let $A \in \mathbb{R}^{d \times n}$, $\epsilon \in (0, 1)$, $k \le \min\{n, d\}$ be a positive integer. There exists a quantum algorithm* QLOWRANKCMM$(A, k, \epsilon, \delta)$ *that constructs an $\epsilon$-coreset $C$ of $A$ for the $k$-subspace cost with probability at least $1 - \delta$. The size of the coreset is at most $O(k \log(k/\delta)/\epsilon^2)$, the runtime is $\widetilde{O}(n^{0.5}dk^{0.5}\epsilon^{-1} + dk^{\omega-1})$, and the number of queries to the columns of $A$ is $\widetilde{O}(\epsilon^{-1}n^{0.5}k^{0.5})$.*

We note that in addition, $C$ is a column subset selection of $A$:

**Definition C.3** (Rank-$k$ Column Subset Selection). *For $n' < n$, a subset of $A$'s columns $C \in \mathbb{R}^{d \times n'}$ is a $(1 + \epsilon)$ factor column subset selection if there exists a rank-$k$ matrix $X \in \mathbb{R}^{n' \times n}$ with*

$$\|A - CX\|_F^2 \le (1 + \epsilon)\|A - A_k\|_F^2.$$

We utilize this fact to further derive an algorithm for outputting a low-rank approximation of $A$, which could subsequently be generalized to tensor. We state a tool for solving a bilinear multiple response regression.

**Lemma C.4** (Generalized Low-Rank Approximation (Friedland & Torokhti, 2007)). *Let $A \in \mathbb{R}^{d \times n}$, $B \in \mathbb{R}^{d \times k'}$ and $C \in \mathbb{R}^{k' \times d}$, let $k \le \min\{n, d\}$ be a positive integer. The following bilinear regression problem*

$$\min_{X:\mathrm{rank}(X) \le k} \|A - BXC\|_F^2$$

*is minimized by $X_* = B^\dagger[P_B A P_C]_k C^\dagger$ where $P_B, P_C$ are the projection matrices onto $B, C$ respectively.*

**Theorem C.5.** *Let $A \in \mathbb{R}^{d \times n}$ and $\epsilon \in (0, 0.1)$, and let $k \le \min\{d, n\}$ be a positive integer. Then, there exists a quantum algorithm (Algorithm 7) that outputs a pair of rank-$k$ matrices $M \in \mathbb{R}^{d \times k}, N \in \mathbb{R}^{n \times k}$ such that*

$$\|A - MN^\top\|_F^2 \le (1 + \epsilon) \cdot \|A - A_k\|_F^2$$

*holds with probability at least $0.99$. Moreover, Algorithm 7 runs in time*

$$\widetilde{O}(\epsilon^{-1}n^{0.5}dk^{0.5} + dk^{\omega-1} + \epsilon^{-1.5}d^{0.5}k^{1.5} + \epsilon^{-2}n^{0.5}k^{1.5} + \epsilon^{-3}nk),$$

*and uses $\widetilde{O}(\epsilon^{-1}n^{0.5}k^{0.5})$ queries to the columns and $\widetilde{O}(\epsilon^{-1.5}d^{0.5}k^{0.5})$ to the rows of $A$.*

*Proof.* We start by proving the correctness of Algorithm 7. First note that $C$ is a column subset selection (Definition C.3), meaning that there exists a rank-$k$ matrix $X$ with

$$\|A - CX\|_F^2 \le (1 + \epsilon)\|A - A_k\|_F^2,$$

---

**Algorithm 7** Quantum low-rank approximation.

---

1: **procedure** QLOWRANK($A \in \mathbb{R}^{d \times n}, k, \epsilon$)
2:      $k_1 \leftarrow O(\epsilon^{-2} k \log k)$
3:      $k_2 \leftarrow O(k_1 \log k_1 + \epsilon^{-1} k_1)$
4:      $k_3 \leftarrow O(k_2 \log k_2 + \epsilon^{-1} k_2)$
5:      $C \leftarrow$ QLOWRANKCMM($A, k, \epsilon, 0.001$)           $\triangleright$ $C \in \mathbb{R}^{d \times k_1}$, Corollary C.2.
6:      $S \leftarrow$ QLS($C, k_2, 0.001$)                $\triangleright$ $S \in \mathbb{R}^{k_2 \times d}$, Theorem B.12.
7:      $T_1 \leftarrow$ QLS($C, k_2, 0.001$)             $\triangleright$ $T_1 \in \mathbb{R}^{k_2 \times d}$, Theorem B.12.
8:      $T_2 \leftarrow$ QLS($SA, k_3, 0.001$)         $\triangleright$ $T_2 \in \mathbb{R}^{n \times k_3}$, Theorem B.12.
9:      $X, Y \leftarrow \min_{X \in \mathbb{R}^{k_1 \times k}, Y \in \mathbb{R}^{k \times k_2}} \|T_1 C X Y S A T_2 - T_1 A T_2\|_F^2$
10:     $\widehat{M} \leftarrow (T_1 C)^\dagger [P_{T_1 C} T_1 A T_2 P_{SAT_2}]_k (SAT_2)^\dagger$     $\triangleright$ $\widehat{M} \in \mathbb{R}^{k_1 \times k_2}$ and $\mathrm{rank}(\widehat{M}) = k$.
11:     Write $\widehat{M}$ into factored form $\widehat{M} = \widehat{X}\widehat{Y}$           $\triangleright$ $\widehat{X} \in \mathbb{R}^{k_1 \times k}, \widehat{Y} \in \mathbb{R}^{k \times k_2}$.
12:     **return** $C\widehat{X}, \widehat{Y}SA$ in factored form
13: **end procedure**

---

solving the above regression exactly is costly, so we employ a leverage score sampling matrix $S$ of matrix $C$, and consider the sketched regression

$$\min_{X:\mathrm{rank}(X)\leq k} \|SCX - SA\|_F^2,$$

letting $\widehat{X}$ denote the optimal solution to the above regression, then by Lemma A.13, we know that

$$\|A - C\widehat{X}\|_F^2 \leq (1+\epsilon) \min_{X:\mathrm{rank}(X)\leq k} \|A - CX\|_F^2$$
$$\leq (1+\epsilon)^2 \|A - A_k\|_F^2,$$

for simplicity, we scale $\epsilon$ so that the last inequality holds with multiplicative factor $1 + \epsilon$. To find $\widehat{X}$, we note that $\widehat{X} = (SC)^\dagger SA$, which means that the optimal solution lives in the row span of matrix $SA$. Writing $\widehat{X} = \widehat{Y}SA$, we see that

$$\min_{Y:\mathrm{rank}(Y)\leq k} \|A - CYSA\|_F^2 \leq (1+\epsilon)\|A - A_k\|_F^2.$$

To further speed up, we employ two leverage score samplings to reduce dimensions. Let $T_1$ be the leverage score sampling matrix of $C$, then by Lemma A.13, we could solve the regression $\min_{Z:\mathrm{rank}(Z)\leq k} \|T_1 A - T_1 C Z\|_F^2$ and recover $Y$ through $\min_Y \|Z - YSA\|_F^2$ (where the latter could be solved exactly), let $Y_1$ denote the optimal solution to the $Y$ recovered through this procedure and $Z_1$ be the optimal solution to the first regression, then $Y_1 = Z_1(SA)^\dagger$. $Z_1$ has the guarantee that

$$\|CZ_1 - A\|_F^2 \leq (1+\epsilon) \min_{Z:\mathrm{rank}(Z)\leq k} \|CZ - A\|_F^2$$
$$\leq (1+\epsilon)^2 \|A - A_k\|_F^2$$

and subsequently

$$\|CY_1 SA - A\|_F^2 \leq (1+\epsilon)^2 \|A - A_k\|_F^2,$$

follow the same argument, we could also sample according to the leverage score of $SA$ and sketch on the right. By properly scaling $\epsilon$, we could then conclude that the optimal cost of

$$\min_{Z:\mathrm{rank}(Z)\leq k} \|T_1 C Z S A T_2 - T_1 A T_2\|_F^2$$

is at most $1 + \epsilon$ factor of $\|A - A_k\|_F^2$, as desired.

For the running time, by Corollary C.2, generating $C$ takes $\widetilde{O}(\epsilon^{-1} n^{0.5} dk^{0.5} + dk^{\omega-1})$ time, generating the matrix $S$ with a total row count of $k_2$ takes $\widetilde{O}(\sqrt{dk_2}k_1 + k_1^\omega) = \widetilde{O}(\epsilon^{-1.5} d^{0.5} k^{1.5})$ time. Computing $SA$ is simply selecting and rescaling $k_2$ rows from $A$, which takes $O(nk_2) = \widetilde{O}(\epsilon^{-3} nk)$ time. Generating $T_2$ takes $\widetilde{O}(\sqrt{nk_3}k_2 + k_2^\omega) = \widetilde{O}(\epsilon^{-2} n^{0.5} k^{1.5})$ time. Finally, computing $T_1 C$, $SAT_2$, their pseudoinverses and projection takes $\mathrm{poly}(k/\epsilon)$ time, since forming these matrices is

simply selecting and rescaling entries, and the resulting matrices are of size $\text{poly}(k/\epsilon)$. Computing $T_1 A T_2$ takes $\text{poly}(k/\epsilon)$ by selecting and rescaling such number of entries from $A$, hence $\widehat{M}$ can be computed in $\text{poly}(k/\epsilon)$ time.

In summary, Algorithm 7 takes time

$$\widetilde{O}(\epsilon^{-1} n^{0.5} dk^{0.5} + dk^{\omega-1} + \epsilon^{-1.5} d^{0.5} k^{1.5} + \epsilon^{-2} n^{0.5} k^{1.5} + \epsilon^{-3} nk). \qquad \square$$

# D  QUANTUM KERNEL LOW-RANK APPROXIMATION

Given a set of points $\{x_1, \ldots, x_n\} \subset \mathbb{R}^d$ and a positive definite kernel function $\mathsf{K} : \mathbb{R}^d \times \mathbb{R}^d \to \mathbb{R}$, the kernel low-rank approximation problem asks to find a pair $M, N \in \mathbb{R}^{n \times k}$ such that $\|K - MN^\top\|_F^2 \leq (1+\epsilon) \cdot \|K - K_k\|_F^2$, where $K \in \mathbb{R}^{n \times n}$ is the kernel matrix induced by $\mathsf{K}$, with $K_{i,j} = \mathsf{K}(x_i, x_j)$. Note that explicitly forming the matrix $K$ takes $\Omega(n^2)$ evaluations of $\mathsf{K}(\cdot, \cdot)$, which is usually too expensive to be afforded. Since $\mathsf{K}$ is positive definite, there exists feature mapping $\phi : \mathbb{R}^d \to \mathbb{R}^m$ such that $K = \Phi \Phi^\top$ where $\Phi \in \mathbb{R}^{n \times m}$ with the $i$-th row being $\phi(x_i)$. Musco & Musco (2017) gives a low-rank approximation for $\Phi$ using $\widetilde{O}(\epsilon^{-2} nk)$ evaluations of $\mathsf{K}(\cdot, \cdot)$ and an additional $\widetilde{O}(\epsilon^{-2(\omega-1)} nk^{\omega-1})$ time. Musco & Woodruff (2017); Bakshi et al. (2020) show that the low-rank approximation guarantee can be achieved, albeit with $\widetilde{O}(\epsilon^{-1} nk)$ kernel evaluations and an additional $\widetilde{O}(\epsilon^{-(\omega-1)} nk^{\omega-1})$ time[3]. In this section, we will present a quantum algorithm based on the techniques developed in Section B and C, that computes a low-rank approximation for the kernel matrix using *sublinear number of kernel evaluations and additional operations.*

Before diving into our main result, we introduce some notations. We will extensively use $KD_1$ or $D_2^\top K D_1$ to denote a weighted sampling of $K$, in particular,

- If $D \in \mathbb{R}^{n \times t}$, we use $D^\top K_i \in \mathbb{R}^t$ to denote the vector $v$ with $v_j := D(j) \cdot \mathsf{K}(x_i, x_j)$, where $j \in D$ is the $j$-th sample of $D$, and $D(j)$ is the corresponding weight;

- If $D \in \mathbb{R}^{n \times t}$, we use "$KD$ in factored form" to denote a data structure that when $i$-th row is queried, compute $v \in \mathbb{R}^t$ where $v_j := D(j) \cdot \mathsf{K}(x_i, x_j)$ for $j \in D$.

- If $D_1 \in \mathbb{R}^{n \times t_1}$ and $D_2 \in \mathbb{R}^{n \times t_2}$, we use "$D_2^\top K D_1$ in factored form" to denote a data structure that supports queries to either row or column, where for $i$-th row, it computes a vector $v^{\text{row}} \in \mathbb{R}^{t_1}$ where $v_j^{\text{row}} := D_1(j) D_2(i) \cdot \mathsf{K}(x_i, x_j)$ for $j \in D_1$ and $i \in D_2$. Similarly the operation applies to the column.

- Sometimes given $KD \in \mathbb{R}^{n \times t_1}$ in factored form, we will compose it with another matrix $M \in \mathbb{R}^{t_1 \times t_2}$, we use "$KDM$ in factored form" to denote a data structure that supports row queries, such that when the $i$-th row is queried, it returns $Mv$ where $v_j := D(j) \cdot \mathsf{K}(x_i, x_j)$ for $j \in D$.

**Theorem D.1.** *There exists a quantum algorithm (Algorithm 8) that given any set of points $X = \{x_1, \ldots, x_n\} \subset \mathbb{R}^d$ and a positive definite kernel function $\mathsf{K} : \mathbb{R}^d \times \mathbb{R}^d \to \mathbb{R}$ and any positive integer $k \leq n$, $\epsilon \in (0, 1)$, runs in*

$$\widetilde{O}(n^{0.75} k^{1.25} \epsilon^{-1.25} (\mathcal{T}_\mathsf{K} + k\epsilon^{-1}) + n^{0.5} k^{1.5} \epsilon^{-2.5} (\mathcal{T}_\mathsf{K} + \epsilon^{-0.5}) + n^{0.5} k^{\omega-0.5} \epsilon^{0.5-\omega})$$

*time, where $\mathcal{T}_\mathsf{K}$ is the time to evaluate $\mathsf{K}$ on any pair of points $x_i, x_j$, and $\widetilde{O}(\epsilon^{-2} k^2 + \epsilon^{-2.5} n^{0.5} k^{1.5} + \epsilon^{-1.25} n^{0.75} k^{1.25})$ queries to the points in $X$, and returns a pair of rank-$k$ matrices $M, N \in \mathbb{R}^{n \times k}$ (given implicitly in factor form) such that*

$$\|K - MN^\top\|_F^2 \leq (1 + \epsilon) \|K - K_k\|_F^2$$

*holds with probability at least $0.99$.*

*Proof.* Our algorithm could be interpreted a quantum implemented of a generalization of Bakshi et al. (2020), where they only tackle the case where $\mathsf{K}(x_i, x_j) = x_i^\top x_j$, and we are given directly the kernel matrix $K$. We also note several differences between ours and Bakshi et al. (2020):

---

[3]Note that Musco & Woodruff (2017); Bakshi et al. (2020) phrase their algorithm as a low-rank approximation for PSD matrix $A$, and their runtime is stated in terms of reads to $A$. Observe that a read to an entry of $A$ could be translated to one kernel evaluation.

---

**Algorithm 8** Quantum kernel low-rank approximation.

---

1: **procedure** QLowRankKernel($\{x_1, \ldots, x_n\} \in (\mathbb{R}^d)^n, \mathsf{K} : \mathbb{R}^d \times \mathbb{R}^d \to \mathbb{R}, k, \epsilon, \delta$)

2:     $c \leftarrow 1000$

3:     $t \leftarrow c\sqrt{\frac{nk}{\epsilon}} \log(n/\delta)$

4:     $k' \leftarrow \widetilde{O}(k/\epsilon)$

5:     $D_1 \leftarrow$ QNyströmKernel($\{x_1, \ldots, x_n\}, \mathsf{K}, k', \delta/6$) with each GRLS scaled by $\sqrt{\frac{n}{\epsilon k}}$     $\triangleright$ Algorithm 9, $D_1 \in \mathbb{R}^{n \times t}$, oversample columns.

6:     $D_2 \leftarrow$ QNyströmKernel($x_1, \ldots, x_n\}, \mathsf{K}, k', \delta/6$) with each GRLS scaled by $\sqrt{\frac{n}{\epsilon k}}$     $\triangleright$ Algorithm 9, $D_2 \in \mathbb{R}^{n \times t}$, oversample rows.

7:     $C \leftarrow KD_1$ in factored form     $\triangleright C \in \mathbb{R}^{n \times t}$.

8:     $R \leftarrow D_2^\top KD_1$ in factored form     $\triangleright R \in \mathbb{R}^{t \times t}$.

9:     $\epsilon_0 \leftarrow 0.01$

10:     $\widetilde{R} \leftarrow$ QLowRankCMM($R, k/\epsilon, \epsilon_0, \delta/6$)     $\triangleright$ Corollary C.2, $\widetilde{R} \in \mathbb{R}^{t \times \epsilon^{-1} k \log(k/\delta)}$.

11:     $Z \leftarrow$ top-$k/\epsilon$ singular vectors of $\widetilde{R}$     $\triangleright Z \in \mathbb{R}^{t \times k/\epsilon}$

12:         $\triangleright$ Solve the regression $\min_{W \in \mathbb{R}^{n \times k/\epsilon}} \|C - WZ^\top\|$.

13:     Implement oracle for $p_i = \min\{1, \sqrt{\frac{n}{\epsilon k}} \cdot \|z_i\|_2^2\}$ where $z_i$ is the $i$-th row of $Z$

14:     $D_3 \leftarrow$ QSample($p$)     $\triangleright D_3 \in \mathbb{R}^{t \times k'}$.

15:         $\triangleright$ Solve the surrogate regression $\min_W \|CD_3 - WZ^\top D_3\|$.

16:     $W \leftarrow CD_3(Z^\top D_3)^\dagger$ in factored form     $\triangleright W = K(D_1 D_3 (Z^\top D_3)^\dagger) \in \mathbb{R}^{n \times k/\epsilon}$.

17:         $\triangleright$ Solve the regression $\min_{Y:\text{rank}(Y) \leq k} \|K - WYW^\top\|_F^2$.

18:     $D_4 \leftarrow$ QLS($W, k'/\epsilon^2, \delta/6$)     $\triangleright D_4 \in \mathbb{R}^{n \times k'/\epsilon^2}$, sample rows.

19:     $D_5 \leftarrow$ QLS($W, k'/\epsilon^2, \delta/6$)     $\triangleright D_5 \in \mathbb{R}^{n \times k'/\epsilon^2}$, sample columns.

20:     Compute $D_4^\top W$ and $W^\top D_5$     $\triangleright D_4^\top W \in \mathbb{R}^{k'/\epsilon^2 \times k/\epsilon}, W^\top D_5 \in \mathbb{R}^{k/\epsilon \times k'/\epsilon^2}$.

21:     $P_{D_4^\top W} \leftarrow D_4^\top W(W^\top D_4 D_4^\top W)^\dagger W^\top D_4, P_{W^\top D_5} \leftarrow W^\top D_5(D_5^\top WW^\top D_5)^\dagger D_5^\top W$

22:     Compute $D_4^\top KD_5$     $\triangleright D_4^\top KD_5 \in \mathbb{R}^{k'/\epsilon^2 \times k'/\epsilon^2}$.

23:     Compute $[P_{D_4^\top W}(D_4^\top KD_5)P_{W^\top D_5}]_k$     $\triangleright [P_{D_4^\top W}(D_4^\top AD_5)P_{W^\top D_5}]_k \in \mathbb{R}^{k'/\epsilon^2 \times k'/\epsilon^2}$ of rank-$k$.

24:     $Y_* \leftarrow (D_4^\top W)^\dagger[P_{D_4^\top W}(D_4^\top KD_5)P_{W^\top D_5}]_k(W^\top D_5)^\dagger$     $\triangleright Y_* \in \mathbb{R}^{k/\epsilon \times k/\epsilon}$ of rank-$k$.

25:     $U_* \leftarrow$ top-$k$ singular vectors of $Y_*$     $\triangleright U_* \in \mathbb{R}^{k/\epsilon \times k}$.

26:     $D_6 \leftarrow$ QLS($WU_*, k/\epsilon, \delta/6$)     $\triangleright D_6 \in \mathbb{R}^{n \times k/\epsilon}$.

27:         $\triangleright$ Solve the regression $\min_{N \in \mathbb{R}^{k \times n}} \|D_6^\top K - D_6^\top WU_*N\|_F^2$.

28:     $N \leftarrow (D_6^\top WU_*)^\dagger(D_6^\top K)$

29:     **return** $WU_*, N$ in factored form

30: **end procedure**

---

- To compute the initial $t \times t$ matrix, we use quantum Nyström method to sample from the generalized ridge leverage score of $K^{1/2}$, then rescale;

- To compute the low-rank approximation of the $t \times t$ matrix, we use quantum low-rank approximation algorithm developed in preceding section;

- To solve the spectral regression $\min_{W \in \mathbb{R}^{n \times k/\epsilon}} \|C - WZ^\top\|$, we use quantum sampling algorithm to sample from (rescaled) row norms of $Z$;

- The rank-constrained regression in Bakshi et al. (2020) is by first computing an orthonormal basis of $W$, denoted by $Q$, then solve the regression $\min_{X:\text{rank}(X) \leq k} \|K - QXQ^\top\|_F^2$. To solve this regression, Bakshi et al. (2020) samples rows and columns of $K$ according to column norms of $Q$, then solve the sketched regression after subsampling via these two matrices. Given the optimal solution $X_*$, Bakshi et al. (2020) finds an orthonormal basis of $X_*$ as $U_*$, set $M$ as $QU_*$ and then sample rows of $K$ according to row norms of $M$. In our case, we can't afford to form $Q$ (because $W \in \mathbb{R}^{n \times k/\epsilon}$), but we could instead solve the regression $\min_{Y:\text{rank}(Y) \leq k} \|K - WYW^\top\|_F^2$, then $X$ could be recovered via $Y \mapsto TYT^\top$ where $T$ is the change-of-basis matrix. We then solve all subsequent regression using $Y$ instead of $X$.

---

**Algorithm 9** Quantum generalized ridge leverage score sampling via recursive Nyström method.

---

1: **procedure** QNYSTRÖMKERNEL($\{x_1, \ldots, x_n\} \in (\mathbb{R}^d)^n, \mathsf{K} : \mathbb{R}^d \times \mathbb{R}^d \to \mathbb{R}^m, s, \delta$)
2:     $c \leftarrow 100$
3:     $T \leftarrow O(\log(n/s))$
4:     Let $S_0 \subset_{1/2} S_1 \subset_{1/2} \ldots \subset_{1/2} S_T = [n]$ ▷ Starting from $[n]$, uniformly sampling half of the indices.
5:     Set $k$ to be the largest integer with $ck \log(2k/\delta) \leq s$
6:     $M_0 \leftarrow \{\mathsf{K}(x_i, x_j)\}_{(i,j) \in S_0 \times S_0}$                          ▷ $|S_0| = s$.
7:     Let $D_0 \in \mathbb{R}^{n \times |S_0|}$ be the sampling matrix of $S_0$
8:     **for** $t = 1 \to T$ **do**
9:         $\lambda \leftarrow \frac{1}{k} \sum_{i=k+1}^{s} \sigma_i(M_{t-1})$
10:        $\widehat{M} \leftarrow (M_{t-1} + \lambda I_s)^{-1}$
11:        ▷ Let $D_{t-1}^\top K_i := \{D_{t-1}(j) \cdot \mathsf{K}(x_i, x_j)\}_{j \in D_{t-1}} \in \mathbb{R}^s$ for $i \in S_t$ where $D_{t-1}(j)$ is the weight corresponding to $x_j$ specified by $D_{t-1}$.
12:        Implement oracle for $q_i \leftarrow \frac{5}{\lambda} \cdot (\mathsf{K}(x_i, x_i) - (D_{t-1}^\top K_i)^\top \widehat{M} D_{t-1}^\top K_i)$ for $i \in S_t$
13:                                       ▷ $p_i = \min\{1, 16 q_i \log(2k/\delta)\}$.
14:        $\widetilde{D}_t \leftarrow \text{QSAMPLE}(p)$                          ▷ $\widetilde{D}_t \in \mathbb{R}^{|S_t| \times s}$.
15:        $D_t \leftarrow D_{S_t} \cdot \widetilde{D}_t$                          ▷ $D_t \in \mathbb{R}^{n \times s}$.
16:        $M_t \leftarrow \{D_t(i) D_t(j) \cdot \mathsf{K}(x_i, x_j)\}_{(i,j) \in D_t \times D_t}$          ▷ $M_t \in \mathbb{R}^{s \times s}$.
17:     **end for**
18:     **return** $D_T$
19: **end procedure**

---

To prove the correctness of the algorithm, we note that except for the above steps, all other steps are identical to the algorithm of Bakshi et al. (2020), so we just need to show our quantum implementation preserves key properties of Bakshi et al. (2020). For computing the sampling matrices $D_1$ and $D_2$, the only difference is when computing the generalized ridge leverage scores of $K^{1/2}$, Bakshi et al. (2020) uses fast matrix multiplication to compute all scores while we use quantum sampling algorithm to do so, so the guarantees of the sampling probabilities remain unchanged. The next major difference is we use quantum low-rank approximation of Corollary C.2, that provides precisely the desired $\epsilon$-coreset (and subsequently low-rank approximation). Forming the matrix $D_3$ is almost identical to that of Bakshi et al. (2020) except we use quantum sampling procedure to generate it.

We will focus on solving the rank-constrained regression $\min_{Y:\text{rank}(Y) \leq k} \|K - WYW^\top\|_F^2$, which is the major divergence of our approach and that of Bakshi et al. (2020). In Bakshi et al. (2020), since they could afford linear in $n$ time, they compute an orthonormal basis for $W$ denoted by $Q$, and instead solving the regression $\min_{X:\text{rank}(X) \leq k} \|K - QXQ^\top\|_F^2$. Let $T \in \mathbb{R}^{k/\epsilon \times k/\epsilon}$ be the change-of-basis matrix such that $QT = W$, then we observe that $X$ could be recovered via the following procedures:

- Solve

$$\min_{Y:\text{rank}(Y) \leq k} \|K - WYW^\top\|_F^2 \tag{1}$$

, let $Y_*$ denote the optimal solution of the above regression;

- Set $X_* := RY_*R^\top$.

To see $X_*$ is the optimal to the rank-constrained regression against $Q$, note

$$QX_*Q^\top = QRY_*R^\top Q^\top$$
$$= WY_*W^\top,$$

and if there exists a solution $X'$ with lower cost, then

$$\|K - WR^\top X'RW^\top\|_F^2 = \|K - QX'Q^\top\|_F^2$$
$$< \|K - QX_*Q^\top\|_F^2$$

$$= \|K - WY_*W^\top\|_F^2,$$

contradicting the definition of $Y_*$. Both Bakshi et al. (2020) and Algorithm 8 construct leverage score sampling matrices according to the leverage scores of $W$ (in the context of Bakshi et al. (2020), they sample according to the row norms of $Q$, which are the leverage scores of $W$), then we solve the surrogate regression

$$\min_{Y:\mathrm{rank}(Y)\leq k} \|D_4^\top K D_5 - D_4^\top W Y W^\top D_5\|_F^2, \tag{2}$$

it suffices to show that the optimal solution of Eq. (2) is a good approximation to the optimal solution of Eq. (1). To prove this, note that both $D_4$ and $D_5$ sample $k'/\epsilon^2$ rows and columns together with the fact $W \in \mathbb{R}^{n \times k'}$ implies that they are weak affine embeddings (Lemma A.12). However, $K - WYW^\top$ is not an affine subspace, so we could instead consider the matrix $H \in \mathbb{R}^{k' \times n}$ and let $H_* := \arg\min_{H \in \mathbb{R}^{k' \times n}} \|A - WH\|_F^2$ and $K_* = K - WH_*$. With probability at least $1 - \delta$, we have

$$\|D_4^\top K - D_4^\top WH\|_F^2 - \|D_4^\top K_*\|_F^2 = (1 \pm \epsilon) \cdot \|K - WH\|_F^2 - \|K_*\|_F^2,$$

for all $H \in \mathbb{R}^{k' \times n}$. Since it holds for all $H$, it in particular holds for all $H = YW^\top$, hence, with probability at least $1 - \delta$,

$$\|D_4^\top K - D_4^\top WYW^\top\|_F^2 - \|D_4^\top K_*\|_F^2 = (1 \pm \epsilon) \cdot \|K - WYW^\top\|_F^2 - \|K_*\|_F^2.$$

We could then run a symmetric argument on $D_5$: consider the regression $\min_{Z \in \mathbb{R}^{k'/\epsilon^2 \times k'}} \|D_4^\top K - ZW^\top\|_F^2$. Let $Z' := \arg\min_Z \|D_4^\top K - ZW^\top\|_F^2$ and $(D_4^\top K)' := D_4^\top K - Z'W^\top$. With probability at least $1 - \delta$ and due to Lemma A.12,

$$\|D_4^\top K D_5 - ZW^\top D_5\|_F^2 - \|(D_4^\top K)'D_5\|_F^2 = (1 \pm \epsilon) \cdot \|D_4^\top K - ZW^\top\|_F^2 - \|(D_4^\top K)'\|_F^2,$$

this holds for all $Z \in \mathbb{R}^{k'/\epsilon^2 \times k'}$ in particular $Z = D_4^\top WY$. Plug in such $Z$ yields

$$\|D_4^\top K D_5 - D_4^\top WYW^\top D_5\|_F^2 - \|(D_4^\top K)'D_5\|$$
$$= (1 \pm \epsilon)^2 \cdot (\|K - WYW^\top\|_F^2 + \|D_4^\top K_*\|_F^2 - \|K_*\|_F^2) - \|(D_4^\top K)'\|_F^2,$$

holds with probability at least $1 - 2\delta$. Observe that the additive error is at most $\Delta := (1 + \epsilon)^2(\|D_4^\top K_*\|_F^2 - \|K_*\|_F^2 + \|(D_4^\top K)'D_5\|_F^2 - \|(D_4^\top K)'\|_F^2)$, it is fixed and independence of $Y$. We will further show that the magnitude of $\Delta$ is small, let $\mathrm{OPT} := \min_{Y:\mathrm{rank}(Y)\leq k} \|K - WYW^\top\|_F^2$, then $\Delta = O(\mathrm{OPT})$. To see this, we first observe that

$$\|K_*\|_F^2 = \|K - WH_*\|_F^2$$
$$\leq \mathrm{OPT},$$

this is because $H_*$ is the optimal solution to a regression problem with larger solution space. Next, we will show $\|D_4^\top K_*\|_F^2$ is a constant approximation to $\|K_*\|_F^2$ with constant probability, via Markov's inequality:

$$\mathbb{E}[\|D_4^\top K_*\|_F^2] = \mathbb{E}[\mathrm{tr}[K_*^\top D_4 D_4^\top K_*]]$$
$$= \mathrm{tr}[K_*^\top \mathbb{E}[D_4 D_4^\top] K_*]$$
$$= \mathrm{tr}[K_*^\top I_n K_*]$$
$$= \|K_*\|_F^2,$$

since $D_4$ is a leverage score sampling matrix. Hence, by Markov's inequality, with probability at least $1 - 1/300$, $\|D_4^\top K_*\|_F^2 \leq 300\|K_*\|_F^2$. Hence, $\|D_4^\top K_*\|_F^2 - \|K_*\|_F^2 = O(\mathrm{OPT})$. Next, note that

$$\|(D^\top K)'\|_F^2 = \|D_4^\top K - Z'W^\top\|_F^2$$
$$\leq \min_{Y:\mathrm{rank}(Y)\leq k} \|D_4^\top K - D_4^\top WYW^\top\|_F^2$$
$$= O(\mathrm{OPT}),$$

where the second step is again, by $Z'$ is a solution to an optimization problem with larger solution space, and the last step is again, by Markov's inequality. By similar argument, we could conclude that $\|(D_4^\top K)'D_5\|_F^2 = O(\mathrm{OPT})$. Hence, we have shown that $\Delta = O(\mathrm{OPT})$.

Let $Y_* := \arg\min_{Y:\text{rank}(Y)\leq k} \|D_4^\top K D_5 - D_4^\top W Y W^\top D_5\|_F^2$, and set $g(X) = \|D_4^\top K D_5 - D_4^\top W X W^\top D_5\|_F^2$ to be the cost of approximate regression, and $f(X) = \|K - WXW^\top\|_F^2$ to be the cost of the exact regression respectively, then we could conclude with the preceding argument that

$$g(Y_*) \geq (1-\epsilon)f(Y_*) + \Delta, \tag{3}$$

on the other hand, if we let $Y'$ be the solution to $f$, i.e., $f(Y') = \text{OPT}$, then it must be the case that $g(Y_*) \leq g(Y')$ and similarly

$$g(Y') \leq (1+\epsilon)f(Y') + \Delta$$
$$= (1+\epsilon) \cdot \text{OPT} + \Delta \tag{4}$$

combining Eq. (3), (4) and the fact that $g(Y_*) \leq g(Y')$, we obtain

$$(1-\epsilon)f(Y_*) + \Delta \leq (1+\epsilon) \cdot \text{OPT} + \Delta,$$
$$f(Y_*) \leq \frac{1+\epsilon}{1-\epsilon} \cdot \text{OPT} + \frac{\epsilon}{1-\epsilon} \cdot \Delta$$
$$\leq (1+\epsilon)^2 \cdot \text{OPT} + O(\epsilon) \cdot \Delta$$
$$= (1+\epsilon)^2 \cdot \text{OPT} + O(\epsilon) \cdot \text{OPT}$$
$$= (1 + O(\epsilon)) \cdot \text{OPT},$$

as desired. This establishes that the optimal solution to Eq. (2) is a good approximation to Eq. (1), and the optimal solution of Eq. (2) admits a closed-form (see Theorem 4.15 of Bakshi et al. (2020)), which is precisely what has been computed on line 29 of Algorithm 8.

Observe that we already have a good (partial) low-rank approximation solution, as per the proof of Theorem 4.16 of Bakshi et al. (2020),

$$\min_{X:\text{rank}(X)\leq k} \|K - QXQ^\top\|_F^2 \leq (1+\epsilon) \cdot \|K - K_k\|_F^2,$$

and we have established that the value of Eq. (1) is the same as the LHS of the above inequality, hence we already have a rank-$k$ solution in factored form, which is $WY \in \mathbb{R}^{n \times k}$. Compute the top-$k$ left vectors of $Y_*$, denoted as $U_*$, and write $Y_* = U_* V_*$. Plug in the decomposition into the regression, we get

$$\|K - WU_* V_* W^\top\|_F^2 \leq (1+\epsilon)\|K - K_k\|_F^2,$$

by setting $M := WU_*$ and the right low-rank factor could be found by solving

$$\min_{N \in \mathbb{R}^{n \times k}} \|K - MN^\top\|_F^2 \leq \|K - WU_* V_* W^\top\|_F^2$$
$$\leq (1+\epsilon)\|K - K_k\|_F^2.$$

To solve the regression, we employ leverage score sampling on the rows of $M$, by Lemma A.13, it suffices to sample $k/\epsilon$ rows and the solution to the sketched regression

$$\min_{N \in \mathbb{R}^{n \times k}} \|D_6^\top K - D_6^\top M N^\top\|_F^2,$$

denoted by $N_*$, satisfies

$$\|K - MN_*^\top\|_F^2 \leq (1+\epsilon) \min_{N \in \mathbb{R}^{n \times k}} \|K - MN\|_F^2$$
$$\leq (1+\epsilon)^2 \|K - K_k\|_F^2.$$

Finally, by properly scaling $\epsilon$, we conclude the proof of correctness.

Next, we analyze the runtime of Algorithm 8, item by item as follows:

- Form the generalized ridge leverage score sampling matrix $D_1$ and $D_2$ (Algorithm 9) involves selecting $O(k'^2)$ entries from $K$, which could be implemented by $k'^2$ evaluations to the kernel function. In the loop, we compute the SVD of an $k' \times k'$ matrix, takes $O(k'^\omega)$ time, and forming $\widehat{M}$ also takes $O(k'^\omega)$ time. Next, we need to analyze the complexity of implementing the sampling oracle, for any fixed $i$, we form $D_{t-1}^\top K_i$ by forming a vector

of length $k'$ through $k'$ kernel evaluations and an extra $k'^2$ time for the quadratic form. To oversample $t$ rows/columns, we could simply scale the sampling probability, this yields a larger sum of all $p_i$'s:

$$\sum_{i=1}^{n} p_i = \widetilde{O}(\sqrt{nk/\epsilon}),$$

thus, the overall runtime of this part is

$$\widetilde{O}(\sqrt{n \sum_i p_i}) \cdot (k'\mathcal{T}_{\mathsf{K}} + k'^2) + k'^2\mathcal{T}_{\mathsf{K}} + k'^\omega$$

$$= \widetilde{O}(n^{0.75}k^{1.25}\epsilon^{-1.25}(\mathcal{T}_{\mathsf{K}} + k\epsilon^{-1})) + k^2\epsilon^{-2}(\mathcal{T}_{\mathsf{K}} + k^{\omega-2}\epsilon^{2-\omega}).$$

- For matrices $C$ and $R$, we do not explicit compute the data structure for them.

- Form the low-rank approximation $\widetilde{R}$ of matrix $R$, we need to show that the generic quantum sampling algorithm could be implemented even though the input is given in factor form. Observe that the algorithm requires uniformly sampling columns of the input matrix, which is oblivious to the input. To form the initial coreset $C_0$, we need to query a total of $\widetilde{O}(\sqrt{nk/\epsilon}) \times \widetilde{O}(k/\epsilon)$ entries of $K$, which can be done in $\widetilde{O}(n^{0.5}k^{1.5}\epsilon^{-1.5})$ kernel evaluations. Then we compute the SVD of this matrix, in time $\widetilde{O}(n^{0.5}k^{\omega-0.5}\epsilon^{0.5-\omega})$. Subsequently we need to impelement the classical ridge leverage score data structure (Algorithm 6), which can be done in time $\widetilde{O}(n^{0.5}k^{\omega-0.5}\epsilon^{0.5-\omega})$ and then apply the random Gaussian matrix takes $\widetilde{O}(n^{0.5}k^{1.5}\epsilon^{-1.5})$ time. To implement each query, we can form the query vector by $\widetilde{O}(n^{0.5}k^{0.5}\epsilon^{-0.5})$ kernel evaluations and an additional $\widetilde{O}(n^{0.5}k^{0.5}\epsilon^{-0.5})$ time. The total runtime is

$$\widetilde{O}(n^{0.75}k^{1.25}\epsilon^{-1.25} + n^{0.5}k^{1.5}\epsilon^{-1.5}) \cdot \mathcal{T}_{\mathsf{K}}.$$

- Form matrix $Z$ by computing SVD of $\widetilde{R}$, since $\widetilde{R} \in \mathbb{R}^{\sqrt{nk/\epsilon} \times k/\epsilon}$, this step could be done in time $O(\epsilon^{0.5-\omega}n^{0.5}k^{\omega-0.5})$.

- Form the sampling matrix $D_3$ involves sampling according to a rescaled row norm of $Z$, where each oracle call could be implemented in time $O(k/\epsilon)$ time, and the sum of $p_i$'s is

$$\sum_i p_i = \sqrt{\frac{n}{\epsilon k}} \cdot \sum_i \|z_i\|_2^2$$

$$= \sqrt{\frac{n}{\epsilon k}} \cdot \|Z\|_F^2$$

$$= \sqrt{\frac{nk}{\epsilon^3}}$$

because $Z$ has orthonormal columns. Thus, the overall runtime of this step is

$$\widetilde{O}(\sqrt{nk/\epsilon^4} \cdot k/\epsilon) = \widetilde{O}(n^{0.5}k^{1.5}\epsilon^{-3}).$$

- Form matrix $W$, we only need to explicitly compute $(Z^\top D_3)^\dagger$, which is a small matrix and could be computed in time $\widetilde{O}(k^\omega/\epsilon^\omega)$. Note that again, we won't explicit compute the data structure for $W$.

- Form the leverage score sampling matrix $D_4$ and $D_5$ with respect to $W$ and sample $k'/\epsilon^2$ rows/columns. The argument is similar to forming that of $\widetilde{R}$, except we use Algorithm 4 and the size of matrix $C$ is $\widetilde{O}(k/\epsilon^2) \times k/\epsilon$. Since we need to oversample $k'/\epsilon^2 = \widetilde{O}(k/\epsilon^3)$ rows and columns, we could scale the scores accordingly and make the sum of probabilities be at most $\widetilde{O}(k/\epsilon^3)$. To implement the oracle call, note that we need to make $\widetilde{O}(k^2\epsilon^{-3})$ kernel evaluations to form the initial matrix $C_0$, and subsequent operations such as SVD

and applying an JL matrix takes time $\widetilde{O}(k^\omega \epsilon^{-\omega-1})$. Then the query can be implemented by forming each row of $W$ using $k\epsilon^{-1}$ kernel evaluations with an additional $\widetilde{O}(k\epsilon^{-1})$ time. Thus, the total runtime is

$$\widetilde{O}(n^{0.5} k^{1.5} \epsilon^{-2.5}) \cdot \mathcal{T}_{\mathsf{K}}.$$

- Form matrix $D_4^\top W$ and $W^\top D_5$ could be done via selecting entries, in time $\widetilde{O}(k^2 \epsilon^{-4}) \cdot \mathcal{T}_{\mathsf{K}}$.

- Form the projection matrices $P_{D_4^\top W}$ and $P_{W^\top D_5}$ takes time $\mathrm{poly}(k/\epsilon)$.

- Form the matrix $D_4^\top K D_5$ is again selecting $\mathrm{poly}(k/\epsilon)$ entries from $K$, in time $\mathrm{poly}(k/\epsilon) \cdot \mathcal{T}_{\mathsf{K}}$.

- Compute $[P_{D_4^\top W}(D_4^\top W D_5) P_{W^\top D_5}]_k$ involves multiplying a sequence of $\mathrm{poly}(k/\epsilon)$ size matrices, and computing an SVD, which takes $\mathrm{poly}(k/\epsilon)$ time.

- Form the matrix $Y_*$ involves computing the pseudoinverse of $\mathrm{poly}(k/\epsilon)$ size matrices and multiplying them together, which takes $\mathrm{poly}(k/\epsilon)$ time. Computing the top-$k$ singular vectors of $Y_*$ also takes $\mathrm{poly}(k/\epsilon)$ time.

- Form the sampling matrix $D_6$ involves performing leverage score sampling according to matrix $WU_* \in \mathbb{R}^{n \times k}$ with a smaller target row count $k/\epsilon$, so the runtime is subsumed by the time to form $D_4$ and $D_5$.

- Finally, forming the matrix $N$ only requires computing $(D_6^\top WU_*)^\dagger$, which takes $\mathrm{poly}(k/\epsilon) \cdot \mathcal{T}_{\mathsf{K}}$ time.

Hence, the overall running time of Algorithm 8 is

$$\widetilde{O}(n^{0.75} k^{1.25} \epsilon^{-1.25}(\mathcal{T}_{\mathsf{K}} + k\epsilon^{-1}) + n^{0.5} k^{1.5} \epsilon^{-2.5}(\mathcal{T}_{\mathsf{K}} + \epsilon^{-0.5}) + n^{0.5} k^{\omega-0.5} \epsilon^{0.5-\omega}). \qquad \square$$

# E   QUANTUM $(k, p)$-SUBSPACE APPROXIMATION

In this section, we consider a generalized version of the $k$-subspace cost studied in Section B.3, for which we call the $(k, p)$-subspace cost (Woodruff & Yasuda, 2025): let $\mathcal{F}_k$ be the space of all $k$-dimensional subspace, then

$$\mathrm{cost}(A, x) = \left( \sum_{i=1}^n \|a_i^\top(I - P_x)\|_2^p \right)^{1/p}.$$

By defining the matrix $(p, 2)$-norm as

$$\|Y\|_{p,2} = \left( \sum_{i=1}^n \|e_i^\top Y\|_2^p \right)^{1/p},$$

then we could alternatively write the cost function as

$$\mathrm{cost}(A, F) = \|A(I - P_x)\|_{p,2}.$$

The $k$-subspace cost function we studied in Section B.3 is just the $(k, 2)$-subspace cost, and Woodruff & Yasuda (2025) has shown that, similar to the $k$-subspace cost, one could sample according to the powers of the ridge leverage score. We recall their main result in the following.

**Lemma E.1** (Theorem 3.9 and 3.11 of Woodruff & Yasuda (2025)). *Let $A \in \mathbb{R}^{n \times d}$ and $\epsilon \in (0, 1)$, let $S$ be the sampling matrix that samples according to the distribution $\{p_i\}_{i=1}^n$ where*

$$p_i = \begin{cases} \min\{1, n^{p/2-1} \overline{\tau}_i(A, \lambda_{A_s})^{p/2}/\alpha\}, & \text{if } p \geq 2, \\ \min\{1, \overline{\tau}_i(A, \lambda_{A_s})^{p/2}/\alpha\}, & \text{if } 1 \leq p < 2. \end{cases}$$

*Then, $\|SA(I - P_x)\|_{p,2} = (1 \pm \epsilon)\|A(I - P_x)\|_{p,2}$ for all $x \in \mathcal{F}_k$. Moreover,*

- *For $p \geq 2$, $\alpha = O(\epsilon^2)/\log^3 n$ and $s = O(k/\epsilon^p)$, and $S$ samples $O(k^{p/2}/\epsilon^{O(p^2)} \cdot \log^{O(p)} n)$ rows;*

- *For $1 \leq p < 2$, $\alpha = O(\epsilon^2)/\log^3 n$ and $s = O(k/\epsilon^2)$, and $S$ samples $O(k/\epsilon^{O(1)} \cdot \log^{O(1)} n)$ rows.*

*Finally, the algorithm runs in $\widetilde{O}(\mathrm{nnz}(A) + d^\omega)$ time.*

To speed up their algorithm, we note that the dominating runtime part is to sample from the rescaled leverage score distribution, and we could use Theorem B.23 with an inflated sample size.

**Theorem E.2.** *There exists a quantum algorithm that achieves the same guarantee as Lemma E.1 while*

- *For $p \geq 2$, it runs in time $\widetilde{O}(n^{1-1/p}dk^{0.5}/\epsilon^{p/2} + d^\omega)$ and uses $\widetilde{O}(n^{1-1/p}k^{0.5}/\epsilon^{p/2})$ queries to the points in $A$.*

- *For $p \in [1, 2)$, it runs in time $\widetilde{O}(n^{1-p/4}dk^{p/4}/\epsilon + d^\omega)$ and uses $\widetilde{O}(n^{1-p/4}k^{p/4}/\epsilon)$ queries to the points in $A$.*

*Proof.* By Theorem 3.9 and Theorem 3.11 of Woodruff & Yasuda (2025), we know that the sum of sampling probabilities could be upper bounded by $O(sn^{1-2/p})$ for $p \geq 2$ and $O(s^{p/2}n^{1-p/2})$ for $p \in [1, 2)$, meaning that for $p \geq 2$, we obtain a total number of queries being $\widetilde{O}(k^{0.5}n^{1-1/p}/\epsilon^{p/2})$ with per query cost $d$, plus the preprocessing time of $d^\omega$ gives the result. For $p \in [1, 2)$, this bound becomes $\widetilde{O}(k^{p/4}n^{1-p/4}/\epsilon)$. $\square$

# F    QUANTUM TENSOR LOW-RANK APPROXIMATION

In this section, we provide a quantum algorithm for computing the Frobenius norm low-rank approximation of a 3rd order tensor $A \in \mathbb{R}^{n \times n \times n}$. The goal is to find a rank-$k$ tensor $B := \sum_{i=1}^{k} u_i \otimes v_i \otimes w_i$ for $u_i, v_i, w_i \in \mathbb{R}^n$, such that $\|A - B\|_F^2 \leq (1+\epsilon) \cdot \mathrm{OPT}$ where $\mathrm{OPT} := \inf_{B:\mathrm{rank}(B)=k} \|A - B\|_F^2$. The first caveat is that such an optimal rank-$k$ solution might not even exist. We provide algorithms with $1 + \epsilon$ relative error when optimal rank-$k$ solution exists, and an additive error solution when it does not (in such case, $\mathrm{OPT} = 0$ so one has to allow small additive errors). We will then generalize the result for $q$-th order tensor where $q \geq 3$.

## F.1    PRELIMINARY

Given a 3rd order tensor $A \in \mathbb{R}^{n \times n \times n}$, we define the rank of $A$ as the smallest integer $k$ such that $A = \sum_{i=1}^{k} u_i \otimes v_i \otimes w_i$ where $u_i, v_i, w_i \in \mathbb{R}^n$. We use $\otimes$ to denote the Kronecker product of two matrices, i.e., for $A \in \mathbb{R}^{a \times b}, B \in \mathbb{R}^{c \times d}$, $A \otimes B \in \mathbb{R}^{ac \times bd}$ and $A \otimes B = \begin{bmatrix} A_{1,1}B & A_{1,2}B & \dots & A_{1,b}B \\ \vdots & \vdots & \dots & \vdots \\ A_{a,1}B & A_{a,2}B & \dots & A_{a,b}B \end{bmatrix}$.

We use $\odot$ to denote a product of two matrices defined as for $A \in \mathbb{R}^{a \times b}, B \in \mathbb{R}^{a \times d}$, $A \odot B \in \mathbb{R}^{a \times bd}$ where $A \odot B = \begin{bmatrix} A_{1,*} \otimes B_{1,*} \\ A_{2,*} \otimes B_{2,*} \\ \vdots \\ A_{a,*} \otimes B_{a,*} \end{bmatrix}$, i.e., the matrix formed by computing tensor product between corresponding rows of $A$ and $B$. Given a tensor $A \in \mathbb{R}^{n_1 \times n_2 \times n_3}$ and three matrices $B_1 \in \mathbb{R}^{n_1 \times d_1}, B_2 \in \mathbb{R}^{n_2 \times d_2}$ and $B_3 \in \mathbb{R}^{n_3 \times d_3}$, we define the $(\cdot, \cdot, \cdot)$ operator as

$$A(B_1, B_2, B_3)_{i,j,l} = \sum_{i'=1}^{n_1} \sum_{j'=1}^{n_2} \sum_{l'=1}^{n_3} A_{i',j',l'}(B_1)_{i',i}(B_2)_{j',j}(B_3)_{l',l}, \forall (i, j, l) \in [d_1] \times [d_2] \times [d_3],$$

subsequently, $A(B_1, B_2, B_3) \in \mathbb{R}^{d_1 \times d_2 \times d_3}$. One could also set any of the $B_i$'s as $I_{n_i}$ and for example, $A(B_1, I_{n_2}, I_{n_3}) \in \mathbb{R}^{d_1 \times n_2 \times n_3}$. When the dimension of the identity matrix is clear

from context, we abbreviate it as $I$ for notational simplicity. For $A \in \mathbb{R}^{n_1 \times n_2 \times n_3}$, we use $A_1 \in \mathbb{R}^{n_1 \times n_2 n_3}, A_2 \in \mathbb{R}^{n_2 \times n_1 n_3}$ and $A_3 \in \mathbb{R}^{n_3 \times n_1 n_2}$ to denote the three matrices such that the $[3] \setminus \{i\}$ dimensions are flattened.

We also state an algorithm due to Song et al. (2019) for sampling according to leverage scores of $U \odot V$:

**Lemma F.1.** *Given two matrices $U \in \mathbb{R}^{k \times n_1}$ and $V \in \mathbb{R}^{k \times n_2}$, there exists an algorithm*

$$\text{TENSORLEVERAGESCORE}(U, V, n_1, n_2, k, \epsilon, R_{\text{sample}})$$

*that takes*

$$O((n_1 + n_2) \cdot \text{poly}(\log(n_1 n_2), k, \epsilon^{-1}) \cdot R_{\text{sample}})$$

*time to generate a weighted sampling matrix $D \in \mathbb{R}^{n_1 n_2 \times R_{\text{sample}}}$ according to the leverage score distribution of the columns of $U \odot V$.*

To obtain our fixed-parameter tractable algorithm for rank-$k$ tensor low-rank approximation, we require the following result from Song et al. (2019):

**Lemma F.2.** *Let $\max\{t_i, d_i\} \leq n$, given a $t_1 \times t_2 \times t_3$ tensor $A$ and three matrices: $T_1 \in \mathbb{R}^{t_1 \times d_1}, T_2 \in \mathbb{R}^{t_2 \times d_2}$ and $T_3 \in \mathbb{R}^{t_3 \times d_3}$, if for any $\delta > 0$ there exist a solution to*

$$\min_{X_1, X_2, X_3} \| \sum_{i=1}^{k} (T_1 X_1)_i \otimes (T_2 X_2)_i \otimes (T_3 X_3)_i - A \|_F^2 := \text{OPT},$$

*and each entry of $X_i$ could be expressed with $O(n^\delta)$ bits, then there exists an algorithm that takes $n^{O(\delta)} \cdot 2^{O(d_1 k + d_2 k + d_3 k)}$ time and outputs three matrices $\widehat{X}_1, \widehat{X}_2$ and $\widehat{X}_3$ such that $\| \sum_{i=1}^{k} (T_1 \widehat{X}_1)_i \otimes (T_2 \widehat{X}_2)_i \otimes (T_3 \widehat{X}_3)_i - A \|_F^2 = \text{OPT}$.*

### F.2 APPROXIMATE REGRESSION VIA SAMPLING RESPONSES

The key we will be utilizing is the following lemma that, to solve a regression up to $(2 + \epsilon)$ factor, it is sufficient to sample the response matrix. As a consequence, we obtain a slew of tensor low-rank approximation algorithms with a $(4 + \epsilon)$-approximation ratio. This is worse than what is achieved in Song et al. (2019), but we would like to point out this is inherent due to all prior algorithms rely on *oblivious subspace embedding*. In fact, their algorithms utilize OSEs to show an *existence* argument: consider any rank-$k$ regression $\min_X \|XA - B\|_F^2$ where we do not have access to the design matrix $A$ but access to the target matrix $B$. One could still apply an OSE $S$ on the right of $A$ and solve the sketched regression $\min_X \|XAS^\top - BS^\top\|_F^2$ and argue the solution to the sketched regression is a good approximation. However, if one is only allowed to perform sampling procedures, then it is instructive to sample according to the structure of the unknown matrix $A$. In the following, we show that it is in fact enough to *sample from $B$*, this would not lead to a $1 + \epsilon$ approximate solution to the original regression problem, but we still manage to prove this is a $2 + \epsilon$ approximate solution. This is surprising — as an adversary could set $B$ so that the resulting sampling procedure misses all important entries of $A$. Hence, we devise an approach that utilizes the low-rank approximation of the sampled matrix $B$ to provide a good solution to the regression.

**Lemma F.3.** *Let $A \in \mathbb{R}^{k \times n}, B \in \mathbb{R}^{n \times d}$ and $\epsilon \in (0, 1)$, consider the following rank-constrained regression problem:*

$$\min_{X:\text{rank}(X) \leq k} \|XA - B\|_F^2, \tag{5}$$

*for $r = k/\epsilon^2$, let $S \in \mathbb{R}^{r \times n}$ be the ridge leverage score sampling matrix of $B$, then there exists a solution $X'$ in the column span of $BS^\top$, such that*

$$\|X'A - B\|_F^2 \leq (2 + \epsilon) \min_{X:\text{rank}(X) \leq k} \|XA - B\|_F^2.$$

*Proof.* Throughout the proof, let $\text{OPT} := \min_{X:\text{rank}(X) \leq k} \|XA - B\|_F^2$. We first note that if we sample columns of $B$ according its ridge leverage scores with $r$ columns, then we obtain an $\epsilon$-coreset of $B$ as for all rank-$k$ projection matrix $Q$,

$$(1 - \epsilon) \cdot \|B - QB\|_F^2 \leq \|BS^\top - QBS^\top\|_F^2 \leq (1 + \epsilon) \cdot \|B - QB\|_F^2,$$

in particular, let $Q_*$ be the projection onto the top-$k$ principal components of $B$, then the above suggests that

$$\|BS^\top - Q_* BS^\top\|_F^2 \leq (1+\epsilon) \cdot \|B - B_k\|_F^2$$
$$\leq (1+\epsilon) \cdot \text{OPT},$$

because $B_k$ is the optimal rank-$k$ solution. On the other hand, let $Q'$ be the projection onto the top-$k$ principal components of $BS^\top$, then

$$\|B - Q'B\|_F^2 \leq \frac{1}{1-\epsilon} \|BS^\top - Q'BS^\top\|_F^2$$
$$\leq \frac{1}{1-\epsilon} \|BS^\top - Q_* BS^\top\|_F^2$$
$$\leq \frac{1+\epsilon}{1-\epsilon} \cdot \text{OPT},$$

by scaling $\epsilon$, we get the cost is at most $1+\epsilon$ factor of OPT. We will set $X' := Q'BA^\dagger$, we obtain

$$\|X'A - B\|_F^2 = \|Q'BA^\dagger A - B\|_F^2$$
$$= \|(Q'B - B)A^\dagger A + B(A^\dagger A - I)\|_F^2$$
$$= \|Q'B - B\|_F^2 + \|B(I - A^\dagger A)\|_F^2$$
$$\leq (1+\epsilon) \cdot \text{OPT} + \|BA^\dagger A - B\|_F^2$$
$$\leq (1+\epsilon) \cdot \text{OPT} + \text{OPT}$$
$$= (2+\epsilon) \cdot \text{OPT}$$

where we use Pythagorean theorem and the fact that $BA^\dagger$ is the optimal solution to the regression. Write $BS^\top = U\Sigma V^\top$, then $Q' = U_k U_k^\top$, so $X'$ lies in the column span of $U_k$ which in turn, is a subset of the column span of $BS^\top$. $\qquad\square$

**Remark F.4.** *One might wonder whether the bound obtained in Lemma F.3 is loose, we provide an instance where sampling according to $B$ would necessarily give a $2$-approximation, hence showing the tightness of Lemma F.3. Consider both $A$ and $B$ are $n$-dimensional column vectors (hence $k = 1$), where $A = e_i + e_n$ for $i$ randomly chosen from $[n-1]$, and $B = e_n$. It is not hard to see that the optimal solution to the regression $\min_{x \in \mathbb{R}} \|Ax - B\|_2^2$ is given by $x = \frac{1}{2}$, with the cost $\frac{1}{2}$. On the other hand, if we perform any variant of importance sampling on $B$ would, with high probability, only hits the last entry of $B$ since all the mass is on the last entry, while missing the $i$-th entry for which $A$ is nonzero. Conditioning on this event, then the subsampled regression becomes $\min_{x \in \mathbb{R}} \|e_n x - e_n\|_2^2$ with an optimal solution $x' = 1$. Plug in $x'$ to the original regression would give a cost of $1$, which is only a $2$-approximation to the optimal cost.*

### F.3 QUANTUM BICRITERIA TENSOR LOW-RANK APPROXIMATION

We design a quantum bicriteria tensor low-rank approximation algorithm that outputs a rank-$k^2/\epsilon^4$ tensor that approximates rank-$k$ low-rank approximation of $A$.

**Theorem F.5.** *Given a 3rd order tensor $A \in \mathbb{R}^{n \times n \times n}$ and a positive integer $k \leq n$, $\epsilon \in (0, 0.1)$, there exists an algorithm (Algorithm 10) which takes $\widetilde{O}(\epsilon^{-1} n^2 k^{0.5} + n \operatorname{poly}(k/\epsilon))$ time and outputs three matrices $U, V, W \in \mathbb{R}^{n \times r}$ with $r = \widetilde{O}(k^2/\epsilon^4)$ such that*

$$\|\sum_{i=1}^r U_i \otimes V_i \otimes W_i - A\|_F^2 \leq (4+\epsilon) \cdot \min_{\operatorname{rank}-k \ A_k} \|A - A_k\|_F^2$$

*with probability 0.99.*

*Proof.* The proof will be similar to that of Theorem F.9. Let $U^*, V^*, W^*$ be the optimal rank-$k$ factor, set $Z_1 \in \mathbb{R}^{k \times n^2}$ to be the matrix where $i$-th row is $V_i^* \otimes W_i^*$, then clearly

$$\min_{U \in \mathbb{R}^{n \times k}} \|UZ_1 - A_1\|_F^2 = \text{OPT}$$

---

**Algorithm 10** Quantum bicriteria rank-$k^2/\epsilon^4$ tensor low-rank approximation algorithm.

---

1: **procedure** QBICRITERIATENSORLOWRANK($A \in \mathbb{R}^{n \times n \times n}, k, \epsilon$)
2:    $s_1, s_2 \leftarrow \widetilde{O}(k/\epsilon^2)$
3:    $C_1 \leftarrow$ QLOWRANKCMM($A_1, k, \epsilon, 0.001$)    ▷ $C_1 \in \mathbb{R}^{n \times s_1}$.
4:    $C_2 \leftarrow$ QLOWRANKCMM($A_2, k, \epsilon, 0.001$)    ▷ $C_2 \in \mathbb{R}^{n \times s_2}$.
5:    Form $\widehat{U}$ by repeating each column of $C_1$ by $s_2$ times    ▷ $\widehat{U} \in \mathbb{R}^{n \times s_1 s_2}$.
6:    Form $\widehat{V}$ by repeating each column of $C_2$ by $s_1$ times    ▷ $\widehat{V} \in \mathbb{R}^{n \times s_1 s_2}$.
7:    $s_3 \leftarrow O(s_1 s_2 \log(s_1 s_2) + s_1 s_2/\epsilon)$
8:    $\epsilon_0 \leftarrow 0.0001$
9:    $D_3 \leftarrow$ TENSORLEVERAGESCORE($\widehat{U}^\top, \widehat{V}^\top, n, n, s_1 s_2, \epsilon_0, s_3$)    ▷ $D_3 \in \mathbb{R}^{n^2 \times s_3}$.
10:    $B \leftarrow (\widehat{U}^\top \odot \widehat{V}^\top) D_3$    ▷ $B \in \mathbb{R}^{s_1 s_2 \times s_3}$.
11:    $\widehat{W} \leftarrow A_3 D_3 B^\dagger$
12:    **return** $\widehat{U}, \widehat{V}, \widehat{W}$
13: **end procedure**

---

where OPT is the optimal cost and the cost is achieved by picking $U$ as $U^*$. By Lemma F.3, there exists a solution $\overline{U} = C_1 X_1$ in the column span of $C_1$ such that

$$\|\overline{U} Z_1 - A_1\|_F^2 \le (2 + \epsilon) \cdot \text{OPT}, \tag{6}$$

we setup $Z_2 \in \mathbb{R}^{k \times n^2}$ where the $i$-th row of $Z_2$ is $\overline{U}_i \otimes W_i^*$, and consider the regression

$$\min_{V \in \mathbb{R}^{n \times k}} \|V Z_2 - A_2\|_F^2,$$

if we pick $V$ as $V^*$, then it degenerates to Eq. (6), so the optimal cost of the above regression is at most $(2 + \epsilon) \cdot \text{OPT}$. By Lemma F.3, we could find a solution $\overline{V} = C_2 X_2$ with

$$\|\overline{V} Z_2 - A_2\|_F^2 \le (2 + \epsilon)^2 \cdot \text{OPT}.$$

Finally, set $Z_3 \in \mathbb{R}^{k \times n^2}$ with the $i$-th row being $\overline{U}_i \otimes \overline{V}_i$, and we know that

$$\min_{W \in \mathbb{R}^{n \times k}} \|W Z_3 - A_3\|_F^2 \le (2 + \epsilon)^2 \cdot \text{OPT},$$

similar to the proof of Theorem F.9, we create $Z_3' \in \mathbb{R}^{s_1 s_2 \times n^2}$ such that $(Z_3')_{(i,j)} = (C_1)_i \otimes (C_2)_j$ hence $Z_3' = \widehat{U}^\top \odot \widehat{V}^\top$ for $\widehat{U}, \widehat{V}$ defined in Algorithm 10. As $Z_3$ is in the row span of $Z_3'$, we could alternatively consider

$$\min_{W \in \mathbb{R}^{n \times s_1 s_2}} \|W Z_3' - A_3\|_F^2$$

where one could solve up to $1 + \epsilon$ approximation by using leverage score sampling of matrix $Z_3'$, and the optimal solution is indeed given by $A_3 D_3 (Z_3' D_3)^\dagger$, which is precisely the matrix $\widehat{W}$ we have computed. Therefore, we end up with an approximate solution whose cost is at most $(2 + \epsilon)^2 (1 + \epsilon) \cdot \text{OPT} = (4 + O(\epsilon)) \cdot \text{OPT}$. The rank of these matrices is $s_1 s_2 = \widetilde{O}(k^2/\epsilon^4)$ as advertised.

Finally, for the running time, computing $C_1$ and $C_2$ takes $\widetilde{O}(\epsilon^{-1} n^2 k^{0.5} + n \operatorname{poly}(k/\epsilon))$ time, and computing the leverage score sampling matrix $D_3$ takes $O(n \operatorname{poly}(k/\epsilon))$ by Lemma F.1. Forming the matrix $B$ naïvely would take $O(n^2 k)$ time, but we could compute entries of $B$ on demand: the sampling matrix $D_3$ tells us which entries among the $n^2$ need to be computed, and one only needs to compute $s_3 = \operatorname{poly}(k/\epsilon)$ of them. Further, computing each entry takes $O(1)$ time, so the overall time to form $B$ is $\operatorname{poly}(k/\epsilon)$. Computing $A_3 D_3$ could be done via selecting a total of $n \operatorname{poly}(k/\epsilon)$ entries, so the overall runtime is

$$\widetilde{O}(\epsilon^{-1} n^2 k^{0.5} + n \operatorname{poly}(k/\epsilon)). \qquad \square$$

### F.4 QUANTUM TENSOR LOW-RANK APPROXIMATION: FIXED-PARAMETER TRACTABLE ALGORITHM

The main result of this subsection is the following:

**Theorem F.6.** *Given a 3rd order tensor $A \in \mathbb{R}^{n \times n \times n}$ such that each entry could be written with $O(n^\delta)$ bits for $\delta > 0$. Define $\mathrm{OPT} := \inf_{\mathrm{rank}-k} \|A - A_k\|_F^2$, for any $k \geq 1$ and $\epsilon \in (0,1)$, define $n^{\delta'} = O(n^\delta 2^{O(k^2/\epsilon)})$.*

- *If $\mathrm{OPT} > 0$, and there exists a tensor $A_k = U^* \otimes V^* \otimes W^*$ with $\|A - A_k\|_F^2 = \mathrm{OPT}$, and $\max\{\|U^*\|_F, \|V^*\|_F, \|W^*\|_F\} \leq 2^{O(n^{\delta'})}$, then there exists an algorithm that takes $(n^2 k^{0.5}/\epsilon + n \operatorname{poly}(k/\epsilon) + 2^{O(k^2/\epsilon)})n^\delta$ time in the unit cost RAM model with word size $O(\log n)$ bits and outputs $n \times k$ matrices $U, V, W$ such that*

$$\|U \otimes V \otimes W\|_F^2 \leq (4 + \epsilon) \mathrm{OPT} \tag{7}$$

  *with probability at least $0.99$ and entries of $U, V, W$ fit in $n^{\delta'}$ bits;*

- *If $\mathrm{OPT} > 0$ and $A_k$ does not exist, and there exists $U', V', W' \in \mathbb{R}^{n \times k}$ with $\max\{\|U'\|_F, \|V'\|_F, \|W'\|_F\} \leq 2^{O(n^{\delta'})}$ with $\|U' \otimes V' \otimes W' - A\|_F^2 \leq (1 + \epsilon/4) \mathrm{OPT}$, then we can find $U, V, W$ with Eq. (7) holds;*

- *If $\mathrm{OPT} = 0$ and $A_k$ does not exist and there exists a solution $U^*, V^*, W^*$ with each entry in $n^{O(\delta')}$ bits, then Eq. (7) holds;*

- *If $\mathrm{OPT} = 0$ and there exists three $n \times k$ matrices $U, V, W$ such that $\max\{\|U\|_F, \|V\|_F, \|W\|_F\} \leq 2^{O(n^{\delta'})}$ and*

$$\|U \otimes V \otimes W - A\|_F^2 \leq (4 + \epsilon)\mathrm{OPT} + 2^{-\Omega(n^{\delta'})} = 2^{-\Omega(n^{\delta'})},$$

  *then we can output $U, V, W$ with the above guarantee.*

*Further, if $A_k$ exists, we can output a number $Z$ such that $\mathrm{OPT} \leq Z \leq (4 + \epsilon)\mathrm{OPT}$. For all the cases above, the algorithm runs in the same time as the first case, and succeeds with probability at least $0.999$.*

The proof will be a consequence of Theorem F.7 and Lemma F.8, which we will discuss in the following sections.

### F.4.1 META ALGORITHM AND BOUNDED ENTRY ASSUMPTION

---
**Algorithm 11** Quantum FPT rank-$k$ low-rank approximation.

---
1: **procedure** QFPTLOWRANK($A, k, \epsilon$)                    ▷ Theorem F.7
2:   $s_1 \leftarrow s_2 \leftarrow \widetilde{O}(k/\epsilon^2)$
3:   $C_1 \leftarrow$ QLOWRANKCMM($A_1, k, \epsilon, 0.0001$)         ▷ $C_1 \in \mathbb{R}^{n \times s_1}$.
4:   $C_2 \leftarrow$ QLOWRANKCMM($A_2, k, \epsilon, 0.0001$)         ▷ $C_2 \in \mathbb{R}^{n \times s_2}$.
5:   Form $B_1$ by consecutively repeating each column of $C_1$ by $s_2$ times
6:   Form $B_2$ by consecutively repeating each column of $C_2$ by $s_1$ times
7:   $d_3 \leftarrow O(s_1 s_2 \log(s_1 s_2) + s_1 s_2/\epsilon)$
8:   $D_3 \leftarrow$ TENSORLEVERAGESCORE($B_1^\top, B_2^\top, n, n, s_1 s_2, \epsilon_0, d_3$)
9:   $M_3 \leftarrow A_3 D_3$
10:  $Y_1, Y_2, Y_3, C \leftarrow$ QSUBLINEARREDUCTION($A, A_1 S_1, A_2 S_2, A_3 S_3, n, s_1, s_2, d_3, k, \epsilon$).   ▷ Algorithm 12
11:    Create variables for $X_i \in \mathbb{R}^{s_i \times k}, \forall i \in [3]$
12:    Run polynomial system verifier for $\|(Y_1 X_1) \otimes (Y_2 X_2) \otimes (Y_3 X_3) - C\|_F^2$
13:    **return** $C_1 X_1, C_2 X_2$, and $M_3 X_3$
14: **end procedure**

---

**Theorem F.7.** *Given a 3rd order tensor $A \in \mathbb{R}^{n \times n \times n}$, for any $k \geq 1, \epsilon \in (0,1)$ and $\delta > 0$, there is a quantum algorithm which takes $n^2 k^{0.5}/\epsilon + n^{O(\delta)} 2^{O(k^2/\epsilon)}$ time where $\delta$ is defined as in Lemma F.2,*

$\widetilde{O}(nk^{0.5}/\epsilon)$ *queries to the rows, columns and tubes of A, and outputs three matrices* $U \in \mathbb{R}^{n \times k}$, $V \in \mathbb{R}^{n \times k}$, $W \in \mathbb{R}^{n \times k}$ *such that*

$$\left\| \sum_{i=1}^{k} U_i \otimes V_i \otimes W_i - A \right\|_F^2 \leq (4+\epsilon) \min_{\text{rank}-k \ A_k} \|A_k - A\|_F^2$$

*holds with probability* 0.99.

*Proof.* We define OPT as

$$\text{OPT} = \min_{\text{rank}-k \ A'} \|A' - A\|_F^2.$$

Suppose the optimal $A_k = U^* \otimes V^* \otimes W^*$. We fix $V^* \in \mathbb{R}^{n \times k}$ and $W^* \in \mathbb{R}^{n \times k}$. We use $V_1^*, V_2^*, \cdots, V_k^*$ to denote the columns of $V^*$ and $W_1^*, W_2^*, \cdots, W_k^*$ to denote the columns of $W^*$. We consider the following optimization problem,

$$\min_{U_1, \cdots, U_k \in \mathbb{R}^n} \left\| \sum_{i=1}^{k} U_i \otimes V_i^* \otimes W_i^* - A \right\|_F^2,$$

which is equivalent to

$$\min_{U_1, \cdots, U_k \in \mathbb{R}^n} \left\| \begin{bmatrix} U_1 & U_2 & \cdots & U_k \end{bmatrix} \begin{bmatrix} V_1^* \otimes W_1^* \\ V_2^* \otimes W_2^* \\ \cdots \\ V_k^* \otimes W_k^* \end{bmatrix} - A \right\|_F^2.$$

We use matrix $Z_1$ to denote $\begin{bmatrix} \text{vec}(V_1^* \otimes W_1^*) \\ \text{vec}(V_2^* \otimes W_2^*) \\ \cdots \\ \text{vec}(V_k^* \otimes W_k^*) \end{bmatrix} \in \mathbb{R}^{k \times n^2}$ and matrix $U$ to denote $\begin{bmatrix} U_1 & U_2 & \cdots & U_k \end{bmatrix}$. Then we can obtain the following equivalent objective function,

$$\min_{U \in \mathbb{R}^{n \times k}} \|UZ_1 - A_1\|_F^2.$$

Notice that $\min_{U \in \mathbb{R}^{n \times k}} \|UZ_1 - A_1\|_F^2 = \text{OPT}$, since $A_k = U^* Z_1$. By Lemma F.3, we know that if we sample columns of $A_1$ according to its ridge leverage score distribution with $\widetilde{O}(k/\epsilon^2)$ columns and let $C_1$ denote the resulting matrix, then there exists a solution $\widehat{U} = C_1 X_1$ in the column span of $C_1$, such that

$$\|\widehat{U} Z_1 - A_1\|_F^2 \leq (2+\epsilon) \min_{U \in \mathbb{R}^{n \times k}} \|UZ_1 - A_1\|_F^2$$
$$= (2+\epsilon) \cdot \text{OPT},$$

which implies

$$\left\| \sum_{i=1}^{k} \widehat{U}_i \otimes V_i^* \otimes W_i^* - A \right\|_F^2 \leq (2+\epsilon) \cdot \text{OPT}.$$

To write down $\widehat{U}_1, \cdots, \widehat{U}_k$, we use the given matrix $A_1$, and we create $s_1 \times k$ variables for matrix $X_1$.

As our second step, we fix $\widehat{U} \in \mathbb{R}^{n \times k}$ and $W^* \in \mathbb{R}^{n \times k}$, and we convert tensor $A$ into matrix $A_2$. Let matrix $Z_2$ denote $\begin{bmatrix} \text{vec}(\widehat{U}_1 \otimes W_1^*) \\ \text{vec}(\widehat{U}_2 \otimes W_2^*) \\ \cdots \\ \text{vec}(\widehat{U}_k \otimes W_k^*) \end{bmatrix}$. We consider the following objective function,

$$\min_{V \in \mathbb{R}^{n \times k}} \|VZ_2 - A_2\|_F^2,$$

for which the optimal cost is at most $(2 + \epsilon) \cdot \text{OPT}$.

By playing a similar argument and utilizing Lemma F.3, we could obtain matrix $C_2$ with $\widetilde{O}(k/\epsilon^2)$ rescaled columns of $A_2$, such that there exists a solution $\widehat{V} = C_2 X_2$ with

$$\|\widehat{V}Z_2 - A_2\|_F^2 \leq (2 + \epsilon) \min_{V \in \mathbb{R}^{n \times k}} \|VZ_2 - A_2\|_F^2 \leq (2 + \epsilon)^2 \cdot \text{OPT},$$

which implies

$$\left\| \sum_{i=1}^k \widehat{U}_i \otimes \widehat{V}_i \otimes W_i^* - A \right\|_F^2 \leq (2 + \epsilon)^2 \cdot \text{OPT}.$$

To write down $\widehat{V}_1, \cdots, \widehat{V}_k$, we need to use the given matrix $A_2$, and we need to create $s_2 \times k$ variables for matrix $X_2$.

As our third step, we fix the matrices $\widehat{U} \in \mathbb{R}^{n \times k}$ and $\widehat{V} \in \mathbb{R}^{n \times k}$. Let matrix $Z_3$ denote $\begin{bmatrix} \text{vec}(\widehat{U}_1 \otimes \widehat{V}_1) \\ \text{vec}(\widehat{U}_2 \otimes \widehat{V}_2) \\ \cdots \\ \text{vec}(\widehat{U}_k \otimes \widehat{V}_k) \end{bmatrix}$. We convert tensor $A \in \mathbb{R}^{n \times n \times n}$ into matrix $A_3 \in \mathbb{R}^{n \times n^2}$. Since $\widehat{U} = C_1 X_1$ and $\widehat{V} = C_2 X_2$, define the matrix $Z_3' \in \mathbb{R}^{d_3 \times n^2}$ where, if we use $(i, j)$ to index rows of $Z_3'$, then $(Z_3')_{(i,j)} = (C_1)_i \otimes (C_2)_j$, and a key observation is there exists a matrix $Y \in \mathbb{R}^{k \times d_3}$ with $Z_3 = Y Z_3'$. To form $Z_3'$, we take the approach of forming $B_1$ and $B_2$ by repeating columns a fixed number of times, for example, $B_1$ is defined as

$$[(C_1)_1 \quad (C_1)_1 \quad \ldots \quad (C_1)_1 \quad \ldots \quad (C_1)_k \quad \ldots \quad (C_1)_k]$$

where each column is repeated for $s_2$ times, and one could verify that $Z_3' = B_1 \odot B_2$.

We consider the following objective function,

$$\min_{W \in \mathbb{R}^{n \times k}} \|W Z_3 - A_3\|_F^2,$$

which is equivalent to

$$\min_{W \in \mathbb{R}^{n \times k}, Y \in \mathbb{R}^{k \times d_3}} \|W Y Z_3' - A_3\|_F^2,$$

if we employ leverage score sampling on the columns of $Z_3'$, then by Lemma A.13, we could find a pair of matrices $\widehat{W}, \widehat{Y}$ with

$$\|\widehat{W}\widehat{Y}Z_3' - A_3\|_F^2 \leq (1 + \epsilon) \min_{W \in \mathbb{R}^{n \times k}, Y \in \mathbb{R}^{k \times d_3}} \|W Y Z_3' - A_3\|_F^2$$

$$= (1 + \epsilon) \min_{W \in \mathbb{R}^{n \times k}} \|W Z_3 - A_3\|_F^2$$

$$\leq (1 + \epsilon)(2 + \epsilon)^2 \cdot \text{OPT}.$$

We briefly explain how to obtain the factorization of $\widehat{W}, \widehat{Y}$, consider solving the regression

$$\min_{T \in \mathbb{R}^{n \times d_3}} \|T Z_3' D_3 - A_3 D_3\|_F^2$$

where $D_3 \in \mathbb{R}^{n^2 \times d_3}$ is the leverage score sampling matrix of $Z_3'$, then $T = A_3 D_3 (Z_3' D_3)^\dagger$ and we could take the top-$k$ left singular vectors as $\widehat{W}$ and the remaining part as $\widehat{Y}$. All we have shown is that $\widehat{W}$ is in the column span of $A_3 D_3$ with a cost at most $(1 + \epsilon)(2 + \epsilon)^2$ of the optimal cost, as $\widehat{W} = T P_k = A_3 D_3 (Z_3' D_3)^\dagger P_k$ where $P_k$ is the projection onto the top-$k$ left singular vectors of $T$.

Thus, we have established that

$$\min_{X_1, X_2, X_3} \left\| \sum_{i=1}^k (C_1 X_1)_i \otimes (C_2 X_2)_i \otimes (A_3 D_3 X_3)_i - A \right\|_F^2 \leq (1 + \epsilon)(2 + \epsilon)^2 \cdot \text{OPT}.$$

Let $V_1 = C_1, V_2 = C_2, V_3 = A_3 D_3$, we then apply Lemma F.8, and we obtain $\widehat{V}_1, \widehat{V}_2, \widehat{V}_3, C$. We then apply Lemma F.2. Correctness follows by rescaling $\epsilon$ by a constant factor and note that $(1 + \epsilon)(2 + \epsilon)^2 = 4 + O(\epsilon)$.

**Running time.** Regarding the running time, computing $C_1$ and $C_2$ takes $\widetilde{O}(\epsilon^{-1}n^2k^{0.5} + n\operatorname{poly}(k/\epsilon))$ time, and computing $D_3$ takes $\widetilde{O}(n\operatorname{poly}(k/\epsilon))$ time. To create matrices $Y_1, Y_2, Y_3$ and $C$, by Lemma F.8, it takes $\widetilde{O}(n^{0.5}\operatorname{poly}(k/\epsilon))$ time, and the runtime of the polynomial system verifier is due to Lemma F.2. $\qquad\square$

### F.4.2 Input Size Reduction in Sublinear Time

---

**Algorithm 12** Input size reduction via leverage score sampling.

---

1: **procedure** QSUBLINEARREDUCTION($A, V_1, V_2, V_3, n, b_1, b_2, b_3, k, \epsilon$) $\qquad\qquad$ ▷ Lemma F.8
2: $\qquad c_1 \leftarrow c_2 \leftarrow c_3 \leftarrow \operatorname{poly}(k/\epsilon)$
3: $\qquad T_1 \leftarrow \mathrm{QLS}(V_1, c_1, 0.0001)$
4: $\qquad T_2 \leftarrow \mathrm{QLS}(V_2, c_2, 0.0001)$
5: $\qquad T_3 \leftarrow \mathrm{QLS}(V_3, c_3, 0.0001)$
6: $\qquad \widehat{V}_i \leftarrow T_i V_i \in \mathbb{R}^{c_i \times b_i}, \forall i \in [3]$.
7: $\qquad C \leftarrow A(T_1, T_2, T_3) \in \mathbb{R}^{c_1 \times c_2 \times c_3}$
8: $\qquad$ **return** $\widehat{V}_1, \widehat{V}_2, \widehat{V}_3$ and $C$
9: **end procedure**

---

**Lemma F.8.** *Let* $\operatorname{poly}(k/\epsilon) \geq b_1 b_2 b_3 \geq k$. *Given a tensor* $A \in \mathbb{R}^{n \times n \times n}$ *and three matrices* $V_1 \in \mathbb{R}^{n \times b_1}$, $V_2 \in \mathbb{R}^{n \times b_2}$, *and* $V_3 \in \mathbb{R}^{n \times b_3}$, *there exists an algorithm that takes* $n^{0.5} \cdot \operatorname{poly}(k/\epsilon)$ *time,* $\widetilde{O}(n^{0.5})$ *row queries to* $V_1, V_2, V_3$, *and outputs a tensor* $C \in \mathbb{R}^{c_1 \times c_2 \times c_3}$ *and three matrices* $\widehat{V}_1 \in \mathbb{R}^{c_1 \times b_1}$, $\widehat{V}_2 \in \mathbb{R}^{c_2 \times b_2}$ *and* $\widehat{V}_3 \in \mathbb{R}^{c_3 \times b_3}$ *with* $c_1 = c_2 = c_3 = \operatorname{poly}(k/\epsilon)$, *such that with probability at least* 0.99, *for all* $\alpha > 0, X_1, X_1' \in \mathbb{R}^{b_1 \times k}, X_2, X_2' \in \mathbb{R}^{b_2 \times k}, X_3, X_3' \in \mathbb{R}^{b_3 \times k}$ *satisfy that,*

$$\left\| \sum_{i=1}^k (\widehat{V}_1 X_1')_i \otimes (\widehat{V}_2 X_2')_i \otimes (\widehat{V}_3 X_3')_i - C \right\|_F^2 \leq \alpha \left\| \sum_{i=1}^k (\widehat{V}_1 X_1)_i \otimes (\widehat{V}_2 X_2)_i \otimes (\widehat{V}_3 X_3)_i - C \right\|_F^2,$$

*then,*

$$\left\| \sum_{i=1}^k (V_1 X_1')_i \otimes (V_2 X_2')_i \otimes (V_3 X_3')_i - A \right\|_F^2 \leq (1+\epsilon)\alpha \left\| \sum_{i=1}^k (V_1 X_1)_i \otimes (V_2 X_2)_i \otimes (V_3 X_3)_i - A \right\|_F^2.$$

*Proof.* Let $X_1 \in \mathbb{R}^{b_1 \times k}, X_2 \in \mathbb{R}^{b_2 \times k}, X_3 \in \mathbb{R}^{b_3 \times k}$. Define $\mathrm{OPT} := \| \sum_{i=1}^k (V_1 X_1)_i \otimes (V_2 X_2)_i \otimes (V_3 X_3)_i - A\|_F^2$. First, we define $Z_1 = ((V_2 X_2)^\top \odot (V_3 X_3)^\top) \in \mathbb{R}^{k \times n^2}$. (Note that, for each $i \in [k]$, the $i$-th row of matrix $Z_1$ is $\operatorname{vec}((V_2 X_2)_i \otimes (V_3 X_3)_i)$.) Then, by flattening we have

$$\left\| \sum_{i=1}^k (V_1 X_1)_i \otimes (V_2 X_2)_i \otimes (V_3 X_3)_i - A \right\|_F^2 = \|V_1 X_1 \cdot Z_1 - A_1\|_F^2.$$

We choose a sparse diagonal sampling matrix $T_1 \in \mathbb{R}^{c_1 \times n}$ with $c_1 = \operatorname{poly}(k, 1/\epsilon)$ rows. Let $Y_1 := \arg\min_{Y \in b_1 \times n^2} \|V_1 Y - A_1\|_F^2$ and $A_1^* := V_1 Y_1 - A_1$. Since $V_1$ has $b_1 \leq \operatorname{poly}(k/\epsilon)$ columns, according to Lemma A.12 with probability 0.999, for all $X_1 \in \mathbb{R}^{b_1 \times k}, Z \in \mathbb{R}^{k \times n^2}$,

$$(1-\epsilon)\|V_1 X_1 Z - A_1\|_F^2 - \|A_1^*\|_F^2 \leq \|T_1 V_1 X_1 Z - T_1 A_1\|_F^2 - \|T_1 A_1^*\|_F^2$$
$$\leq (1+\epsilon)\|V_1 X_1 Z - A_1\|_F^2 - \|A_1^*\|_F^2.$$

Therefore, we have

$$\|T_1 V_1 X_1 \cdot Z_1 - T_1 A_1\|_F^2$$

$$= (1 \pm \epsilon) \left\| \sum_{i=1}^k (V_1 X_1)_i \otimes (V_2 X_2)_i \otimes (V_3 X_3)_i - A \right\|_F^2 + \underbrace{\|T_1 A_1^*\|_F^2 - \|A_1^*\|_F^2}_{\Delta_1}.$$

Second, we unflatten matrix $T_1 A_1 \in \mathbb{R}^{c_1 \times n^2}$ to obtain a tensor $A' \in \mathbb{R}^{c_1 \times n \times n}$. Then we flatten $A'$ along the second direction to obtain $A_2 \in \mathbb{R}^{n \times c_1 n}$. We define $Z_2 = (T_1 V_1 X_1)^\top \odot (V_3 X_3)^\top \in \mathbb{R}^{k \times c_1 n}$. Then, by flattening,

$$\|V_2 X_2 \cdot Z_2 - A_2\|_F^2 = \|T_1 V_1 X_1 \cdot Z_1 - T_1 A_1\|_F^2$$

$$= (1 \pm \epsilon) \left\| \sum_{i=1}^k (V_1 X_1)_i \otimes (V_2 X_2)_i \otimes (V_3 X_3)_i - A \right\|_F^2 + \Delta_1.$$

We choose a diagonal sampling matrix $T_2 \in \mathbb{R}^{c_2 \times n}$ with $c_2 = \text{poly}(k, 1/\epsilon)$ rows. Then according to Lemma A.12 with probability 0.999, for all $X_2 \in \mathbb{R}^{b_2 \times k}$, $Z \in \mathbb{R}^{k \times c_1 n}$,

$$(1 - \epsilon)\|V_2 X_2 Z - A_2\|_F^2 - \|A_2^*\|_F^2 \le \|T_2 V_2 X_2 Z - T_2 A_2\|_F^2 - \|T_2 A_2^*\|_F^2$$

$$\le (1 + \epsilon)\|V_2 X_2 Z - A_2\|_F^2 - \|A_2^*\|_F^2,$$

for $A_2^*$ defined similarly as $A_1^*$. Define $\Delta_2 = \|T_2 A_2^*\|_F^2 - \|A_2^*\|_F^2$, we have

$$\|T_2 V_2 X_2 \cdot Z_2 - T_2 A_2\|_F^2$$

$$= (1 \pm \epsilon)\|V_2 X_2 \cdot Z_2 - A_2\|_F^2$$

$$= (1 \pm \epsilon)^2 \left\| \sum_{i=1}^k (V_1 X_1)_i \otimes (V_2 X_2)_i \otimes (V_3 X_3)_i - A \right\|_F^2 + (1 \pm \epsilon)\Delta_1 + \Delta_2.$$

Third, we unflatten matrix $T_2 A_2 \in \mathbb{R}^{c_2 \times c_1 n}$ to obtain a tensor $A''(= A(T_1, T_2, I)) \in \mathbb{R}^{c_1 \times c_2 \times n}$. Then we flatten tensor $A''$ along the last direction (the third direction) to obtain matrix $A_3 \in \mathbb{R}^{n \times c_1 c_2}$. We define $Z_3 = (T_1 V_1 X_1)^\top \odot (T_2 V_2 X_2)^\top \in \mathbb{R}^{k \times c_1 c_2}$. Then, by flattening, we have

$$\|V_3 X_3 \cdot Z_3 - A_3\|_F^2 = \|T_2 V_2 X_2 \cdot Z_2 - T_2 A_2\|_F^2$$

$$= (1 \pm \epsilon)^2 \left\| \sum_{i=1}^k (V_1 X_1)_i \otimes (V_2 X_2)_i \otimes (V_3 X_3)_i - A \right\|_F^2 + (1 \pm \epsilon)\Delta_1 + \Delta_2.$$

We choose a diagonal sampling matrix $T_3 \in \mathbb{R}^{c_3 \times n}$ with $c_3 = \text{poly}(k, 1/\epsilon)$ rows. Then according to Lemma A.12 with probability 0.999, for all $X_3 \in \mathbb{R}^{b_3 \times k}$, $Z \in \mathbb{R}^{k \times c_1 c_2}$,

$$(1 - \epsilon)\|V_3 X_3 Z - A_3\|_F^2 + \Delta_3 \le \|T_3 V_3 X_3 Z - T_3 A_3\|_F^2 \le (1 + \epsilon)\|V_3 X_3 Z - A_3\|_F^2 + \Delta_3$$

for $\Delta_3 := \|A_3^*\|_F^2 - \|T_3 A_3^*\|_F^2$. Therefore, we have

$$\|T_3 V_3 X_3 \cdot Z_3 - T_3 A_3\|_F^2$$

$$= (1 \pm \epsilon)^3 \left\| \sum_{i=1}^k (V_1 X_1)_i \otimes (V_2 X_2)_i \otimes (V_3 X_3)_i - A \right\|_F^2 + (1 \pm \epsilon)^2 \Delta_1 + (1 \pm \epsilon)\Delta_2 + \Delta_3.$$

We will argue the additive error terms are small. Examine the term $\Delta_1$, in particular $\|A_1 - V_1 Y_1\|_F^2$, it is not hard to see that $\|A_1 - V_1 Y_1\|_F^2 \le \text{OPT}$ as one could simply realize the cost by choosing $Y_1$ according to $V_2 X_2$ and $V_3 X_3$. By Markov's inequality and the leverage score sampling matrix $T_1$ is an unbiased estimator for the matrix Frobenious norm squared, we could conclude the term $\|T_1 A_1^*\|_F^2 = O(\text{OPT})$ holds with constant probability. Similarly, for $\|A_2^*\|_F^2$, we see that $\|V_2 Y_2 - A_2\|_F^2 \le \text{OPT}$ by choosing $Y_2$ according to the other two factors. One could conclude analogously that $\Delta_2, \Delta_3 = O(\text{OPT})$. Let $\Delta$ be the sum of all additive error terms, and we have $\Delta = O(\text{OPT})$.

Let $\widehat{V}_i$ denote $T_i V_i$ for all $i \in [3]$ and $C \in \mathbb{R}^{c_1 \times c_2 \times c_3}$, and for $\alpha > 1$, if we have

$$\left\| \sum_{i=1}^k (\widehat{V}_1 X_1')_i \otimes (\widehat{V}_2 X_2')_i \otimes (\widehat{V}_3 X_3')_i - C \right\|_F^2 \le \alpha \left\| \sum_{i=1}^k (\widehat{V}_1 X_1)_i \otimes (\widehat{V}_2 X_2)_i \otimes (\widehat{V}_3 X_3)_i - C \right\|_F^2,$$

and we further define $f(X_1, X_3, X_3) = \|\sum_{i=1}^k (V_1 X_1)_i \otimes (V_2 X_2)_i \otimes (V_3 X_3)_i - A\|$ and $g(X_1, X_2, X_3) = \|\sum_{i=1}^k (\widehat{V}_1 X_1)_i \otimes (\widehat{V}_2 X_2)_i \otimes (\widehat{V}_3 X_3)_i - C\|$, by above derivations we could conclude

$$(1 - \epsilon)f(X_1, X_2, X_3) + (1 - \epsilon)\Delta \le g(X_1, X_2, X_3) \le (1 + \epsilon)f(X_1, X_2, X_3) + (1 + \epsilon)\Delta$$

by properly scaling $\epsilon$, then

$$(1 - \epsilon)f(X_1', X_2', X_3') + (1 - \epsilon)\Delta \leq g(X_1', X_2', X_3')$$
$$\leq \alpha \cdot g(X_1, X_2, X_3)$$
$$\leq \alpha \cdot ((1 + \epsilon)f(X_1, X_2, X_3) + (1 + \epsilon)\Delta),$$

thus

$$f(X_1', X_2', X_3') \leq \alpha \cdot (1 + \epsilon)f(X_1, X_2, X_3) + O(\epsilon) \cdot \mathrm{OPT}$$
$$= \alpha \cdot (1 + O(\epsilon)) \cdot \mathrm{OPT},$$

the proof is completed by recalling the definition of OPT and rescaling $\epsilon$.

**Running time.** Since all $V_1, V_2$ and $V_3$ have $n$ rows and $\mathrm{poly}(k/\epsilon)$ columns, computing the quantum leverage score sampler takes time $\widetilde{O}(n^{0.5}\mathrm{poly}(k/\epsilon))$. To compute the matrix $C$, we note that since $T_1, T_2$ and $T_3$ are sampling matrices, each of them has only $\mathrm{poly}(k/\epsilon)$ entries. On the other hand, by the definition of $A(T_1, T_2, T_3)$, we note an entry of $A$ needs to be examined and computed if and only if all corresponding entries of $T_1, T_2$ and $T_3$ are nonzero. As these three sampling matrices have at most $\mathrm{poly}(k/\epsilon)$ overlaps on nonzero entries, computing $A(T_1, T_2, T_3)$ amounts to select a subset of $\mathrm{poly}(k/\epsilon)$ entries from $A$ and rescale, hence could be done in $\mathrm{poly}(k/\epsilon)$ time. $\qquad\square$

### F.5 QUANTUM TENSOR COLUMN, ROW AND TUBE SUBSET SELECTION APPROXIMATION

In this section, we design a quantum algorithm for selecting a subset of columns, rows and tubes of a tensor so that there exists a tensor $U$ of rank-$\mathrm{poly}(k/\epsilon)$, together with these subsets, gives a good low-rank approximation to $A$.

---

**Algorithm 13** Quantum tensor CRT subset selection.

---

1: **procedure** QCRTSELECTION($A \in \mathbb{R}^{n \times n \times n}, k, \epsilon$)
2: $\quad s_1, s_2 \leftarrow \widetilde{O}(k/\epsilon^2)$
3: $\quad \epsilon_0 \leftarrow 0.001$
4: $\quad C_1 \leftarrow \text{QLOWRANKCMM}(A_1, k, \epsilon, 0.0001)$ $\qquad\qquad\qquad\qquad \triangleright C_1 \in \mathbb{R}^{n \times s_1}$.
5: $\quad C_2 \leftarrow \text{QLOWRANKCMM}(A_2, k, \epsilon, 0.0001)$ $\qquad\qquad\qquad\qquad \triangleright C_2 \in \mathbb{R}^{n \times s_2}$.
6: $\quad$ Form $B_1$ by consecutively repeating each column of $C_1$ by $s_2$ times $\quad \triangleright B_1 \in \mathbb{R}^{n \times s_1 s_2}$.
7: $\quad$ Form $B_2$ by consecutively repeating each column of $C_2$ by $s_1$ times $\quad \triangleright B_2 \in \mathbb{R}^{n \times s_1 s_2}$.
8: $\quad d_3 \leftarrow O(s_1 s_2 \log(s_1 s_2) + s_1 s_2/\epsilon)$
9: $\quad D_3 \leftarrow \text{TENSORLEVERAGESCORE}(B_1^\top, B_2^\top, n, n, s_1 s_2, \epsilon_0, d_3)$ $\qquad \triangleright D_3 \in \mathbb{R}^{n^2 \times d_3}$.
10: $\quad M_3 \leftarrow A_3 D_3$ $\qquad\qquad\qquad\qquad\qquad\qquad\qquad\qquad\qquad \triangleright M_3 \in \mathbb{R}^{n \times d_3}$.
11: $\quad$ Form $B_1$ by consecutively repeating each column of $C_1$ by $d_3$ times $\triangleright$ Note $B_1$ is formed by repeating a different number of columns.
12: $\quad$ Form $B_3$ by consecutively repeating each column of $M_3$ by $s_1$ times
13: $\quad d_2 \leftarrow O(s_1 d_3 \log(s_1 d_3) + s_1 d_3/\epsilon)$
14: $\quad D_2 \leftarrow \text{TENSORLEVERAGESCORE}(B_1^\top, B_3^\top, n, n, s_1 d_3, \epsilon_0, d_2)$ $\qquad \triangleright D_2 \in \mathbb{R}^{n^2 \times d_2}$.
15: $\quad M_2 \leftarrow A_2 D_2$ $\qquad\qquad\qquad\qquad\qquad\qquad\qquad\qquad\qquad \triangleright M_2 \in \mathbb{R}^{n \times d_2}$.
16: $\quad$ Form $B_2$ by consecutively repeating each column of $M_2$ by $d_3$ times
17: $\quad$ Form $B_3$ by consecutively repeating each column of $M_3$ by $d_2$ times
18: $\quad d_1 \leftarrow O(d_2 d_3 \log(d_2 d_3) + d_2 d_3/\epsilon)$
19: $\quad D_3 \leftarrow \text{TENSORLEVERAGESCORE}(B_2^\top, B_3^\top, n, n, d_2 d_3, \epsilon_0, d_1)$
20: $\quad C \leftarrow A_1 D_1, R \leftarrow A_2 D_2, T \leftarrow A_3 D_3$
21: $\quad$ **return** $C, R, T$
22: **end procedure**

---

**Theorem F.9.** *Given a 3rd order tensor $A \in \mathbb{R}^{n \times n \times n}$ and a positive integer $k \leq n$, $\epsilon \in (0, 0.1)$, there exists an algorithm (Algorithm 13) which takes $\widetilde{O}(\epsilon^{-1}n^2 k^{0.5} + n\,\mathrm{poly}(k/\epsilon))$ time, $\widetilde{O}(\epsilon^{-1}nk^{0.5})$* *queries to the rows, columns and tubes of $A$,* *and outputs three matrices $C \in \mathbb{R}^{n \times c}$, a subset of columns of $A$; $R \in \mathbb{R}^{n \times r}$, a subset of rows of $A$; and $T \in \mathbb{R}^{n \times t}$, a subset of tubes of $A$ where $c, r, t = \mathrm{poly}(k/\epsilon)$, and there exists a tensor $U \in \mathbb{R}^{c \times r \times t}$ such that*

$$\|\sum_{i=1}^c \sum_{j=1}^r \sum_{l=1}^t U_{i,j,l} \cdot C_i \otimes R_j \otimes T_l - A\|_F^2 \leq (4 + \epsilon) \cdot \min_{\text{rank-}k\ A_k} \|A - A_k\|_F^2$$

*holds with probability 0.99.*

*Proof.* Throughout the proof, let $\text{OPT} := \min_{\text{rank-}k \ A_k} \|A - A_k\|_F^2$. Suppose the optimal low-rank factor $A_k = U^* \otimes V^* \otimes W^*$ where $U^*, V^*, W^* \in \mathbb{R}^{n \times k}$. Define a matrix $Z_1 \in \mathbb{R}^{k \times n^2}$, where the $i$-th row of $Z_1$ is $V_i^* \otimes W_i^*$. Note that we do not know $V^*$ and $W^*$, nor can we form the matrix $Z_1$. Consider the following regression problem:

$$\min_{U \in \mathbb{R}^{n \times k}} \|U Z_1 - A_1\|_F^2, \tag{8}$$

clearly, if we set $U$ as $U^*$, then

$$U^* Z_1 = [U_1^* \quad U_2^* \quad \ldots \quad U_k^*] \begin{bmatrix} \text{vec}(V_1^* \otimes W_1^*)^\top \\ \text{vec}(V_2^* \otimes W_2^*)^\top \\ \vdots \\ \text{vec}(V_k^* \otimes W_k^*)^\top \end{bmatrix}$$

$$= (U^* \otimes V^* \otimes W^*)_1,$$

i.e., the optimal $A_k$ flattens along the first dimension. Hence, the optimal cost of Eq. (8) would give OPT. To solve Eq. (8), we compute a projection-cost preserving of $A_1$ (Theorem C.2), and according to Lemma F.3, there exists a solution $\widehat{U}$ in the column span of $C_1$, i.e., $\widehat{U} = C_1 X_1$, and it has cost

$$\|\widehat{U} Z_1 - A_1\|_F^2 \leq (2 + \epsilon) \cdot \text{OPT}.$$

We can then form $Z_2 \in \mathbb{R}^{k \times n^2}$ where the $i$-th row is $\widehat{U}_i \otimes W_i^*$, then we know that

$$\min_{V \in \mathbb{R}^{n \times k}} \|V Z_2 - A_2\|_F^2 \tag{9}$$

is at most $(2 + \epsilon) \cdot \text{OPT}$ as we could choose $V$ as $V^*$. We approximately solve the regression of Eq. (9) against $C_2$, and again by Lemma F.3, we know that there exists a solution $\widehat{V} = C_2 X_2$ such that

$$\|\widehat{V} Z_2 - A_2\|_F^2 \leq (2 + \epsilon) \cdot \text{OPT}$$
$$\leq (2 + \epsilon)^2 \cdot \text{OPT}.$$

We then define $Z_3 \in \mathbb{R}^{k \times n^2}$ where the $i$-th row is $\widehat{U}_i \otimes \widehat{V}_i$, note that $Z_3$ is no longer intractable to us, because we know $\widehat{U}$ and $\widehat{V}$ are in the column span of $C_1, C_2$ respectively. Define $Z_3' \in \mathbb{R}^{d_3 \times n^2}$ such that, if we index the row of $Z_3'$ as $(i, j)$, then $(Z_3')_{(i,j)}$ is $(C_1)_i \otimes (C_2)_j$. Note that $Z_3'$ let us to express the column span of $C_1$ and $C_2$, consequently there exists some $X$ such that $Z_3 = X Z_3'$ (note that due to the property of $\otimes$, column span of $C_1$ and $C_2$ are formed by multiplying on the left instead of on the right). Consequently, consider the following optimization problem

$$\min_{W \in \mathbb{R}^{n \times k}, X \in \mathbb{R}^{k \times d_3}} \|W X Z_3' - A_3\|_F^2, \tag{10}$$

as one could set $X$ so that $Z_3 = X Z_3'$, we have the cost of Eq. (10) is at most $(2 + \epsilon)^2 \cdot \text{OPT}$. Computing the leverage score sampling of $Z_3'$ using TENSORLEVERAGESCORE and by Lemma A.13, we have that if we solve the following regression

$$\min_{Y \in \mathbb{R}^{n \times d_3}} \|Y Z_3' D_3 - A_3 D_3\|_F^2,$$

with optimal being $Y' = A_3 D_3 (Z_3' D_3)^\dagger$, then

$$\|A_3 D_3 (Z_3' D_3)^\dagger Z_3' - A_3\|_F^2 \leq (1 + \epsilon) \cdot \min_{Y \in \mathbb{R}^{n \times z_3}} \|Y Z_3' - A_3\|_F^2$$
$$\leq (1 + \epsilon)(2 + \epsilon)^2 \cdot \text{OPT},$$

this suggests we could consider the regression

$$\min_{X \in k \times d_3} \|A_3 D_3 X Z_3' - A_3\|_F^2 \tag{11}$$

as $X = (Z_3'D_3)^\dagger$ is a good solution. Letting $W' := A_3D_3 \in \mathbb{R}^{n \times d_3}$, define $Z_2' \in \mathbb{R}^{d_2 \times n^2}$ with $\widehat{U}$ and $W'$ such that $(Z_2')_{(i,j)} = (C_1)_i \otimes (W')_j$, note that $Z_2'$ contains the column span of $C_1$ and $W'$, and although $Z_2$ is not in the row span of $Z_2'$ as in the case of $Z_3$, the $W'$ component of $Z_2'$ gives good approximation to $W^*$ as we have shown above. Hence, if we consider

$$\min_{V \in \mathbb{R}^{n \times k}, X \in \mathbb{R}^{k \times d_2}} \|VXZ_2' - A_2\|_F^2,$$

its cost is upper bounded by Eq. (11) as we could choose $V$ as $\widehat{V}$ and flatten $A$ alongside the third direction to recover the same regression. Employing a similar argument, if we sample according to the leverage score $Z_2'$ and consider

$$\min_{Y \in \mathbb{R}^{n \times d_2}} \|YZ_2'D_2 - A_2D_2\|_F^2,$$

the optimal solution is in the column span of $A_2D_2$ and it blows up the cost by a factor at most $1 + \epsilon$, which gives us the following:

$$\min_{X \in \mathbb{R}^{k \times d_2}} \|A_2D_2XZ_2' - A_2\|_F^2, \tag{12}$$

and the cost of Eq. (12) is at most $(1+\epsilon)^2(2+\epsilon)^2 \cdot \text{OPT}$. Set $V' := A_2D_2$ and repeat the construction of $Z_1'$ with $V', W'$, then we end up with

$$\min_{X \in \mathbb{R}^{k \times d_1}} \|A_1D_1XZ_1' - A_1\|_F^2 \tag{13}$$

whose cost is at most $(1 + \epsilon)^3(2 + \epsilon)^2 \cdot \text{OPT} = (4 + O(\epsilon)) \cdot \text{OPT}$ after properly scaling $\epsilon$. Setting $U' := A_1D_1$, and unwrap $Z_1'$, we see Eq. (13) in fact gives our desired result, as $U', V', W'$ are weighted subset of columns, rows and tubes of $A$, we could craft the desired $C, R, T$ by removing the weights, and completing $U$ by solving the regression Eq. (13), incorporating the solution to the weights. Since our statement only states the existence of such $U$, we do not consider the problem of finding it.

We complete the proof by analyzing its runtime. The most time consuming step is to compute $C_1$ and $C_2$, since we are sampling columns as in the case of Theorem C.2, the runtime of these steps is $\widetilde{O}(\epsilon^{-1}n^2k^{0.5} + n\,\text{poly}(k/\epsilon))$, and it is not hard to see that all subsequent steps take $O(n\,\text{poly}(k/\epsilon))$ time as we either perform operations that run in nearly linear time in $n$ on matrices of size $n \times \text{poly}(k/\epsilon)$, or we select $\text{poly}(k/\epsilon)$ columns from an $n \times n^2$ matrix. $\square$

Note that Theorem F.9 only gives a column, row and tube subset selection, but not with the weights tensor $U$. To output the tensor $U$, we first provide quantum bicriteria tensor low-rank approximation algorithm.

### F.6 TENSOR CURT DECOMPOSITION: FIXED-PARAMETER TRACTABLE AND BICRITERIA

**Theorem F.10** (A modification of Theorem C.40 in Song et al. (2019)). *Given a 3rd order tensor $A \in \mathbb{R}^{n \times n \times n}$, let $k \geq 1$, and let $U_B, V_B, W_B \in \mathbb{R}^{n \times k}$ denote a rank-$k$, $\alpha$-approximation to $A$. Then there is a classical algorithm (Algorithm 14) which takes $O(n\,\text{poly}(k/\epsilon))$ time and outputs three matrices $C \in \mathbb{R}^{n \times c}$ with columns from $A$, $R \in \mathbb{R}^{n \times r}$ with rows from $A$, $T \in \mathbb{R}^{n \times t}$ with tubes from $A$, and a tensor $U \in \mathbb{R}^{c \times r \times t}$ with $\text{rank}(U) = k$ such that $c = r = t = O(k \log k + k/\epsilon)$, and*

$$\left\| \sum_{i=1}^{c} \sum_{j=1}^{r} \sum_{l=1}^{t} U_{i,j,l} \cdot C_i \otimes R_j \otimes T_l - A \right\|_F^2 \leq (1+\epsilon)\alpha \min_{\text{rank}-k\ A'} \|A' - A\|_F^2$$

*holds with probability $9/10$.*

**Theorem F.11** (Bicriteria Tensor CURT Decomposition). *Given a 3rd order tensor $A \in \mathbb{R}^{n \times n \times n}$ and a positive integer $k \leq n$, $\epsilon \in (0, 0.1)$, there exists an algorithm which takes $\widetilde{O}(\epsilon^{-1}n^2k^{0.5} + n\,\text{poly}(k/\epsilon))$ time, $\widetilde{O}(\epsilon^{-1}nk^{0.5})$ queries to the rows, columns and tubes of $A$ and outputs three matrices $C, R, T \in \mathbb{R}^{n \times r}$ with $r = \widetilde{O}(k^2/\epsilon^4)$ and $U \in \mathbb{R}^{r \times r \times r}$ such that*

$$\left\| \sum_{i=1}^{c} \sum_{j=1}^{r} \sum_{l=1}^{t} U_{i,j,l} \cdot C_i \otimes R_j \otimes T_l - A \right\|_F^2 \leq (4+\epsilon) \cdot \min_{\text{rank}-k\ A_k} \|A - A_k\|_F^2$$

*with probability $0.99$.*

---

**Algorithm 14** Converting a tensor low-rank approximation to a CURT decomposition.

---

1: **procedure** FROMLOWRANKTOCURT$(A, U_B, V_B, W_B, n, k, \epsilon)$  $\quad\quad\quad\triangleright$ Lemma F.10
2: $\quad\quad d_1 \leftarrow d_2 \leftarrow d_3 \leftarrow O(k \log k + k/\epsilon)$.
3: $\quad\quad \epsilon_0 \leftarrow 0.01$.
4: $\quad\quad$ Form $B_1 = V_B^\top \odot W_B^\top \in \mathbb{R}^{k \times n^2}$
5: $\quad\quad D_1 \leftarrow$TENSORLEVERAGESCORE$(V_B^\top, W_B^\top, n, n, k, \epsilon_0, d_1)$
6: $\quad\quad$ Form $\widehat{U} = A_1 D_1 (B_1 D_1)^\dagger \in \mathbb{R}^{n \times k}$.
7: $\quad\quad$ Form $B_2 = \widehat{U}^\top \odot W_B^\top \in \mathbb{R}^{k \times n^2}$
8: $\quad\quad D_2 \leftarrow$TENSORLEVERAGESCORE$(\widehat{U}^\top, W_B^\top, n, n, k, \epsilon_0, d_2)$.
9: $\quad\quad$ Form $\widehat{V} = A_2 D_2 (B_2 D_2)^\dagger \in \mathbb{R}^{n \times k}$
10: $\quad\quad$ Form $B_3 = \widehat{U}^\top \odot \widehat{V}^\top \in \mathbb{R}^{k \times n^2}$
11: $\quad\quad D_3 \leftarrow$TENSORLEVERAGESCORE$(\widehat{U}^\top, \widehat{V}^\top, n, n, k, \epsilon_0, d_3)$
12: $\quad\quad C \leftarrow A_1 D_1, R \leftarrow A_2 D_2, T \leftarrow A_3 D_3$
13: $\quad\quad U \leftarrow \sum_{i=1}^k ((B_1 D_1)^\dagger)_i \otimes ((B_2 D_2)^\dagger)_i \otimes ((B_3 D_3)^\dagger)_i$
14: $\quad\quad$ **return** $C, R, T$ and $U$
15: **end procedure**

---

*Proof.* It directly follows from combining Theorem F.5 and Lemma F.10. $\qquad\qquad\square$

**Theorem F.12** (Fixed-Parameter Tractable Tensor CURT Decomposition). *Given a tensor $A \in \mathbb{R}^{n \times n \times n}$, we could obtain a tensor CURT decomposition with the guarantee of Theorem F.6, in time $\widetilde{O}(\epsilon^{-1} n^2 k^{0.5} + n \operatorname{poly}(k/\epsilon) + 2^{O(k^2/\epsilon)}) n^\delta$ and $\widetilde{O}(\epsilon^{-1} n k^{0.5})$ queries to the rows, columns and tubes of $A$.*

# G  IMPROVED QUANTUM CORESET ALGORITHM FOR $(k, p)$-CLUSTERING AND APPLICATION

In this section, we give an improved quantum coreset construction for $(k, p)$-clustering. We observe that the coreset obtained in prior work (1) The size scales linearly with $d$, this causes an additional $d^{0.5}$ factor in their final runtime; (2) The coreset consists of points not from $A$ and the weights for these points could be negative, therefore it might pose challenges if one wants to compose it with algorithm that induces optimal-sized coreset.

We begin by recalling the $(k, p)$-clustering problem in $\mathbb{R}^d$: let $A = \{a_1, \ldots, a_n\} \subset \mathbb{R}^d$, $X = (\mathbb{R}^d)^k$ and $\operatorname{cost}(a_i, x) = \min_{j \in [k]} \|a_i - x_j\|_2^p$, where $p \geq 1$ is the power of the distance, and $x_j$ is one of the centers in $x$. When $p = 1$, this is the well-studied $k$-median problem, and when $p = 2$, this captures the $k$-means problem. To construct a coreset, a popular approach is through sensitivity sampling. Here, we demonstrate how to implement the sensitivity sampling framework in quantum sublinear time.

We need the following quantum algorithm, due to Xue et al. (2023), that computes a set of $(\alpha, \beta)$-bicriteria approximation.

**Definition G.1** (Bicriteria Approximation). *Let $A \subset \mathbb{R}^d$, assume* OPT *is the optimal cost of the $(k, p)$-clustering problem for $A$, we say a set $x \subset \mathbb{R}^d$ is an $(\alpha, \beta)$-bicriteria approximation if $|x| \leq \alpha k$ and $\operatorname{cost}(A, x) \leq \beta$ OPT.*

**Lemma G.2** (Lemma 3.7 of Xue et al. (2023)). *Let $A \subset \mathbb{R}^d$, there exists a quantum algorithm that outputs an $(O(\log^2 n), 2^{O(p)})$-bicriteria approximation $x$, to the $(k, p)$-clustering problem for $A$, with probability at least $99/100$. The algorithm uses $\widetilde{O}(\sqrt{nk})$ queries to $A$, $\widetilde{O}(\sqrt{nk}d)$ time and $\operatorname{poly}(k \log n)$ additional preprocessing time.*

We also need a quantum approximate nearest neighbor oracle, which would be crucial to approximately find the center a point belongs to.

**Lemma G.3** (Lemma 3.4 of Xue et al. (2023)). *Let $A \subset \mathbb{R}^d$ and $x \subset \mathbb{R}^d$ wth $|x| = m$, given two parameters $\delta > 0$, $c_\tau \in [2.5, 3)$, there exists a quantum oracle that give $a_i \in A$, returns $\tau(a_i) \in x$,*

*using* $\text{poly}(m \log(n/\delta))$ *preprocessing time. With probability at least* $1 - \delta$, $\tau : A \to x$ *is a mapping such that*

$$\|a_i - \tau(a_i)\|_2^p \leq c_\tau \cdot \text{cost}(a_i, x).$$

*Each query to* $\tau$ *takes* $O(d \text{ poly} \log(mn/\delta))$ *time.*

Note that $\tau$ is also a *partition oracle*, as we could assign $a_i$ to $\tau(a_i)$, which is one of the $m$ clusters.

We need two other ingredients: one being estimate $\text{cost}(A, x) = \sum_{i=1}^n \text{cost}(a_i, x)$ and the other being estimating the number of points falls in each cluster.

**Lemma G.4** (Lemma 6 of Li et al. (2019)). *Let* $C = \{c_1, \dots, c_n\}$ *be a collection of nonnegative numbers, let* $c = \sum_{i=1}^n c_i$, *there exists a quantum algorithm such that given* $\delta > 0$, *it outputs an approximation* $\widetilde{c}$ *where* $\widetilde{c} = (1 \pm \epsilon) \cdot c$ *with probability at least* $1 - \delta$, *using* $\widetilde{O}(\sqrt{n} \log(1/\delta)/\epsilon)$ *queries to* $C$.

**Lemma G.5** (Theorem 4.4 of Xue et al. (2023)). *Let* $A \in (\mathbb{R}^d)^n, x \in (\mathbb{R}^d)^m$ *and* $\tau : A \to x$, *let* $C_j = \{a \in A : \tau(a) = x_j\}$, *let* $\epsilon \in (0, 1/3), \delta > 0$, *then there exists a quantum algorithm that with probability at least* $1 - \delta$, *outputs a list of estimates* $\widetilde{n}_j$ *for all* $j \in [m]$ *where* $\widetilde{n}_j = (1 \pm \epsilon) \cdot |C_j|$, *using* $\widetilde{O}(\sqrt{nm/\epsilon} \log(1/\delta))$ *queries to* $\tau$ *and an additional* $\widetilde{O}((\sqrt{nm/\epsilon} + m/\epsilon) \log(n/\delta))$ *time.*

The algorithm we will be using is based on Huang & Vishnoi (2020), in particular, we use the first stage of their algorithm, as it has two main advantages: (1) It computes a coreset with points only from $A$; (2) The weights are relatively easy to compute. After computing the coreset, we can compose it with the optimal-sized coreset construction algorithm to obtain the final result (Huang et al., 2024).

---

**Algorithm 15** Quantum coreset algorithm for $(k, p)$-clustering: no dependence on $d$ (Huang & Vishnoi, 2020).

---

1: **procedure** QCLUSTER($A \in \mathbb{R}^{n \times d}, \epsilon \in (0, 1)$)
2:      $m \leftarrow O(k \log^2 n)$
3:      $s \leftarrow O((168p)^{10p} \epsilon^{-5p-15} k^5 \log k)$
4:      $\epsilon' \leftarrow 0.01$
5:      Generate $x' \in (\mathbb{R}^d)^m$ via Lemma G.2
6:      Generate $\tau$ on $A, x'$ via Lemma G.3
7:      Let $C_j = \{a \in A : \tau(a) = x_j'\}$ and $n_j = |C_j|$
8:      Generate $\widetilde{n}_1, \dots \widetilde{n}_m$ via Lemma G.5 using $\tau$ with accuracy $\epsilon'$
9:      Generate $\widetilde{\text{cost}}(A, x')$ via Lemma G.4 with accuracy $\epsilon'$
10:      Implement an oracle for any $a_i \in A$ as follows
11:          $x^*(a_i) \leftarrow \tau(a_i)$
12:          $\widetilde{s}_i \leftarrow 2^{4p+2} \cdot \left( \frac{\|a_i - x^*(a_i)\|_2^p}{\widetilde{\text{cost}}(A, x')} + \frac{1}{\widetilde{n}_{i(j)}} \right)$     ▷ Let $i(j)$ denote the index of $x^*(a_i)$ among $x'$
13:          $p_i \leftarrow \min\{1, \widetilde{s}_i\}$
14:      $D \leftarrow$ QSAMPLE($p$)                                 ▷ $\|D\|_0 = s$
15: **end procedure**

---

**Lemma G.6** (Theorem 5.2 of Huang & Vishnoi (2020)). *Let* $A = \{a_1, \dots, a_n\} \subset \mathbb{R}^d, X = (\mathbb{R}^d)^k$, *and define* $\text{cost} : \mathbb{R}^d \times X \to \mathbb{R}_{\geq 0}$ *as* $\text{cost}(a_i, x) = \min_{j \in [k]} \|a_i - x_j\|_2^p$. *Given* $\epsilon, \delta \in (0, 1), p \geq 1$, *suppose quantities in Algorithm 15 are computed exactly except for the bicriteria approximation, then the weights in* $D$ *give rise to an* $\epsilon$-coreset of size $s = \widetilde{O}_p(\epsilon^{-5p-15} k^5)$.

While the quantities in Algorithm 15 are computed approximately, they are all two-sided constant factor approximation, therefore we still get desired guarantees. We present the main result in the following.

**Theorem G.7.** *Let* $A = \{a_1, \dots, a_n\} \subset \mathbb{R}^d, X = (\mathbb{R}^d)^k, p \geq 1, \epsilon \in (0, 1)$, *define* $\text{cost}(a_i, x) = \min_{j \in [k]} \|a_i - x_j\|_2^p$. *There exists a quantum algorithm (Algorithm 15) such that, with probability at least 0.99, constructs an* $\epsilon$-coreset of $A$ with size $\widetilde{O}_p(\epsilon^{-5p-15} k^5)$ *in time* $\widetilde{O}_p(\epsilon^{-2.5p-7.5} n^{0.5} k^{2.5} d)$ *and* $\widetilde{O}_p(\epsilon^{-2.5p-7.5} n^{0.5} k^{2.5})$ *queries to the points in* $A$.

*Proof.* We first prove that it indeed constructs a coreset. There are three main differences between Algorithm 15 and stage 1 of Huang & Vishnoi (2020):

- We use bicriteria approximation while Huang & Vishnoi (2020) computes $k$-approximate centers;

- We have to use approximate nearest neighbor to find the approximate center for each $a_i$;

- We approximately compute $\text{cost}(A, x')$ and $\frac{1}{|C_i|}$.

For the first and second item, one could easily see that Lemma 5.5 and Claim 5.6 of Huang & Vishnoi (2020) do not require exactly $k$-approximate centers, as they only need to use the cost of these approximate centers as a proxy, hence, an $(\alpha, \beta)$-bicriteria approximation is sufficient. Moreover, their proof relies on a simple generalized triangle inequality argument, so as long as the approximate cluster we assign $a_i$ is a constant factor approximation to the optimal distance, we are fine. For the third item, note that by Lemma G.4, we have $\widetilde{\text{cost}}(A, x') = (1 \pm \epsilon') \cdot \text{cost}(A, x')$ and by Lemma G.5, $\widetilde{n}_{i(j)} = (1 \pm \epsilon') \cdot |C_{i(j)}|$, therefore the sampling probability $\widetilde{s}_i$ is a constant factor approximation if we set to the approximate sensitivity $\sigma_1$ used in Huang & Vishnoi (2020). Thus, if we oversample by a constant factor, we could indeed get the desired coreset property according to Lemma G.6. It remains to analyze the runtime.

To generate $x'$, by Lemma G.2, it takes $\widetilde{O}(\sqrt{nk}d)$ time, and oracle $\tau$ takes $\text{poly}(k)$ time to preprocess, and each oracle call to $\tau$ takes $\widetilde{O}(d)$ time due to Lemma G.3. Generate the estimates $\widetilde{n}_j$ for all $j \in [m]$ takes $\widetilde{O}(\sqrt{nm}d) = \widetilde{O}(\sqrt{nk}d)$ time, and $\widetilde{\text{cost}}(A, x')$ takes $\widetilde{O}(\sqrt{n}d)$ time owing to Lemma G.4. Finally, note that each $\widetilde{s}_i$ can be computed in $\widetilde{O}(d)$ time, by Lemma A.14, the sample and weights $D$ can be generated in $\widetilde{O}(\sqrt{ns}d) = \widetilde{O}_p(\epsilon^{-2.5p-7.5}n^{0.5}k^{2.5}d)$ time, as desired. □

**Remark G.8.** *While the coreset size of Theorem G.7 is not optimal, it produces coreset of size $\widetilde{O}_p(\epsilon^{-5p-15}k^5)$. This is sufficient as we could run any refinement to obtain the optimal size, as demonstrated by composing our coreset with the following result due to Huang et al. (2024).*

**Lemma G.9** (Theorem B.1 of Huang et al. (2024))**.** *Let $A = \{a_1, \ldots, a_n\} \subset \mathbb{R}^d$ and $X = (\mathbb{R}^d)^k$, $p \geq 1$, $\epsilon, \delta \in (0, 1)$, and define $\text{cost}(a_i, x) = \min_{j \in [k]} \|a_i - x_j\|_2^p$. There exists a randomized algorithm that with probability at least $1 - \delta$ constructs an $\epsilon$-strong coreset of size $\widetilde{O}_p(\epsilon^{-2}k^{\frac{2p+2}{p+2}})$, in time $\widetilde{O}(ndk)$.*

**Corollary G.10.** *Let $A = \{a_1, \ldots, a_n\} \subset \mathbb{R}^d$ and $X = (\mathbb{R}^d)^k$, $p \geq 1$, $\epsilon, \delta \in (0, 1)$, and define $\text{cost}(a_i, x) = \min_{j \in [k]} \|a_i - x_j\|_2^p$. There exists a quantum algorithm that with probability at least 0.99 constructs: an $\epsilon$-coreset of size $\widetilde{O}_p(\epsilon^{-2}k^{\frac{2p+2}{p+2}})$, in time*

$$\widetilde{O}_p(\epsilon^{-2.5p-7.5}n^{0.5}k^{2.5}d).$$

*Proof.* The proof is by composing Theorem G.7 with Lemma G.9. □

### G.1 QUANTUM ALGORITHM FOR DATA SELECTION

As an application, we study the *data selection pipeline* in machine learning, where the goal is to select a weighted subset of data points that can be used for training or fine-tuning the model, while preserving desirable properties. In this model, data are given as $d$-dimensional embeddings, and a loss function $\ell : \mathbb{R}^d \to \mathbb{R}_{\geq 0}$ is used to grade the quality of the embedding. $\ell$ can be expensive to evaluate, such as a deep neural network. Axiotis et al. (2024) provides a principled way for data selection using the coreset of $(k, p)$-clustering, under some mild assumptions on $\ell$.

**Assumption G.11.** *Let $\Lambda = (\Lambda_1, \ldots, \Lambda_k) \in \mathbb{R}_{\geq 0}^k$, $x \in (\mathbb{R}^d)^k$ and let $E \subseteq \mathbb{R}^d$ be a set of embeddings, we say the loss function is $(p, \Lambda)$-well-behaved with respect to $E$ and $x$ if for any $x_j \in x$ and let $C_j = \{e \in E : \arg\min_{x_i \in x} \|x_i - e\|_2^p = x_j\}$, then for any $e \in C_j$,*

$$|\ell(e) - \ell(x_j)| \leq \Lambda_j \|e - c_j\|_2^p.$$

Define the weighed cost as $\text{cost}^\Lambda(a_i, x) = \Lambda_{i(j)} \text{cost}(a_i, x)$ where we use $i(j)$ to denote the index of the cluster assigned to $a_i$, and similarly $\text{cost}^\Lambda(A, x) = \sum_{i=1}^n \text{cost}^\Lambda(a_i, x) = \sum_{i=1}^k \Lambda_i \sum_{a_j \in C_i} \|a_j - x_i\|_2^p$. Axiotis et al. (2024) essentially proves that under Assumption G.11,

one could perform weighted sampling according to $\text{cost}^\Lambda(a_i, x)$. In addition, the expensive loss function only needs to be evaluated on the centers. For convenience, we state an approximate $k$-centers algorithm below.

**Lemma G.12** (Mettu & Plaxton (2004)). *Let $A = \{a_1, \dots, a_n\} \subset \mathbb{R}^d$ and $X = (\mathbb{R}^d)^k$, let $\delta \in (0, 1)$. Then, there exists an algorithm that computes a solution $x' \in X$ such that*

$$\text{cost}(A, x') \leq 2^{O(p)} \cdot \min_{x \in X} \text{cost}(A, x),$$

*holds with probability at least $1 - \delta$. Moreover, $x'$ can be computed in time*

$$O(ndk + nd\log(n/\delta) + k^2 \log^2 n + \log^2(1/\delta)\log^2 n) = \widetilde{O}(ndk).$$

We know state a quantum implementaion of the adaptive sampling due to Axiotis et al. (2024).

---

**Algorithm 16** Quantum one-round adaptive sampling for data selection.

---

1: **procedure** QDATASELECTION($A \in \mathbb{R}^{n \times d}, x \in (\mathbb{R}^d)^k, \ell : \mathbb{R}^d \to \mathbb{R}_{\geq 0}, \epsilon \in (0, 1)$)
2:     $s \leftarrow O(\epsilon^{-2}), \epsilon' \leftarrow 0.01$
3:     Let $\tau : A \to x$ be that $\tau(a_i) = \arg\min_{x_j \in x} \|a_i - x_j\|_2^p$
4:     Generate $\widetilde{\text{cost}}^\Lambda(A, x')$ via Lemma G.4 with accuracy $\epsilon'$
5:     Let $C_j = \{a \in A : \tau(a) = x_j\}$ and $n_j = |C_j|$
6:     Generate $\widetilde{n}_1, \dots, \widetilde{n}_k$ via Lemma G.5 using $\tau$ with accuracy $\epsilon'$
7:     Compute $\ell(x_1), \dots, \ell(x_k)$
8:     $\text{sum} \leftarrow \sum_{j=1}^k \widetilde{n}_j \cdot \ell(x_j)$
9:     Implement an oracle for each $a_i \in A$ as follows:
10:         $\widehat{\ell}(a_i) \leftarrow \ell(\tau(a_i)), v(a_i) \leftarrow \|a_i - \tau(a_i)\|_2^p$
11:         $q_i \leftarrow \frac{\widehat{\ell}(a_i) + v(a_i)}{\widetilde{\text{cost}}^\Lambda(A, x) + \text{sum}}$
12:         $p_i \leftarrow \min\{1, q_i\}$
13:     $D' \leftarrow \text{QSAMPLE}(p)$
14:     **return** $D'$
15: **end procedure**

---

We then prove Algorithm 16 implements the data selection procedure in sublinear time.

**Theorem G.13.** *Let $\epsilon \in (0, 1), p \geq 1, \Lambda \in \mathbb{R}^k, A \in (\mathbb{R}^d)^n$ and $\ell$ be a loss function that is $(p, \Lambda)$-well-behaved with respect to $A$ and a clustering $x \in (\mathbb{R}^d)^k$. Then, there exists a quantum algorithm (Algorithm 16) that outputs a weight vector $w \in \mathbb{R}_{\geq 0}^n$ with $\|w\|_0 = O(\epsilon^{-2})$, such that*

$$|\sum_{i=1}^n \ell(a_i) - \sum_{i=1}^n w_i \ell(a_i)| \leq \epsilon \cdot (\sum_{i=1}^n \ell(a_i) + 2\,\text{cost}^\Lambda(A, x))$$

*holds with probability at least 0.99. Moreover, the algorithm makes at most $k$ queries to the loss function $\ell$, $\widetilde{O}(\epsilon^{-1} n^{0.5} k^{0.5})$ queries to the points in $A$, and uses an additional $\widetilde{O}(n^{0.5}kd(\epsilon^{-1} + k^{0.5}))$ time.*

*Proof.* We first note that the only difference between Algorithm 16 and Theorem 2 of Axiotis et al. (2024) is that we approximately compute the quantity $\widetilde{\text{cost}}^\Lambda(A, x')$ and $\sum_{i=1}^n \widehat{\ell}(a_i)$, by a similar argument as Theorem G.7, these quantities are estimated within a constant factor, therefore the sampling probability $p_i$ is at most a constant factor of the sampling probability used in Axiotis et al. (2024), we can obtain the same guarantee via oversampling by a constant factor.

To analyze the runtime, note that the oracle $\tau$ can be queried in $O(kd)$ time, and $\widetilde{\text{cost}}^\Lambda(A, x)$ can be computed in $\widetilde{O}(\sqrt{n}kd)$ time by Lemma G.4. $\widetilde{n}_1, \dots, \widetilde{n}_k$ can be estimated in $\widetilde{O}(\sqrt{n}k^{1.5}d)$ time. Finally, each sampling probability can be computed in $O(kd)$ time, so the time for the final sampling is $\widetilde{O}(\epsilon^{-1} n^{0.5}kd)$ time. Thus, the overall runtime is

$$\widetilde{O}(n^{0.5}kd(\epsilon^{-1} + k^{0.5})). \qquad \square$$

Note that compute the weights classically would take $O(ndk)$ time, so ours is the first to achieve this goal in sublinear in $n$ time. To compute a set of approximate $k$-centers, one could either directly use the bicriteria approximation due to Lemma G.2 and use these centers as a proxy instead, or first compute an $\epsilon$-coreset using Theorem G.7 then apply Lemma G.12 to find the approximate $k$-centers using the coreset.

## H    LOWER BOUND

In this section, we provide a quantum query lower bound on computing a rank-$k$, $1/2$-additive-multiplicative spectral approximation to a matrix $A \in \mathbb{R}^{n \times d}$. We show that $\Omega(\sqrt{dk})$ queries to the columns of $A$ are needed.

**Theorem H.1.** *For any positive integers $n, d$, and $k \leq d$, there is a family of matrices $A \in \mathbb{R}^{n \times d}$ for which finding a constant factor additive-multiplicative spectral approximation of rank-$k$ requires $\Omega(\sqrt{dk})$ column queries to $A$.*

*Proof.* Without loss of generality let $k$ divide $d$, let $z_1, \ldots, z_k \in \{0, 1\}^{d/k}$ be a collection of bit strings, we construct $A$ similar to the construction of Apers & Gribling (2024) but padding extra 0's: we start a matrix $\overline{A} \in \mathbb{R}^{k \times d}$, consists of $k$ blocks of $k \times d/k$: for the $j$-th block, it contains $z_j$ as its $j$-th row, and zero elsewhere. We then pad $n - k$ rows of zeros to form the $n \times d$ matrix $A$, one could visualize $A$ as follows:

$$A = \begin{bmatrix} z_1^\top & 0 & \ldots & 0 \\ 0 & z_2^\top & \ldots & 0 \\ \vdots & \vdots & \vdots & \vdots \\ 0 & 0 & \ldots & z_k^\top \\ 0 & 0 & \ldots & 0 \\ \vdots & \vdots & \vdots & \vdots \\ 0 & 0 & \ldots & 0 \end{bmatrix}$$

where $0$ is the zero vector of size $d/k$. Note that $A$ is rank-$k$, hence $A_k = A$. Consequently,

$$AA^\top = \begin{bmatrix} \|z_1\|_0 & 0 & \ldots & 0 & \ldots & 0 \\ 0 & \|z_2\|_0 & \ldots & 0 & \ldots & 0 \\ \vdots & \vdots & \vdots & \vdots & \vdots & \vdots \\ 0 & 0 & \ldots & \|z_k\|_0 & \ldots & 0 \\ 0 & 0 & \ldots & 0 & \ldots & 0 \\ \vdots & \vdots & \vdots & \vdots & \vdots & \vdots \\ 0 & 0 & \ldots & 0 & \ldots & 0 \end{bmatrix}$$, i.e., its top-$k$ diagonal entries are the number of

nonzeros in each of $z_i$'s. Note that a rank-$k$ additive-multiplicative spectral approximation has the guarantee that

$$0.5 CC^\top - 0.5 \frac{\|A - A_k\|_F^2}{k} I_n \preceq AA^\top \preceq 1.5 CC^\top + 0.5 \frac{\|A - A_k\|_F^2}{k} I_n$$

since $A$ is rank-$k$, we have $\|A - A_k\|_F^2 = 0$ and therefore, the approximation $C$ has the property that

$$0.5 CC^\top \preceq AA^\top \preceq 1.5 CC^\top,$$

since $AA^\top$ is diagonal, we must have the nonzero diagonals of $CC^\top$ is a $0.5$-approximation to the nonzero diagonals of $AA^\top$. This allows us to compute $(\mathrm{OR}(z_1), \ldots, \mathrm{OR}(z_k))$ where $\mathrm{OR}(x) = x_1 \vee x_2 \vee \ldots \vee x_{d/k}$. By a similar argument as Apers & Gribling (2024), this would require $\Omega(k\sqrt{d/k}) = \Omega(\sqrt{dk})$ quantum queries to the bit strings $z_1, \ldots, z_k$. Finally, note that a column query to $A$ can be simulated by a query access to one of the $z_j$'s. This completes the proof. $\quad\square$

# I SIMULATING RANDOM ORACLE ACCESS

In this section, we discuss how to simulate query access to a uniformly random string without impairing the runtime of our algorithm. We observe that this is necessary, as our classical oracle is inherently randomized, converting it into a quantum circuit will produce a superposition of estimates, incurring additional approximation errors. We resolve this issue by "derandomize" the algorithm by giving it query access to a uniformly random string. Specifically, suppose an algorithm has runtime $q$, then we need to attach a random string of length at most $q$. As observed in Apers & Gribling (2024), if we only care about query complexity, then we could explicitly sample these bitstrings and use QRAM to access them. In addition, if we want to achieve a sublinear runtime, then explicitly drawing these bitstrings would be too slow. On the other hand, Beals et al. (2001) shows that any $(k/2)$-query quantum algorithm cannot distinguish a uniformly random from a $k$-wise independent string, hence it is enough to use a $k$-wise independent hash function to simulate query access to uniformly random strings. The following result is due to Apers & De Wolf (2022) and elaborated in Apers & Gribling (2024):

**Lemma I.1** (Lemma 3.12 in Apers & Gribling (2024)). *Consider any quantum algorithm with runtime $q$ that uses a uniformly random string, then we can simulate this algorithm with a quantum algorithm with runtime $\widetilde{O}(q)$ without random string, using an additional QRAM of $\widetilde{O}(q)$ bits.*

Hence, to implement our algorithm in QRAM, we apply I.1 to derandomize the procedure, this would allow us to use more standard approaches to deal with deterministic routines: we convert them to quantum oracles using quantum arithmetic gates and Toffoli gates, which is common in quantum algorithm literature.

# J QUERY COMPLEXITY OF OUR ALGORITHMS

In this section, we summarize the complexity of our algorithms in terms of *query complexity*, which is independent of the QRAM model: as noted in Apers & De Wolf (2022); Apers & Gribling (2024), we can remove the QRAM assumption at the cost of a polynomial increase in the *time complexity*.

| | Reference | Query Type | Complexity |
|---|---|---|---|
| $k$-Median Clustering | Theorem G.7 | Rows | $\epsilon^{-10}n^{0.5}k^{2.5}$ |
| $k$-Means Clustering | Theorem G.7 | Rows | $\epsilon^{-12.5}n^{0.5}k^{2.5}$ |
| $(k,p)$-Clustering | Theorem G.7 | Rows | $\epsilon^{-2.5p-7.5}n^{0.5}k^{2.5}$ |
| $\ell_{p\neq 2}$ Regression | Theorem B.18 | Rows | $\epsilon^{-1}n^{0.5}d^{0.5\vee p/4}$ |
| $(k,p\leq 2)$-Subspace Approx. | Theorem E.2 | Rows | $\epsilon^{-1}n^{1-p/4}k^{p/4}$ |
| $(k,p\geq 2)$-Subspace Approx. | Theorem E.2 | Rows | $\epsilon^{-p/2}n^{1-1/p}k^{0.5}$ |
| Low-Rank | Theorem C.2 | Columns | $\epsilon^{-1}n^{0.5}k^{0.5}$ |
| PSD Low-Rank | Theorem D.1 | Entries | $\epsilon^{-1.25}n^{0.75}k^{1.5}$ |
| Kernel Low-Rank | Theorem D.1 | Kernel Evals | $\epsilon^{-1.25}n^{0.75}k^{1.5}$ |
| Tensor Low-Rank: Rank-$k$ | Theorem F.6 | Rows, Columns and Tubes | $\epsilon^{-1}nk^{0.5}$ |
| Tensor Low-Rank: Bicriteria | Theorem F.5 | Rows, Columns and Tubes | $\epsilon^{-1}nk^{0.5}$ |

Table 3: Query complexity of our algorithms. We specify the types of query to measure the complexity for various problems, for most of the problems, the input is $A \in \mathbb{R}^{n\times d}$ that consists of $n$ points in $d$-dimensional Euclidean space, hence the row query corresponds to querying the points. For low-rank approximation, the input is $A \in \mathbb{R}^{d\times n}$ and the query corresponds to column queries. For PSD and kernel low-rank approximation, the query corresponds to evaluating the kernel function over two points, where in the PSD low-rank approximation, this is equivalent to querying the entries of $A$. Finally, for tensor low-rank approximation, we allow queries to the rows, columns and tubes of $A$, which would require at most a constant factor more qubits to store in the QRAM.

## LLM USAGE DISCLOSURE

LLMs were used only to polish language, such as grammar and wording. These models did not contribute to idea creation or writing, and the authors take full responsibility for this paper's content.

