# OpenReview forum: "Sublinear Time Quantum Sensitivity Sampling"
_ICLR.cc/2026/Conference — Submitted to ICLR 2026_

### Official Review · Reviewer_3wfw · 2025-10-26

**Soundness:** 3
**Presentation:** 2
**Contribution:** 3
**Rating:** 4
**Confidence:** 3

**Summary:**

This paper proposes a unified quantum framework to perform sensitivity-based coreset sampling in sublinear time. Applications include low-rank approximations for the Frobenius norm, kernel functions, and tensors, with extensions to subspace approximation, regression, and clustering. Their methods rely mostly on a method where approximate sensitivities are recursively computed and resamples are made based sensitivity estimates and where sampling is aided by Grover search to produce quantum speedups.

**Strengths:**

1. The paper provides quantum speedups for a variety of tasks. They provide the first sampling-based algorithms for several low-rank approximation tasks and algorithmic improvements for clustering and regression.
2. The theoretical work and proofs are solid and written in great detail.
3. Query lower bounds are provided for spectral approximations.

**Weaknesses:**

1. The complexity terms in Tables 1 and 2 seem a bit too generous in some cases. For example, in the first row in Table 2, for the runtime cost for low rank approximations, the table states $$nd^{0.5}k^{0.5},$$ while Theorem C.5 states $$\epsilon^{-1}nd^{0.5}k^{0.5} + nk^{\omega-1}+\epsilon^{-1.5}n^{0.5}k^{1.5}+\epsilon^{-2}d^{0.5}k^{1.5}+\epsilon^{-3}kd.$$ Putting in the assumption that $\epsilon \in \mathcal{O}(1)$ and $k \le \min(n,d)$, we should get the reduction
$$nd^{0.5}k^{0.5} + nk^{\omega-1}+kd.$$
Another example is the missing $d^\omega$ in the subspace approximation compared to Theorem E.2.
2. Minor point, but it might be better to separate classical and quantum prior algorithms into separate columns.
3. Further see questions below.

**Questions:**

1. In the paper, many runtime estimates are presented with the assumption that $\epsilon \in \mathcal{O}(1)$. I wonder how this asssumption is justified? To my knowledge, a lot of Grover-type algorithms in optimization (e.g. SDP solvers) have a decrease in dimension dependency but have an increase error dependence. How does this part compare to classical methods?
2. Are there any theoretical proofs that overestimates can indeed suffice as a replacement for the exact estimate (or indeed, an $\epsilon$-close estimate)?

---

> ### Author Response · Authors · 2025-11-17
>
> We thank the reviewer for their detailed review and for acknowledging the solidity of our theoretical work and the quantum speedups achieved.
>
> * Regarding complexity terms in Tables 1 and 2 being "too generous". The reviewer observes that some terms seem omitted in the complexity bounds presented in the tables. The tables are intended to provide a simplified comparison by focusing on the dominant terms, particularly in the "big data" regime common to machine learning applications, where the number of points $n$ is significantly larger than the dimension $d$ and parameter $k$. In this regime, we treat polynomial dependence on $d$ and $k$ as lower-order terms. This practice is common in randomized numerical linear algebra [CW13, WY25] and quantum algorithms [AG24]. We have added a note to the table captions in the revision to clarify this assumption.
>
> * Specific to Low-Rank Approximation: We have unified the notation so the input matrix is $A\in \mathbb{R}^{d\times n}$ and the sublinear dependence is on $n$. We note that if the goal is coreset construction, Corollary C.2 gives a runtime of $\widetilde O(\epsilon^{-1}n^{0.5}dk^{0.5}+dk^{\omega-1})$. For $\epsilon=O(1)$ and $n\gg d,k$, the $n^{0.5}dk^{0.5}$ term clearly dominates. If the goal is to output the full low-rank approximation, we must spend at least linear time in $n$ and $d$ to write down the output. In this scenario, it is common to assume $n=O(d)$ (e.g., [CMM17]). Under the reduction suggested by the reviewer for Theorem C.5, the runtime is $n^{0.5}dk^{0.5}+dk^{\omega-1}+nk$ if $n=O(d)$, the first term is $O(n^{1.5})$, dominating the latter terms which are $O(n)$. We prioritized the dominant term for simplicity.
>
> * Regarding separating classical and quantum runtimes. We agree that separating the runtimes would enhance presentation. We used the condensed format purely due to space constraints in the main paper.
>
> * Regarding the assumption that $\epsilon=O(1)$. We only assume $\epsilon=O(1)$ in the tables for presentation simplicity. Our algorithms generally have a $\epsilon^{-1}$ factor on the leading term. Crucially, while our advantage diminishes if $\epsilon$ is very small (e.g., $\epsilon=n^{-0.5}$), classical algorithms often have worse dependencies on $\epsilon^{-2}$ in their overall runtime. For instance, the classical low-rank approximation algorithm of [CMM17] has a full runtime of $\widetilde O(nd+\epsilon^{-2}dk^2)$. Setting $\epsilon=n^{-0.5}$ would increase the classical runtime to $\widetilde O(ndk^2)$. Therefore, under reasonable choices of $\epsilon$ (constant or polylogarithmic in $n$), our algorithms offer significant improvements.
>
> * Regarding theoretical proof that overestimates can replace exact estimates. Yes, we rigorously prove this. The crux of our argument relies on the sufficient conditions for the recursive sampling framework, detailed in Assumption B.1 (specifically, consistent total weights and the importance sampling bound). The details are in the proof of Theorem B.3. In short, if $\widetilde w_i$ are overestimates of the exact estimates $w_i$, the sample complexity depends on $\sum_i \widetilde w_i$. If we obtain constant factor overestimates, we simply increase the sample complexity by that constant factor. Overestimation is necessary to maintain the approximation guarantee; underestimation would violate the sampling requirements.
>
> References
>
> [ADW22] Apers and de Wolf. Quantum speedup for graph sparsification, cut approximation, and laplacian solving. SIAM Journal on Computing, 2022.
>
> [AG24] Apers and Gribling. Quantum speedups for linear programming via interior point methods. QIP 2024.
>
> [CMM17] Cohen, Musco and Musco. Input sparsity time low-rank approximation via ridge leverage score sampling. SODA 2017.
>
> [CW13] Clarkson and Woodruff. Low rank approximation and regression in input sparsity time. STOC 2013.
>
> [XCLJ23] Xue, Chen, Li and Jiang. Near-optimal quantum coreset construction algorithms for clustering. ICML 2023.
>
> [WY25] Woodruff and Yasuda. Root ridge leverage score sampling for $\ell_p$ subspace approximation. FOCS 2025.

---

> > ### Comment · Reviewer_3wfw · 2025-11-27
> >
> > I thank the authors for their feedback and response. Given that my main concern was the discrepancy in terms of the algorithmic complexity, I'm happy to raise my score to 6.

---

### Official Review · Reviewer_psLS · 2025-10-31

**Soundness:** 4
**Presentation:** 4
**Contribution:** 3
**Rating:** 8
**Confidence:** 3

**Summary:**

The paper proposes a quantum computing approach to sensitivity sampling for coreset creation aimed at a range of machine learning tasks including clustering, linear regression, and low rank matrix approximation.

**Strengths:**

The paper builds on quantum leverage score sampling approach based on Grover’s search introduced by Apers and Gribling, and extends it to sensitivity sampling. Then, they show that the resulting coreset selection method leads to improvements in complexity for k-median and k-mean clustering, linear regression, and low-rank matrix approximation, offering a clear advanced over the current state-of-the-art. The fact that the proposed algorithm’s complexity does not depend on data-specific parameters is another strong point.

**Weaknesses:**

The paper operates within the QRAM-based model of computation, which, while common in the field, is likely to limit its practical applicability on medium-term hardware.

**Questions:**

N/A

---

> ### Author Response · Authors · 2025-11-17
>
> We thank the reviewer for their positive assessment and recognition of our work as technically strong.
>
> * Regarding the QRAM model and practical applicability. The reviewer correctly points out that the QRAM model limits practical applicability on medium-term hardware. We acknowledge this limitation. However, we want to emphasize the distinction between time complexity and query complexity. While the sublinear time complexity relies on QRAM, the query complexity of our algorithm is independent of the QRAM assumption. This represents a robust theoretical advancement in understanding the efficiency of these algorithms. We have added a new Section J (Table 3) summarizing the query complexities for different problems to highlight this contribution.

---

### Official Review · Reviewer_y2Nv · 2025-10-31

**Soundness:** 3
**Presentation:** 2
**Contribution:** 2
**Rating:** 6
**Confidence:** 3

**Summary:**

The authors study quantum speedups for *coreset construction problem*, i.e., given a set $A$ and a cost function $\text{cost}(\cdot,\cdot)$ on $A\times X$, find a subset $B$ such that $\sum_{b\in B} w_b f(b, x) =(1\pm \epsilon) \sum_{a\in A}f(a,x)$, which is a significant step in downstream optimization and data selection tasks. The standard approach performs a weighted sampling procedure where each point is drawn with probability $s_i = \max_{x\in X} \frac{cost(A_i,x)}{cost(A,x)}$, called *sensitivity*, and assigns weight $1/s_i$ to the drawn point which has sample complexity $O(\epsilon^{-2}(\sum_{i=1}^n s_i) \text{dim X} \log(1/\delta))$ to find an $\epsilon$ coreset with probability at least $1-\delta$. Since the computation of $s_i$ requires a pass over each element of $A$, this process has $O(n)$ dependency. The goal of the paper is to reduce $n$ dependency to sub-linear by using quantum computing techniques, mainly quantum amplitude amplification/Grover's search. The authors give hybrid algorithms to construct coreset for various problems including clustering, regression and low rank approximation.

The main ingredient seems to be the recursive procedure to do weighted sampling originally used by Apers &Gribling in the context of quantum linear programming. This procedure is used as a submodule in the various classical algorithms and the authors obtained speedups either in $n$ or problem dimension $d$ under QRAM model compared to both classical and quantum algorithms. The given algorithms achieve runtimes independent of data-specific parameters unlike the previous work.

**Strengths:**

The studied problems are relevant and important for modern machine learning algorithms. As the bottleneck to obtain sublinear runtimes is due to sensitivity computation, it is natural to consider quantum algorithms which are known to give speedups for similar problems. Although the quantum part of the algorithm is previously known, the paper has still include novelty as the previous work did not address the problems considered here. Some of submodules such as quantum weighted sampling could be seen as an independent tool that could be used for different problems in the future although it was developed in previous works.

Each algorithm has been given in detail in the appendix as pseudocode and the analysis seem to be thorough and correct. The paper does not provide empirical evidence, yet at this stage the proposed algorithms cannot be implemented on the existing hardware because of error overheads. However, the level of the rigor in the analysis is sufficient to compensate this.

The authors also proposed open questions for further directions which would be appreciated by the community for further research on the subject.

**Weaknesses:**

One major weakness is that the paper assumes access to quantum read/classical-write (QRAM) which could in general change the over all runtime of the quantum algorithms significantly, however the paper does not discuss the effect of this structure. Without this input model, the quantum state needs to be prepared from scratch which adds additional loop to each of these algorithms. It seems that the previous paper by Apers & Gribling also have this input model, and the remark in that paper states that this assumption can be removed by paying poly cost in *time* which can potentially remove the claimed speedup over classical algorithms. Note that this is not  very important in Apers & Gribling paper as their claim is on the *query complexity* not time. If this remark is not applicable in this paper, this has to be discussed and justified properly.

It is also known that realizing QRAM structure might be challenging. From the theoretical perspective, it is completely valid to assume this structure; however this also needs to be highlighted in the main text. The paper also claims that the runtime is independent of data set parameters. However, the loading to QRAM initially might have data dependent overhead. Similarly input representation depends on how QRAM is implemented, therefore it becomes important whether we can load this set to QRAM.


It is also not clear how Grover’s algorithm is incorporated into different classical algorithms. Although the main body introduces the Grover’s algorithm, it is not clear at intuitive level how each application makes use of QSample routine,. Along the lines of my previous comment, it seems that QRAM model also makes the algorithms somewhat straightforward (not to trivialize the contribution of the paper but I would think without QRAM, the algorithms might need significant modification).

**Questions:**

Questions/Suggestions

1. The runtime $T_\text{sensitivity}$ does not seem to be instantiated for the specific problems in the main text and details seem to be postponed. Can authors give explicit bounds for this for different applications?
2. Based on my comment above, how would QRAM assumption change the runtime and the algorithm overall?
3. Is $\delta$ around lines 60 failure probability of the algorithm?
4. Around lines 212, I believe the algorithm is to repeat Grover’s algorithm m times where each run takes $O(\sqrt{n/m})$ calls?
5. Lemma A.14 is the result for main quantum procedure, but it is not referenced properly from the main text. The runtime does not seem to be trivial by itself.
6. Around line 221, if p_i does not form a distribution, the paper should not say probabilities but it might say a measure proportional to weights p_i. Otherwise, it is a confusing statement.
7. The paper claims sublinear runtime but some results are sublinear in $n$ and some of them are in $d$. This should be clarified at the beginning.

---

> ### Author Response · Authors · 2025-11-17
>
> We thank the reviewer for recognizing the relevance of the problems studied and the thoroughness of our analysis. We address the concerns regarding the QRAM assumption and runtime specifications below.
>
> * Regarding the QRAM assumption and its runtime implications. The reviewer correctly notes our reliance on the QRAM model. We wish to clarify the distinction between *time* complexity and *query* complexity in our results. To achieve sublinear *time* complexity, the QRAM assumption is indeed necessary, as pointed out in prior work (e.g., Section 1.2 in [ADW22]). This is required to simulate query access to the random strings needed for the derandomization process.
>
>     However, our algorithms also achieve sublinear *query* complexity, which is independent of the QRAM assumption. Query complexity measures the number of accesses made to the input data. This is a significant theoretical contribution that holds even if QRAM is not practically realized. We have summarized these query complexity results in the new Section J, Table 3 of the revised draft.
>
>     As noted by [AG24], dropping the QRAM assumption might incur polynomial overhead in time complexity, but the sublinear query complexity is maintained. We have added a remark in our discussion of the quantum computation model (Section I) acknowledging the practical challenges of realizing QRAM while emphasizing the robustness of our query complexity improvements.
>
> * Regarding ${\cal T}\_{\rm sensitivity}$ for different applications. We apologize for postponing the definitions in the initial draft as they were explained in detail in Section B. We have clarified the values of ${\cal T}_{\rm sensitivity}$ (the time to compute sensitivities) for the applications studied. They are summarized below:
>
> | Application                | ${\cal T}_{\rm sensitivity}$ |
> |----------------------------|------------------------------|
> | $(k,p)$-Clustering         | $\widetilde O(d)$            |
> | $\ell_{p\neq 2}$ Regression | $\widetilde O_p(d)$         |
> | $(k,p)$-Subspace Approx.   | $\widetilde O(d)$            |
>
>
> * Regarding specific questions.
>
>     * Line 60: Yes, $\delta$ is the failure probability. We have clarified this definition in the draft.
>
>     * Line 212 (Grover Search): Yes, the algorithm uses repeated Grover search for $m$ items, where each run takes $O(\sqrt{n/m})$ calls. This is a standard technique utilized in [ADW22, XCLJ23, AG24].
>
>     * Lemma A.14: We have revised the draft and properly referenced this lemma in Section 2.1.
>
>     * Use of $p_i$: We have revised the notation to clarify that $p_i\in [0,1]$ are values and each item is sampled with probability.
>
>     * Sublinear in $n$. To unify the presentation, we have adjusted the notation for low-rank approximation so the input matrix is $A\in \mathbb{R}^{d\times n}$. This ensures that all analyzed algorithms exhibit sublinear dependence on $n$ in both queries and runtime.
>
> References
>
> [ADW22] Apers and de Wolf. Quantum speedup for graph sparsification, cut approximation, and laplacian solving. SIAM Journal on Computing, 2022.
>
> [XCLJ23] Xue, Chen, Li and Jiang. Near-optimal quantum coreset construction algorithms for clustering. ICML 2023.
>
> [AG24] Apers and Gribling. Quantum speedups for linear programming via interior point methods. QIP 2024.

---

### Official Review · Reviewer_fi9D · 2025-11-01

**Soundness:** 2
**Presentation:** 2
**Contribution:** 3
**Rating:** 2
**Confidence:** 4

**Summary:**

This submission presents a sublinear-time quantum algorithm for constructing a coreset via sensitivity sampling, which can be used as a subroutine to speed up other problems such as k-median and k-means clustering, $\ell_p$ regression, and low-rank approximation. The main source of quantum speedup comes from a variant of Grover's algorithm for sampling a subset, which relies on an oracle for computing the sensitivity (or Lewis weights in some applications).

**Strengths:**

This manuscript contains a lot of results that improve on previous ones in a certain regime. The main technical contribution is from the classical side, where the authors focused on how to efficiently estimate sensitivity or weights in different applications so that the quantum algorithm can use them. The results are summarized in theorems which are supported by proofs.

**Weaknesses:**

First of all, this submission is difficult to read. Maybe this is natural since it contains too many results. I appreciate the high-level technical overview in Section 2; however, I found that the technical overview is still very technical. Reading Section 2 doesn't help with grasping the intuition; I would rather directly read the appendices. I think the authors should work on improving the presentation in this section.

In addition, the presentation lacks clarity. For example, in Line 57, is the number of samples here the size of the coreset? In Line 84, does the final sample size mean the same thing? In Lines 164-165, is the $s$ (upper bound on the total weight) the same as the $s$ in Line 84, and also the $s$ in Line 315? In Lines 220-222, the description of the sampling process is rather confusing. Why not just use the language of "sampling a subset" as in Apers & Gribling (2024)? In Line 233, the matrix $S$ is not defined. In Lines 243-244, "uniformly sample half of the points from $A'$" should be stated more precisely here.

Besides the presentation issues, there's a more critical technical flaw. In the statement of the theorems, a classical oracle that outputs an estimation of sensitivity (or weights) is assumed. However, in Grover's algorithm, one needs a quantum oracle. If a classical subroutine (a classical white-box) is given, additional work needs to be done: one first needs to turn the classical subroutine into a quantum circuit, and then clean the garbage during the conversion by uncomputing the ancillas. It could be more complicated when the classical subroutine generates approximations of the true value with a success probability (like Algorithms 4, 5, 6). In this case, the corresponding quantum oracle will produce a superposition of an estimate (with bounded approximation ratio) and some garbage state. Also, the portion corresponding to the desired estimate itself is a superposition of different approximations. Maybe I am wrong, but I don't think directly plugging this oracle into Grover's algorithm will produce the correct state. Some additional steps need to be done, including concentrating the estimate using the median trick. Also, a careful error analysis of the quantum algorithm with this concentrated oracle should be done to prove the correctness of the entire approach.

There are other minor things:
1. In the second and third bullet points of the abstract, $n, d, k$ are not defined.
2. In Theorem 1.1, a quantum algorithm is randomized, so there's no need to say a "randomized" quantum algorithm
3. Line 202, further extend -> we further extend

**Questions:**

I would be willing to raise my score once my concerns in the previous section can be addressed without a substantial revision.

---

> ### Author Response · Authors · 2025-11-17
>
> We appreciate the reviewer's insightful comments and the opportunity to clarify the technical aspects of our approach.
>
> * Regarding the technical concern on classical oracles and the QRAM model: The reviewer raises a critical point regarding the conversion of classical subroutines (which we refer to as "classical oracles") into quantum circuits, specifically concerning the handling of garbage states and superpositions of estimates.
>
>      Our work operates within the standard Quantum Random Access Memory (QRAM) model, which is standard in the quantum sampling literature [ADW22, XCLJ23, AG24]. In this model, data is stored classically but can be read coherently in superposition.
>
>      The reviewer is correct that directly converting a randomized classical oracle into a quantum circuit can significantly complicate the analysis due to garbage states. However, we circumvent this issue by adopting the "derandomization" approach established by Apers and de Wolf [ADW22]. This technique views a randomized algorithm as a deterministic algorithm supplemented by a random string. [ADW22] demonstrated that a quantum algorithm with $k/2$ queries cannot distinguish a uniformly random string from a $k$-wise independent string. This allows us to use $k$-wise independent hash functions to derandomize the algorithm. As formalized in Claim 1 of [ADW22] and Lemma 3.12 of [AG24]:
>
>     **Lemma** *Any quantum algorithm with runtime $q$ using a uniformly random string can be simulated by a quantum algorithm with runtime $\widetilde O(q)$ without a random string, using an additional QRAM of $\widetilde O(q)$ bits.*
>
>     This means we can treat the classical sensitivity estimation routines as deterministic. They can then be implemented as quantum circuits using standard quantum arithmetic and Toffoli gates. This approach avoids the complications of handling garbage states associated with randomized estimates, thus addressing the reviewer's concern about the correctness of the approach.
>
> * Regarding presentation and clarity. We thank the reviewer for pointing out the lack of clarity, particularly in Section 2. We have significantly revised this section to improve the intuition and presentation, especially concerning the integration of Grover search to speed up the sampling process. We have also clarified the notation for $s$. Throughout the revised manuscript, we now consistently use $s$ to denote the size of the coreset, $S$ for the total sensitivity, and ${\rm sum}(w)$ for the upper bound on the sum of weights in general weighted sampling.
>
> * Regarding minor points. We have addressed the minor issues:
>     * We have defined $n$ (number of points), $d$ (dimension) and $k$ (clustering/rank) in the abstract for the second and third bullet points.
>
>     * We have removed the term "randomized" before "quantum algorithm" in all lemma and theorem statements, as quantum algorithms are inherently randomized.
>
>     * The typo in line 202 ("further extend") has been corrected.
>
> We hope these clarifications and revisions address the reviewer's concerns.
>
> References
>
> [ADW22] Apers and de Wolf. Quantum speedup for graph sparsification, cut approximation, and laplacian solving. SIAM Journal on Computing, 2022.
>
> [XCLJ23] Xue, Chen, Li and Jiang. Near-optimal quantum coreset construction algorithms for clustering. ICML 2023.
>
> [AG24] Apers and Gribling. Quantum speedups for linear programming via interior point methods. QIP 2024.

---

### Author Response · Authors · 2025-11-17
**Global response**

We sincerely thank all the reviewers for their thoughtful feedback and constructive comments. We are encouraged that the reviewers found our contributions relevant for modern machine learning (y2Nv), technically strong (psLS), the theoretical work solid and detailed (3wfw), and recognized our improvements over previous results (fi9D).
We have addressed the concerns raised, particularly regarding the QRAM assumption, the clarity of the complexity bounds, and the presentation of the classical-to-quantum conversion. All the revisions are marked in red in our revised manuscript.

---

### Author Response · Authors · 2025-12-01
**Summary of the discussion period**

Dear Area Chair,

Per the unprecedented situation about open review information leak, ICLR’s decision to revert scores pre-rebuttal and reassign AC, we would like to document our discussion with reviewers during the discussion period. In particular, we would like to note for **Reviewer 3wfw**, after we have addressed their concern on the discrepancy of terms for algorithmic complexity, they are satisfied by our response and **has raised the score from 4 to 6**, as documented in the reviewer’s response in the discussion.

We hope this further clarifies the reviewers’ stances regarding our rebuttal, thank you!

---

### Meta-Review · Area_Chair_5LqS · 2025-12-18

**Summary:**

Reviewers raised a major technical concern that the algorithm requires a classical sampling algorithm, thus the claimed guarantees would not follow immediately in practical applications. Reviewers also found that the presentation of the paper needs to be improved.

**Reviewer Concerns:**

Authors added a few comments on how the classical sampling algorithm may work with the quantum approach. The correctness is unclear .

**Reviewer Scores:**

Likely unchanged.

---

### Decision · Program_Chairs · 2026-01-26

Reject